# Constrained Stochastic Multi-Objective Optimization

## Abstract

This paper aims to address the constrained stochastic multi-objective optimization (CSMOO) problem, where both objectives and constraints involve expectations over random variables. Firstly, to tackle the computational challenge of exact expectation evaluations, we propose two approximation schemes: stochastic approximation, which updates the entire problem using new samples at each iteration, and block stochastic approximation, which updates only subsets of variables iteratively. Secondly, to handle potential infeasibility in the surrogate problems, we develop two strategies: a feasible update reformulation and a rigorously justified penalty scheme equivalent to the original problem. Our framework provides asymptotic convergence guarantees to stationary points that satisfy the Fritz John condition. Experiments on synthetic and real-world wireless communication benchmarks demonstrate superior convergence, stability, and constraint satisfaction over state-of-the-art methods.

## 1 Introduction

Multi-objective optimization (MOO) aims to identify Pareto front (PF) that balances multiple conflicting objectives, a challenge pervasive in real-world applications spanning materials science (Gopakumar et al., 2018; Shi et al., 2023; Baird et al., 2023), wireless communication systems (Sun et al., 2016; Pan et al., 2024; Wang et al., 2024), recommendation systems (Kermany, 2024; Zheng and Wang, 2022; Lin et al., 2018), and industrial design (Tanabe and Ishibuchi, 2020; Wang et al., 2011; Xu et al., 2021). Despite the remarkable successes achieved in unconstrained[1] deterministic MOO problems based on fully observable environments, many real-world applications still face significant challenges: i) data uncertainty: stochastic processes appear naturally across various domains, such as random channel fading in wireless communication systems and stochastic volatility in complex financial market dynamics; ii) critical limitations: constraints are usually imposed in many practical systems including resource limitation in industrial internet of things systems and latency-critical requirements in safety-sensitive autonomous driving. Recent years have witnessed significant advancements in these domains. Stochastic multi-objective gradient methods and their algorithmic variants have emerged in the realm of unconstrained stochastic MOO (Liu and Vicente, 2021; Zhou et al., 2022; Fernando et al., 2023). Meanwhile, for constrained deterministic settings, some methodologies that strategically balance gradient descent directions between conflicting objectives and constraint satisfaction mechanisms have emerged (Fliege and Vaz, 2016). Based on this, a compelling research challenge in generalizing existing algorithms to constrained stochastic MOO (CSMOO) has recently emerged. This emerging field is still far from mature compared to other established MOO paradigms.

In this paper, we propose two novel strategies for constrained stochastic multi-objective optimization (CSMOO), targeting general problems in which both the objectives and constraints involve expectations of nonconvex functions. The main contributions of this paper are summarized as follows

- We propose a family of convex quadratic surrogate functions to approximate the expectations in the CSMOO problem, thereby alleviating the computational intractability of exact expectation evaluation. By employing iterative approximation procedures, we rigorously

---

[1]In this paper, "unconstrained" refers to problems with no constraints or those where the feasible set is a deterministic convex set, whereas "constrained" denotes scenarios involving nonconvex constraints.

establish that the proposed method achieves asymptotic consistency in both function values and gradients with respect to the original expectation functions.

- We propose two strategies to overcome the infeasibility of approximation problems caused by estimation inaccuracies: (i) reformulating the update step as a feasible update subproblem, and (ii) developing a rigorously justified penalty function scheme equivalent to the primal problem. We further design a tailored optimization algorithm and provide convergence analysis with asymptotic guarantees.

- We validate our proposed algorithm through synthetic benchmarks and real-world wireless communication problems, demonstrating superior stability and constraint satisfaction over state-of-the-art methods.

**Notations.** We adopt the following notation conventions. $\Delta_m$ denotes the $m$-dim simplex, $\Delta_m = \{\boldsymbol{\lambda} | \sum_{i=1}^m \lambda_i = 1 \text{ and } \lambda_i \geq 0, \forall i \in [m], \boldsymbol{\lambda} \in \mathbb{R}^m\}$. $[m] = \{1, \ldots, m\}$ is the index set. For a differentiable function $f(\boldsymbol{x})$, $\nabla_r f(\boldsymbol{x})$ denotes the gradient with respect to the $r$-th block of $\boldsymbol{x} \triangleq (\boldsymbol{x}_1, \boldsymbol{x}_2, \cdots, \boldsymbol{x}_k)$.

## 2 RELATED WORKS

Multi-objective optimization has a long research history, with significant contributions from multi-objective evolutionary algorithms (MOEAs). However, MOEAs struggle to scale with machine learning problems. In contrast, gradient-based machine learning, which efficiently handles millions of decision variables, is gaining popularity.

### 2.1 GRADIENT-BASED MULTI-OBJECTIVE OPTIMIZATION

Gradient-based Multi-Objective Optimization (MOO) methodologies are generally categorized into two paradigms: those yielding arbitrary Pareto optimal solutions and those targeting preference-specific outcomes. Early approaches, such as DWA (Liu et al., 2019a), UW (Kendall et al., 2018), and meta-learning frameworks like MOML (Ye et al., 2021; 2024a) and FORUM (Ye et al., 2024b), converge to the Pareto front but lack the steer-ability to align with specific user needs. Conversely, preference-based methods—exemplified by PMTL (Lin et al., 2019), EPO (Mahapatra and Rajan, 2020; 2021), and PMGDA (Zhang et al., 2024)—utilize preference vectors to guide the search toward desired trade-off regions. However, this precision often incurs significant computational overhead due to the requisite manipulation of multiple gradients.

### 2.2 CONSTRAINED MULTI-OBJECTIVE OPTIMIZATION (CMOO)

Evolutionary algorithms have been one of the primary approaches driving the development of the Constrained Multi-Objective Optimization (CMOO) field. Recent advancements in this field include employing deep reinforcement learning to assist operators (Ming et al., 2024a), using multi-task learning for knowledge transfer (Ming et al., 2024b), and developing co-evolutionary frameworks (Tian et al., 2020). Other strategies have focused specifically on the challenge of handling constraints within both decision and objective spaces (Liu and Wang, 2019).

Lately, gradient-based methods have gained traction as a promising alternative. A notable example is the adaptation of the Multiple Gradient Descent Algorithm (MGDA) for constrained problems, which achieved a convergence rate of $O(1/T)$ (Li et al., 2025). This development highlights a growing trend towards leveraging gradient information for more efficient CMOO.

### 2.3 CONSTRAINED STOCHASTIC MULTI-OBJECTIVE OPTIMIZATION (CSMOO)

Recent scholarship on Constrained Stochastic Multi-Objective Optimization (CSMOO) represents a paradigm shift from deterministic scalarization techniques to rigorous, vector-valued stochastic approximation methods (Deb, 2018; Geiersbach and Milz, 2025). While earlier research predominantly relied on evolutionary heuristics, current literature emphasizes the theoretical underpinnings of Stochastic Multi-Gradient (SMG) descent (Yang et al., 2024) and Primal-Dual frameworks. A pivotal advancement is the rigorous formulation of stationarity conditions in noise-corrupted environments;

specifically, the transition from classical KKT conditions to Approximate KKT (AKKT) sequences and Pareto Stationarity measures, which are essential when standard constraint qualifications cannot be verified. To address the systemic bias inherent in non-linear multi-objective gradient estimation, recent algorithms have incorporated Stochastic Mirror Descent (MSMD) (Yang et al., 2024) and Double Sampling (MoDo) strategies (Chen et al., 2023), offering provable convergence rates of $\mathcal{O}(1/\sqrt{T})$ for non-convex landscapes (Chen et al., 2023).

## 3 PRELIMINARIES

**Constrained Stochastic Multi-Objective Optimization (CSMOO).** We consider the problem $\min_{\boldsymbol{x} \in \mathcal{X}} \boldsymbol{f}(\boldsymbol{x})$ s.t. $\boldsymbol{g}(\boldsymbol{x}) \leq 0$, where $\boldsymbol{f}(\boldsymbol{x}) \triangleq \mathbb{E}_{\boldsymbol{\xi}}[\boldsymbol{f}(\boldsymbol{x}, \boldsymbol{\xi})]$ involves $m$ objectives and $\boldsymbol{g}(\boldsymbol{x}) \triangleq \mathbb{E}_{\boldsymbol{\xi}}[\boldsymbol{g}(\boldsymbol{x}, \boldsymbol{\xi})]$ contains $n$ constraints. Here, $\boldsymbol{x} \in \mathcal{X} \subseteq \mathbb{R}^d$ is the decision variable and $\boldsymbol{\xi}$ denotes the random state. Throughout the paper, we assume: (1) $\mathcal{X}$ is compact and convex with diameter $D$; (2) All functions $f_i, g_j$ (and their stochastic counterparts) are continuously differentiable; (3) Both the functions and their gradients are Lipschitz continuous (i.e., the problem is smooth and Lipschitz).

**Pareto Optimality.** We denote $\boldsymbol{f}(\boldsymbol{x}_1) \prec \boldsymbol{f}(\boldsymbol{x}_2)$ (dominance) if $f_i(\boldsymbol{x}_1) \leq f_i(\boldsymbol{x}_2), \forall i$ with at least one strict inequality. A solution $\boldsymbol{x}^*$ is *Pareto optimal* if no $\boldsymbol{x}$ exists such that $\boldsymbol{f}(\boldsymbol{x}) \prec \boldsymbol{f}(\boldsymbol{x}^*)$, and *weakly Pareto optimal* if no $\boldsymbol{x}$ strictly dominates it in all objectives. The *Fritz John (FJ)* necessary condition for a weakly Pareto optimal $\boldsymbol{x}^*$ requires the existence of non-negative, non-zero vectors $\boldsymbol{\nu}, \boldsymbol{\mu}$ such that:

$$\boldsymbol{0} \in \sum_{i=1}^{m} \nu_i \nabla f_i(\boldsymbol{x}^*) + \sum_{j=1}^{n} \mu_j \nabla g_j(\boldsymbol{x}^*) + N_{\mathcal{X}}(\boldsymbol{x}^*) \quad \text{and} \quad \mu_j g_j(\boldsymbol{x}^*) = 0, \forall j, \tag{1}$$

where $N_{\mathcal{X}}(\boldsymbol{x}^*)$ is the normal cone to $\mathcal{X}$ at $\boldsymbol{x}^*$.

**Tchebycheff Scalarization.** To handle the multi-objective nature, we employ the Tchebycheff scalarization, formulating the problem as:

$$\min_{\boldsymbol{x} \in \mathcal{X}} \max_{1 \leq i \leq m} \left\{ \lambda_i (\mathbb{E}_{\boldsymbol{\xi}}[f_i(\boldsymbol{x}, \boldsymbol{\xi})] - z_i) \right\} \quad \text{s.t.} \quad \mathbb{E}_{\boldsymbol{\xi}}[\boldsymbol{g}(\boldsymbol{x}, \boldsymbol{\xi})] \leq 0, \tag{2}$$

where $\boldsymbol{\lambda} \in \Delta_m$ is a preference vector and $\boldsymbol{z}$ is the ideal objective vector with $z_i = \inf_{\boldsymbol{x}} \mathbb{E}[f_i]$.

Detailed preliminary definitions and assumptions are provided in Appendix A.

## 4 METHODOLOGY

As shown in Problem (2), when addressing the general CSMOO problem, the Tchebycheff scalarization approach encounters two major challenges: i) potential infeasibility caused by inaccurate expectation estimation, and ii) the non-differentiability of the objective function. To address these challenges, this section first introduces estimation methods for function values and gradients to construct quadratic surrogate functions, and then presents two methods for Problem (2).

### 4.1 ASYMPTOTICALLY CONSISTENT ESTIMATOR

To mitigate the computational burden of estimating mathematical expectations, we adopt a stochastic approximate strategy to efficiently approximate the expectation values without relying on exhaustive sampling. This approach fundamentally differs from conventional Monte Carlo methods, which require a large number of samples to compute the sample average at every step. Before presenting the update procedure, we first state the following assumptions on the underlying stochastic processes.

**Assumption 1** (Assumptions on the instantaneous values and gradients). *Let $\mathcal{F}_t$ be the $\sigma$-algebra generated by the collection of past realizations of $\boldsymbol{x}$ and $\boldsymbol{\xi}$ up to iteration $t - 1$, i.e. $\mathcal{F}_t = \{(\boldsymbol{x}_\tau, \boldsymbol{\xi}_\tau)\}_{\tau < t}$. The instantaneous values and gradients are unbiased with bounded variance, i.e.,*

*1)* $\mathbb{E}\left[ f_i(\boldsymbol{x}_t) - f_i(\boldsymbol{x}_t, \boldsymbol{\xi}_t) \Big| \mathcal{F}_t \right] = 0, \mathbb{E}\left[ |f_i(\boldsymbol{x}_t) - f_i(\boldsymbol{x}_t, \boldsymbol{\xi}_t)|^2 \Big| \mathcal{F}_t \right] \leq \varepsilon^2 < \infty, \forall t.$

*2)* $\mathbb{E}\left[ \nabla f_i(\boldsymbol{x}_t) - \nabla f_i(\boldsymbol{x}_t, \boldsymbol{\xi}_t) \Big| \mathcal{F}_t \right] = \boldsymbol{0}, \mathbb{E}\left[ \|\nabla f_i(\boldsymbol{x}_t) - \nabla f_i(\boldsymbol{x}_t, \boldsymbol{\xi}_t)\|^2 \Big| \mathcal{F}_t \right] \leq \varepsilon^2 < \infty, \forall t.$

3) $\mathbb{E}\left[g_j(\boldsymbol{x}_t) - g_j(\boldsymbol{x}_t, \boldsymbol{\xi}_t)\Big|\mathcal{F}_t\right] = 0, \mathbb{E}\left[\left|g_j(\boldsymbol{x}_t) - g_j(\boldsymbol{x}_t, \boldsymbol{\xi}_t)\right|^2\Big|\mathcal{F}_t\right] \leq \varepsilon^2 < \infty, \forall t.$

4) $\mathbb{E}\left[\nabla g_j(\boldsymbol{x}_t) - \nabla g_j(\boldsymbol{x}_t, \boldsymbol{\xi}_t)\Big|\mathcal{F}_t\right] = \boldsymbol{0}, \mathbb{E}\left[\left\|\nabla g_j(\boldsymbol{x}_t) - \nabla g_j(\boldsymbol{x}_t, \boldsymbol{\xi}_t)\right\|^2\Big|\mathcal{F}_t\right] \leq \varepsilon^2 < \infty, \forall t.$

Assumption 1 requires the instantaneous values and gradients to be unbiased and have bounded variance, which is weaker than the gradually decreasing variance in Assumption 5.2 of Liu and Vicente (2021). Besides, this assumption is readily satisfied if $\boldsymbol{\xi}_1, \boldsymbol{\xi}_2, \cdots$ are bounded and identically distributed.

**Values Estimation.** At iteration $t$, a new realization of the random vector $\boldsymbol{\xi}_t$ is generated, after which the estimations of the expectations $f_i(\boldsymbol{x}_t)$ and $g_j(\boldsymbol{x}_t)$ are iteratively updated as

$$\hat{f}_i(\boldsymbol{x}_t) = \rho_t f_i(\boldsymbol{x}_t, \boldsymbol{\xi}_t) + (1 - \rho_t)\hat{f}_i(\boldsymbol{x}_{t-1}), \quad \hat{g}_j(\boldsymbol{x}_t) = \rho_t g_j(\boldsymbol{x}_t, \boldsymbol{\xi}_t) + (1 - \rho_t)\hat{g}_j(\boldsymbol{x}_{t-1}), \quad (3)$$

where the initial estimation is set as $\hat{f}_i(\boldsymbol{x}_{-1}) = \hat{g}_j(\boldsymbol{x}_{-1}) = 0$.

**Gradients Estimation.** Compared to estimating the function values, computing the gradients $\nabla f_i(\boldsymbol{x}_t)$ and $\nabla g_j(\boldsymbol{x}_t)$ usually incurs significantly higher computational overhead. To address this, we develop stochastic approximation and block stochastic approximation strategies tailored to problems of different dimensional scales.

*Stochastic approximation strategy.* When the dimension of $\boldsymbol{x}_t$ is small, the instantaneous gradients $\nabla f_i(\boldsymbol{x}_t, \boldsymbol{\xi}_t)$ and $\nabla g_j(\boldsymbol{x}_t, \boldsymbol{\xi}_t)$ can be efficiently obtained. In this case, the expectations of the gradients, $\nabla f_i(\boldsymbol{x}_t), \nabla g_j(\boldsymbol{x}_t)$, are estimated in an iterative manner

$$\hat{\nabla} f_i(\boldsymbol{x}_t) = \rho_t \nabla f_i(\boldsymbol{x}_t, \boldsymbol{\xi}_t) + (1 - \rho_t)\hat{\nabla} f_i(\boldsymbol{x}_{t-1}), \quad \hat{\nabla} g_j(\boldsymbol{x}_t) = \rho_t \nabla g_j(\boldsymbol{x}_t, \boldsymbol{\xi}_t) + (1 - \rho_t)\hat{\nabla} g_j(\boldsymbol{x}_{t-1}), \quad (4)$$

where the initial estimation is set as $\hat{\nabla} f_i(\boldsymbol{x}_{-1}) = \hat{\nabla} g_j(\boldsymbol{x}_{-1}) = \boldsymbol{0}$.

*Block stochastic approximation strategy.* When the dimension of $\boldsymbol{x}_t \triangleq (\boldsymbol{x}_{1,t}, \cdots, \boldsymbol{x}_{k,t})$ is large, it becomes prohibitively expensive to compute the instantaneous gradients with respect to all blocks. Therefore, we propose a block stochastic approximation strategy, where only one block is selected for update at each iteration. To fully specify the algorithm, it is necessary to define the block selection rule. Subsequently, we consider two representative choices.

- Cyclic: In the cyclic selection rule, the blocks are updated sequentially in a predetermined order according to
$$r = (t \bmod k) + 1. \quad (5)$$

- Randomized: In the randomized selection rule, at iteration $t$, a block is chosen independently of the past with probability
$$\mathbb{P}(\text{block } r \text{ being selected}) = p_{r,t} \geq p_{\min} > 0. \quad (6)$$
where $p_{\min}$ guarantees that every block has a strictly positive chance of being selected.

In this case, the expectations of the gradients, $\nabla f_i(\boldsymbol{x}_t) \triangleq (\nabla_1 f_i(\boldsymbol{x}_t), \nabla_2 f_i(\boldsymbol{x}_t), \cdots, \nabla_k f_i(\boldsymbol{x}_t))$ and $\nabla g_j(\boldsymbol{x}_t) \triangleq (\nabla_1 g_j(\boldsymbol{x}_t), \nabla_2 g_j(\boldsymbol{x}_t), \cdots, \nabla_k g_j(\boldsymbol{x}_t))$, are estimated in an iterative manner

$$\hat{\nabla}_r f_i(\boldsymbol{x}_t) = \begin{cases} \rho_t \nabla_r f_i(\boldsymbol{x}_t, \boldsymbol{\xi}_t) + (1 - \rho_t)\hat{\nabla}_r f_i(\boldsymbol{x}_{t-1}), & \text{if block } r \text{ is selected,} \\ \hat{\nabla}_r f_i(\boldsymbol{x}_{t-1}), & \text{otherwise.} \end{cases}$$
$$\hat{\nabla}_r g_j(\boldsymbol{x}_t) = \begin{cases} \rho_t \nabla_r g_j(\boldsymbol{x}_t, \boldsymbol{\xi}_t) + (1 - \rho_t)\hat{\nabla}_r g_j(\boldsymbol{x}_{t-1}), & \text{if block } r \text{ is selected,} \\ \hat{\nabla}_r g_j(\boldsymbol{x}_{t-1}), & \text{otherwise.} \end{cases} \quad (7)$$

To guarantee the asymptotic consistency, we make the following assumptions on the surrogate instantaneous gradient momentum parameter $\{\rho_t\}$.

**Assumption 2** (Assumptions on $\{\rho_t\}$). *The surrogate instantaneous gradient momentum parameter $\rho_t \in [0, 1]$ are chosen such that*

1. $\lim_{t \to \infty} \rho_t = 0, \sum_{t=0}^{\infty} \rho_t = \infty, \sum_{t=0}^{\infty} \rho_t^2 < \infty,$

2. $\sum_{t=0}^{\infty} \|\boldsymbol{x}_t - \boldsymbol{x}_{t-1}\|^2 / \rho_t < \infty$.

Assumption 2 is readily satisfied under the prescribed parameter constraints. Consider the update rule $\boldsymbol{x}_t = \boldsymbol{x}_{t-1} + \gamma_t \boldsymbol{d}_t$, where $\gamma_t$ is the learning rate and $\boldsymbol{d}_t$ is a bounded update direction. Suppose $\rho_t$ and $\gamma_t$ take the forms $\rho_t = 1/t^a$ and $\gamma_t = 1/t^b$, respectively. Then, restricting the exponents $a$ and $b$ to the ranges $0.5 < a < 1$ and $b > 0.5 + 0.5a$ (equivalently, $2b - a > 1$) guarantees the assumption holds. For example, setting $a = 0.6, b = 0.9$ meets these conditions, since $2b - a = 1.2$ lies within the required interval. This illustrates both the feasibility and flexibility of the parameter choices in practice.

Finally, the following theorem establishes the asymptotic consistency, demonstrating that the expectation function in the original problem can be accurately recovered in the asymptotic regime.

**Theorem 1** (Asymptotic consistency and non-asymptotic analysis). *Based on the value estimation in Equation* (3)*, gradient estimation in Equations* (4) *and* (7)*, and Assumptions 1, 2 and 4, we have*

$$\lim_{t \to \infty} \left| \hat{f}_i(\boldsymbol{x}_t) - f_i(\boldsymbol{x}_t) \right| = 0, \qquad \lim_{t \to \infty} \left\| \hat{\nabla} f_i(\boldsymbol{x}_t) - \nabla f_i(\boldsymbol{x}_t) \right\| = 0. \tag{8}$$

*Moreover, considering the update rule $\boldsymbol{x}_t = \boldsymbol{x}_{t-1} + \gamma_t \boldsymbol{d}_t$ and setting $\rho_t = \frac{4}{(t+16)^{1/2}}$ and $\gamma_t = \frac{2}{(t+16)^{3/4}}$, $\mathbb{E}\left[\left\| \hat{\nabla} f_i(\boldsymbol{x}_t) - \nabla f_i(\boldsymbol{x}_t) \right\|^2\right]$ and $\mathbb{E}\left[\|f_i(\boldsymbol{x}_t) - f_i(\boldsymbol{x}_t)\|^2\right]$ decay at a rate of $\mathcal{O}(t^{-1/2})$.*

### 4.2 ALGORITHMIC DEVELOPMENT

To deal with the non-differentiability, the problem can be transformed into the following differentiable constrained stochastic problem by introducing the auxiliary variable $y$

$$\begin{aligned}
\min_{\boldsymbol{x} \in \mathcal{X}, y} \quad & h_0(\boldsymbol{x}, y) \triangleq y \\
\text{s.t.} \quad & h_i(\boldsymbol{x}, y) \triangleq \lambda_i \left( \mathbb{E}_{\boldsymbol{\xi}}(f_i(\boldsymbol{x}, \boldsymbol{\xi})) - z_i \right) - y \leq 0, \quad i \in [m], \\
& h_{m+j}(\boldsymbol{x}, y) \triangleq \mathbb{E}_{\boldsymbol{\xi}}(g_j(\boldsymbol{x}, \boldsymbol{\xi})) \leq 0, \quad j \in [n],
\end{aligned} \tag{9}$$

To solve it, we introduce a CSMOO algorithm for this constrained stochastic optimization problem, which contains three parts. The first part is approximating all expectation functions by the surrogate deterministic quadratic functions based on $(\boldsymbol{x}_t, y_t)$, i.e.,

$$\bar{h}_0(\boldsymbol{x}, y; \boldsymbol{x}_t, y_t) = y + \frac{\tau_0}{2} \left( \|\boldsymbol{x} - \boldsymbol{x}_t\|^2 + \|y - y_t\|^2 \right),$$

$$\bar{h}_i(\boldsymbol{x}, y; \boldsymbol{x}_t, y_t) = \lambda_i(\hat{f}_i(\boldsymbol{x}_t) + (\boldsymbol{x} - \boldsymbol{x}_t)^\top \hat{\nabla} f_i(\boldsymbol{x}_t) - z_i) - y + \frac{\tau_i}{2} \left( \|\boldsymbol{x} - \boldsymbol{x}_t\|^2 + \|y - y_t\|^2 \right),$$

$$\bar{h}_{m+j}(\boldsymbol{x}, y; \boldsymbol{x}_t, y_t) = \hat{g}_j(\boldsymbol{x}_t) + (\boldsymbol{x} - \boldsymbol{x}_t)^\top \hat{\nabla} g_j(\boldsymbol{x}_t) + \frac{\tau_{m+j}}{2} \left( \|\boldsymbol{x} - \boldsymbol{x}_t\|^2 + \|y - y_t\|^2 \right),$$

$$\tag{10}$$

where $(\tau_0, \tau_1, \cdots, \tau_{m+n}) > \boldsymbol{0}$ is a positive vector. The second part is minimizing the surrogate quadratically constrained quadratic program to obtain the auxiliary variable $(\bar{\boldsymbol{x}}_{t+1}, \bar{y}_{t+1})$:

$$\begin{aligned}
(\bar{\boldsymbol{x}}_{t+1}, \bar{y}_{t+1}) = \arg\min_{\boldsymbol{x} \in \mathcal{X}, y} \quad & \bar{h}_0(\boldsymbol{x}, y; \boldsymbol{x}_t, y_t) \\
\text{s.t.} \quad & \bar{h}_i(\boldsymbol{x}, y; \boldsymbol{x}_t, y_t) \leq 0, \quad i \in [m], \\
& \bar{h}_{m+j}(\boldsymbol{x}, y; \boldsymbol{x}_t, y_t) \leq 0, \quad j \in [n],
\end{aligned} \tag{11}$$

The above problem does not generally have a closed-form solution and can be solved efficiently with CVXPY library. Then, we can get the approximate descent direction $\boldsymbol{d}_{t+1} = (\bar{\boldsymbol{x}}_{t+1} - \boldsymbol{x}_t, \bar{y}_{t+1} - y_t)$. The third part is computing $(\boldsymbol{x}_{t+1}, y_{t+1})$ via

$$(\boldsymbol{x}_{t+1}, y_{t+1}) = (\boldsymbol{x}_t, y_t) + \gamma_{t+1} \boldsymbol{d}_{t+1}, \tag{12}$$

where $\gamma_t$ is the learning rate chosen as a nonnegative attenuating scalar. To guarantee the convergence, we make the following assumptions on the learning rate $\{\gamma_t\}$.

**Assumption 3** (Assumptions on $\{\gamma_t\}$). *The learning rates $\gamma_t \in [0, 1]$ should be chosen as follows*

*1.* $\lim_{t \to \infty} \gamma_t = 0, \sum_{t=0}^{\infty} \gamma_t = \infty, \sum_{t=0}^{\infty} \gamma_t^2 < \infty,$

2. $\sum_{t=0}^{\infty} \gamma_t^2 / \rho_t < \infty$.

However, Problem (11) may not always be feasible due to inaccuracies in the estimated function values and gradients. To overcome this challenge, we propose two strategies.

The first strategy is to solve a relaxed problem to update feasibility whenever Problem (11) has no feasible solution, i.e.,

$$
\begin{aligned}
(\bar{\boldsymbol{x}}_{t+1}, \bar{y}_{t+1}) = \underset{\boldsymbol{x} \in \mathcal{X}, y, \delta \geq 0}{\arg\min} \quad & \delta \\
\text{s.t.} \quad & \bar{h}_i(\boldsymbol{x}, y; \boldsymbol{x}_t, y_t) \leq \delta, i \in [m], \\
& \bar{h}_{m+j}(\boldsymbol{x}, y; \boldsymbol{x}_t, y_t) \leq \delta, j \in [n].
\end{aligned}
\tag{13}
$$

The convergence can be established by invoking Theorem 1 in Liu et al. (2019b), in a manner analogous to the arguments presented therein. Therefore, the detailed proof is omitted. Building on this convergence guarantee, we further demonstrate that the algorithm converges to a stationary point satisfying the Fritz John necessary conditions for the CSMOO problem, as defined in Definition 3.

---

**Algorithm 1:** CSMOO-1

---

**Input:** Preference $\boldsymbol{\lambda}$, sequences $\{\rho_t\}$ and $\{\gamma_t\}$, initial $\boldsymbol{x}_0$.
Compute $y_0 = \max_{1 \leq i \leq m}\{\lambda_i\left(\mathbb{E}_{\boldsymbol{\xi}}(f_i(\boldsymbol{x}_0, \boldsymbol{\xi})) - z_i\right)\}$.
**for** $t = 0, 1, \cdots, T-1$ **do**
    The random state $\boldsymbol{\xi}_t$ is realized.
    Update the estimated values and gradients in Equations (3), (4) and (7).
    **if** *Problem (11) is feasible* **then**
        | Solve Problem (11) to get $(\bar{\boldsymbol{x}}_{t+1}, \bar{y}_{t+1})$. // Problem update.
    **else**
        | Solve Problem (13) to get $(\bar{\boldsymbol{x}}_{t+1}, \bar{y}_{t+1})$. // Feasible update.
    **end**
    Update the estimated descent direction $\boldsymbol{d}_{t+1} = (\bar{\boldsymbol{x}}_{t+1} - \boldsymbol{x}_t, \bar{y}_{t+1} - y_t)$.
    Update $(\boldsymbol{x}_{t+1}, y_{t+1}) = (\boldsymbol{x}_t, y_t) + \gamma_{t+1}\boldsymbol{d}_{t+1}$.
**end**
**Output: The final solution $\boldsymbol{x}_T$.**

---

While the CSMOO-1 algorithm mitigates infeasibility by solving complementary approximation subproblems at each iteration, this comes at the expense of increased computational overhead and potentially slower convergence. To address it, the second strategy reformulates Problem (2) into an equivalent form that simplifies the solution process while preserving the theoretical guarantees.

$$
\begin{aligned}
\min_{\boldsymbol{x} \in \mathcal{X}, y_0, y_1 \geq 0} \quad & y_0 + \beta y_1 \\
\text{s.t.} \quad & \lambda_i\left(\mathbb{E}_{\boldsymbol{\xi}}(f_i(\boldsymbol{x}, \boldsymbol{\xi})) - z_i\right) - y_0 \leq 0, i \in [m], \\
& \mathbb{E}_{\boldsymbol{\xi}}(g_j(\boldsymbol{x}, \boldsymbol{\xi})) - y_1 \leq 0, j \in [n],
\end{aligned}
\tag{14}
$$

where $\beta > 0$ denotes the penalty parameter for constraint violations. It is worth noting that, in constructing each approximation problem, feasibility is always guaranteed regardless of the accuracy of the expectation estimates, since the auxiliary variables $y_0$ and $y_1$ can be chosen arbitrarily large. Consequently, the optimization procedure follows the solution framework of Problem (9), while preserving the essential components of (i) the problem update in Equation (11) and (ii) the variable update in Equation (12), but deliberately excluding the feasibility update in Equation (13). The resulting Problem (14) enjoys the following two desirable properties.

**Theorem 2** (Equivalence between Problem (2) and Problem (14))**.** *For any $\boldsymbol{x}$, the optimal $y_0$ and $y_1$ should be set as $y_0 = \max_{1 \leq i \leq m}\{\lambda_i\left(\mathbb{E}_{\boldsymbol{\xi}}(f_i(\boldsymbol{x}, \boldsymbol{\xi})) - z_i\right)\}$ and $y_1 = \max_{1 \leq j \leq n}\{0, \mathbb{E}_{\boldsymbol{\xi}}(g_j(\boldsymbol{x}, \boldsymbol{\xi}))\}$.*

*i) Let $\boldsymbol{x}^*$ be a locally optimal solution of Problem (2). Then, there exists $\beta^*$ such that for all $\beta \geq \beta^*$, $(\boldsymbol{x}^*, y_0^*, 0)$ is a locally optimal solution of Problem (14).*

*ii) Let $\boldsymbol{x}^\star$ be a globally optimal solution of Problem (2). Then, there exists $\beta^\star$ such that for all $\beta \geq \beta^\star$, $(\boldsymbol{x}^\star, y_0^\star, 0)$ is a globally optimal solution of Problem (14).*

**Theorem 3.** *(Weak Pareto Optimality of Problem (18)) Let $(\boldsymbol{x}^*, y_0^*, 0)$ be a locally optimal solution of Problem (14). Then, $\boldsymbol{x}^*$ is the weak Pareto solution of Problem (18).*

---

**Algorithm 2:** CSMOO-2

---

**Input:** Preference $\boldsymbol{\lambda}$, sequences $\{\rho_t\}$ and $\{\gamma_t\}$, initial $\boldsymbol{x}_0$, penalty $\beta$.
Compute $y_{0,0} = \max_{1 \le i \le m}\{\lambda_i \left(\mathbb{E}_{\boldsymbol{\xi}}(f_i(\boldsymbol{x}_0, \boldsymbol{\xi})) - z_i\right)\}, y_{1,0} = \max_{1 \le j \le n}\{0, \mathbb{E}_{\boldsymbol{\xi}}(g_j(\boldsymbol{x}_0, \boldsymbol{\xi}))\}$.
**for** $t = 0, 1, \cdots, T-1$ **do**
    The random state $\boldsymbol{\xi}_t$ is realized.
    Update the estimated values and gradients in Equations (3), (4) and (7).
    Solve the surrogate problem, similar to Problem (14), to get $(\bar{\boldsymbol{x}}_{t+1}, \bar{y}_{0,t+1}, \bar{y}_{1,t+1})$.
    Update the estimated descent direction $\boldsymbol{d}_{t+1} = (\bar{\boldsymbol{x}}_{t+1} - \boldsymbol{x}_t, \bar{y}_{0,t+1} - y_{0,t}, \bar{y}_{1,t+1} - y_{1,t})$.
    Update $(\boldsymbol{x}_{t+1}, y_{0,t+1}, y_{1,t+1}) = (\boldsymbol{x}_t, y_{0,t}, y_{1,t}) + \gamma_{t+1}\boldsymbol{d}_{t+1}$.
**end**
**Output: The final solution $\boldsymbol{x}_T$.**

---

## 5 EXPERIMENTS

### 5.1 SYNTHETIC PROBLEM

**Problem Definition.** We begin with a synthetic CSMOO problem that closely mirrors the formulation in Section 3, where the decision space is set as $\mathcal{X} = [-5, 5]$, and the objectives and constraints are, respectively, set as

$$f_1(x) = \mathbb{E}_{\epsilon \sim \mathcal{N}(0, 0.25)}((x + \epsilon)^2), \qquad f_2(x) = \mathbb{E}_{\epsilon \sim \mathcal{N}(0, 0.25)}((x + \epsilon - 1)^2),$$
$$g_1(x) = \mathbb{E}_{\epsilon \sim \mathcal{N}(0, 0.25)}((x + \epsilon)^4) \le 1, \quad g_2(x) = \mathbb{E}_{\epsilon \sim \mathcal{N}(0, 0.25)}((x + \epsilon - 1)^4) \le 2. \tag{15}$$

**Methods.** We instantiate three algorithmic variants from Sections 3-4: i) **basic**, which performs the problem update only when the problem is feasible; ii) **CSMOO-1**, which replaces an infeasible problem update with a feasible update that alleviates the constraint functions violation; and iii) **CSMOO-2**, which adopts an equivalent penalty reformulation to ensure that each problem remains feasible. Besides, we compare our methods against the analytical solution.

**Metrics.** i) *PF representation:* $\lambda_1 \in [0.01, 0.99]$ is discretized into 21 grid points. For each preference vector $\boldsymbol{\lambda} = (\lambda_1, 1 - \lambda_1)$, the corresponding optimization problem is solved, and the resulting pairs $(\mathbb{E}[f_1(x)], \mathbb{E}[f_2(x)])$ are collected. ii) *Convergence analysis:* for a given $\boldsymbol{\lambda}$, we monitor the convergence of Tchebycheff objective, the maximal constraint violation $\max_j\{0, g_j(x)\}$, and the convergence of the optimization variable.

**Results.** Across a wide range of preferences $\boldsymbol{\lambda}$, all three variants quickly drive the maximal violation to (nearly) zero and stabilize within the feasible region. When the per-iteration approximation is feasible from early iterations, **basic** and **CSMOO-1** produce nearly identical trajectories and solutions; the relaxation in CSMOO-1 is primarily beneficial when approximations are transiently infeasible. **CSMOO-2** exhibits smooth progress thanks to its always-feasible subproblems, and its final solutions match those of the other variants. Importantly, when using feasible-grid $\boldsymbol{z}$, the numerical optima align with analytic solutions (obtained by minimizing the scalarized closed-form expectations over the feasible set). Using analytic $\boldsymbol{z}$ ($z_i = \sigma^2$) produces a predictable shift of the optimizer, consistent with the fact that the two scalarizations are not equivalent. Figure 2 shows the convergence curve of $x$, Tchebycheff objective, and max constraint violation when $\lambda = 0.5$. Figure 1 shows the Pareto front of the synthetic problem using the basic method. More results of different $\boldsymbol{\lambda}$ are shown in Appendix E.

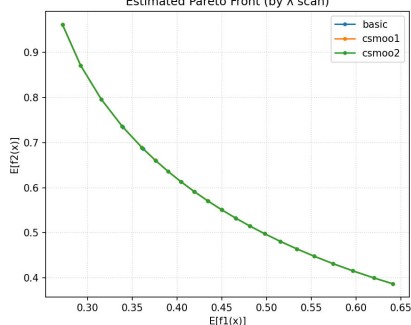

Figure 1: Pareto front of the synthetic problem.

### 5.2 PHYSICAL LAYER SECURITY PROBLEM

**Problem Definition.** Consider a downlink secure communication system that comprises one base station (BS) equipped with $n_t$ antennas, $m$ single-antenna legitimate users, and one single-antenna

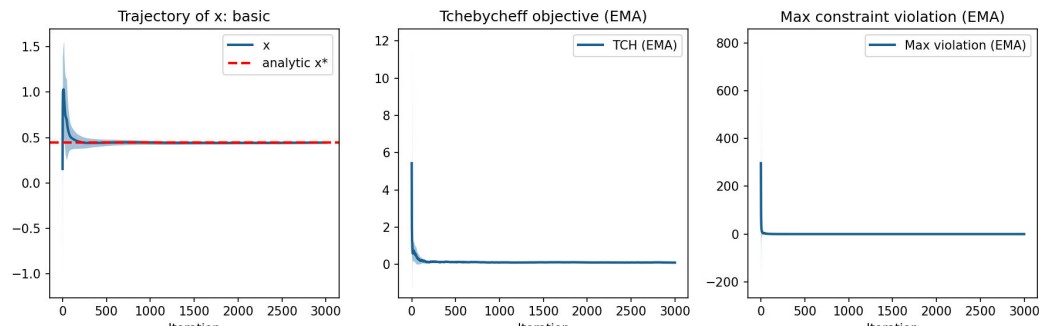

Figure 2: The convergence curve of $x$, Tchebycheff objective and Max constraint violation when $\lambda = [0.5, 0.5]$ using basic method.

eavesdropper. Let $\mathbf{h}_i \in \mathbb{C}^{n_t \times 1}$ and $\mathbf{h}_e \in \mathbb{C}^{n_t \times 1}$ be the channel state information (CSI) of the user $i$ and the eavesdropper, respectively. Define $\mathbf{W} = [\mathbf{w}_1, \cdots, \mathbf{w}_m]$. Thus, the expected rate at user $i$ and the expected rate of the eavesdropper decoding the user $i$'s signal are, respectively, given by

$$\bar{R}_i(\mathbf{W}) = \mathbb{E}_{\mathbf{e}_i} \left[ \log_2 \left( 1 + \frac{|(\hat{\mathbf{h}}_i + \mathbf{e}_i)^H \mathbf{w}_i|^2}{\sigma_i^2 + \sum_{l \neq i} |(\hat{\mathbf{h}}_i + \mathbf{e}_i)^H \mathbf{w}_l|^2} \right) \right],$$

$$\bar{C}_i(\mathbf{W}) = \mathbb{E}_{\mathbf{h}_e} \left[ \log_2 \left( 1 + \frac{1}{\sigma_e^2} |\mathbf{h}_e^H \mathbf{w}_i|^2 \right) \right], \tag{16}$$

where $\hat{\mathbf{h}}_i$ is the estimated CSI; $\mathbf{e}_i$ represents the channel estimation error; $\sigma_i^2$ is the noise at legitimate user $i$; $\mathbf{w}_i$ denotes the beamforming vector intended for user $i$. Let $C_{\text{th}}$ be the maximum tolerable information leakage. Our goal is to maximize the system rate while keeping the maximum information leakage to the eavesdroppers below a desired level.

$$\underset{\mathbf{W} \in \{\mathbf{W} : \|\mathbf{W}\|^2 \leq P_{\max}\}}{\text{maximize}} \quad \bar{\mathbf{R}}(\mathbf{W}) = (\bar{R}_1(\mathbf{W}), \cdots, \bar{R}_m(\mathbf{W}))$$

$$\text{subject to} \quad \bar{C}_i(\mathbf{W}) = \mathbb{E}_{\mathbf{h}_e} \left[ \log_2 \left( 1 + \frac{1}{\sigma_e^2} |\mathbf{h}_e^H \mathbf{w}_i|^2 \right) \right] \leq C_{\text{th}}, \qquad i = 1, \cdots, m. \tag{17}$$

**Method and Baseline.** We evaluate the CSMOO-2 method against the baseline that approximates expectations via sample averages and enforces constraints using a penalty approach. The baseline updates its variables using the classical projected SGD algorithm. Both methods use the same preference vectors, EMA schedules, proximal regularization, and mini-batch sizes. To accelerate the computation of Tchebycheff weights, the baseline adopts a smooth approximation by replacing the max operator with a log-sum-exp function.

**Metrics.** We monitor four quantities throughout training and for final reporting. i) The Tchebycheff objective. ii) The maximal constraint violation. iii) The expected rate on a large held-out Monte Carlo set. iv) The power consumption.

**Results.** Across the entire preference grid the proposed CSMOO-2 consistently achieves higher expected rate pairs than the baseline. The Tchebycheff objective decreases monotonically over iterations. The resulting Pareto curves are smooth and well-behaved when a fixed ideal point is used across all preferences. Overall, the experiments confirm that CSMOO-2 delivers superior convergence and solution quality under stringent leakage constraints, in line with our theoretical analysis. In moderate preferences regimes, CSMOO-2 consistently attains higher expected rates for the emphasized objective while preserving (or improving) the non-emphasized one, yielding strictly better final Tchebycheff values than the baseline under the same feasibility level. Across runs, the standard deviation of the final metrics is small, indicating stable convergence. We also observe that the transmit power remains at the budget boundary due to the explicit projection, which benefits rate maximization without compromising the leakage constraint. Figure 3 shows the convergence of Tchebycheff objective, constraint violation, and power consumption when $\lambda = 0.5$. Figure 4 shows the expected rate metrics comparison against CSMOO-2 and baseline when $\lambda = 0.5$. Figure 5 shows the pareto front of the PLS problem. More results of different $\boldsymbol{\lambda}$ are shown in Appendix E.

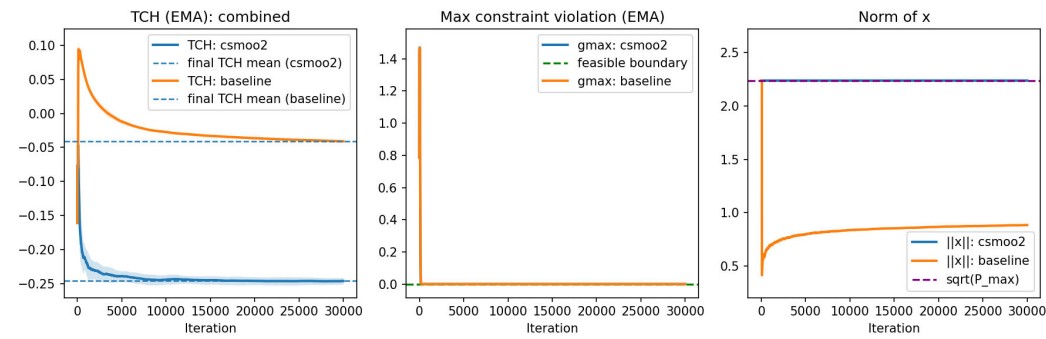

Figure 3: The convergence of Tchebycheff objective, constraint violation, and power consumption.

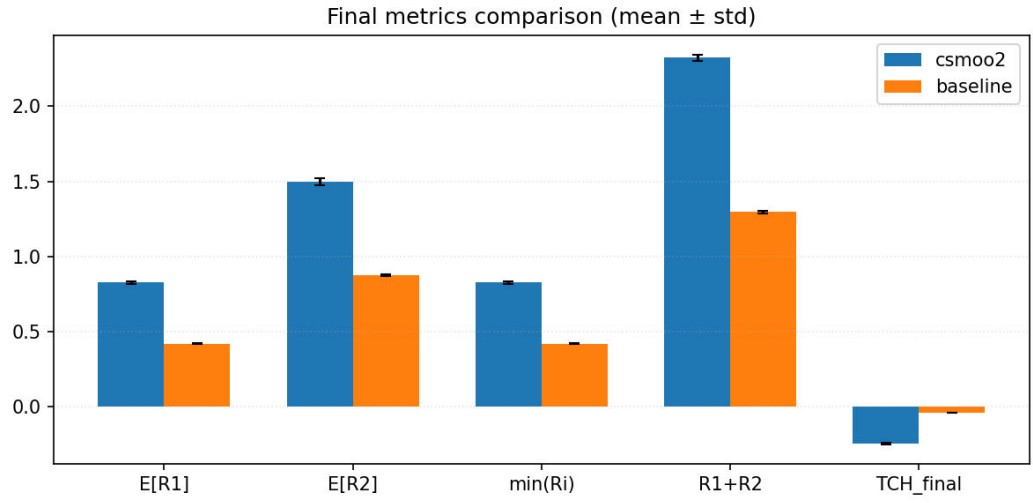

Figure 4: The expected rate metrics comparison: CSMOO-2 against baseline when $\boldsymbol{\lambda} = [0.5, 0.5]$.

# 6 CONCLUSIONS, FURTHER WORKS, AND LIMITATIONS

**Conclusions.** In this paper, we studied CSMOO problems and proposed stochastic approximation and block stochastic approximation schemes to efficiently approximate the original problem. To address potential infeasibility in the surrogate problems, we introduced feasible update reformulations and a penalty-based strategy with theoretical guarantees. Experimental results demonstrate the effectiveness of the proposed algorithms across various multi-objective optimization problems.

**Limitations and Future works.** Our study has two main limitations to address in future works. First, we assume the random variable $\boldsymbol{\xi}$ is independent of the decision variable, which may not hold in many real-world applications. Second, our experiments are limited to moderate-scale problems, and further validation on larger and more complex datasets is needed.

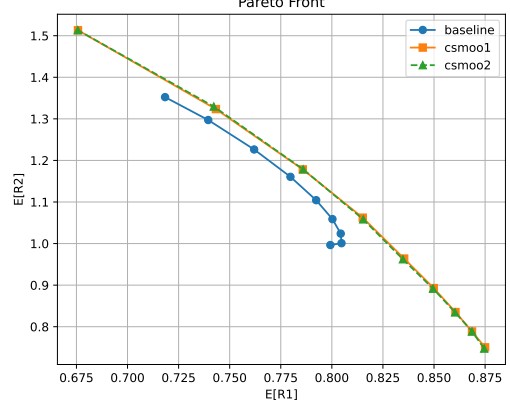

Figure 5: Pareto front of the PLS problem.

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

# Constrained Stochastic Multi-Objective Optimization (Appendix)

This appendix contains the following supplementary materials:

- Detailed preliminary definitions and assumptions are provided in Section A.
- Important lemmas forming the foundation of subsequent proofs are provided in Section B.
- Detailed proofs for the theoretical analysis are proposed in Section C.
- Extensions of CSMOO algorithms to special problem settings are discussed in Section D.
- More experimental results and analyses are provided in Section E

## A    DETAILED PRELIMINARIES

### A.1    CONSTRAINED STOCHASTIC MULTI-OBJECTIVE OPTIMIZATION (CSMOO)

Consider a general CSMOO problem with $m$ objectives and $n$ constraints

$$\min_{\boldsymbol{x} \in \mathcal{X}} \quad \boldsymbol{f}(\boldsymbol{x}) = (f_1(\boldsymbol{x}), \cdots, f_m(\boldsymbol{x})) \triangleq (\mathbb{E}_{\boldsymbol{\xi}}(f_1(\boldsymbol{x}, \boldsymbol{\xi})), \cdots, \mathbb{E}_{\boldsymbol{\xi}}(f_m(\boldsymbol{x}, \boldsymbol{\xi})))$$

$$\text{s.t.} \quad g_j(\boldsymbol{x}) \triangleq \mathbb{E}_{\boldsymbol{\xi}}(g_j(\boldsymbol{x}, \boldsymbol{\xi})) \leq 0, \qquad j \in [n], \tag{18}$$

where $\boldsymbol{x} \in \mathcal{X}$ is the optimization variable with $\mathcal{X} \subseteq \mathbb{R}^d$ being the decision space of the problem; $\boldsymbol{\xi}$ is a random state defined on the probability space $(\Omega, \mathcal{F}, \mathbb{P})$, with $\Omega$ being the sample space, $\mathcal{F}$ being the $\sigma$-algebra generated by subsets of $\Omega$, and $\mathbb{P}$ being a probability measure defined on $\mathcal{F}$. With a slight abuse of notation, we use $f_i(\boldsymbol{x})$ to denote a deterministic function and $f_i(\boldsymbol{x}, \boldsymbol{\xi})$ to denote its stochastic counterpart. We make the following assumptions about the problem structure.

**Assumption 4** (Assumptions on Problem (18)). *The optimization problem must possess the following structural properties, specified with respect to its decision space, function values, and gradients.*

1. *$\mathcal{X}$ is compact and convex with a positive constant $D$, i.e., $\|\boldsymbol{x}_1 - \boldsymbol{x}_2\| \leq D, \forall \boldsymbol{x}_1, \boldsymbol{x}_2 \in \mathcal{X}$.*

2. *The functions $f_i(\boldsymbol{x}, \boldsymbol{\xi}), \forall i$ and $g_j(\boldsymbol{x}, \boldsymbol{\xi}), \forall j$ are continuously differentiable on $\mathcal{X}$. For any given $\boldsymbol{\xi}$, the functions $f_i(\boldsymbol{x}, \boldsymbol{\xi})$ and $g_j(\boldsymbol{x}, \boldsymbol{\xi})$ are Lipschitz continuous with constants $L_{f_i(\boldsymbol{\xi})}$ and $L_{g_j(\boldsymbol{\xi})}$, respectively. Furthermore, the functions $f_i(\boldsymbol{x})$ and $g_j(\boldsymbol{x})$ are Lipschitz continuous with constants $L_{f_i}$ and $L_{g_j}$, respectively.*

3. *For any given $\boldsymbol{\xi}$, the gradients $\nabla f_i(\boldsymbol{x}, \boldsymbol{\xi})$ and $\nabla g_j(\boldsymbol{x}, \boldsymbol{\xi})$ are Lipschitz continuous with constants $L_{\nabla f_i(\boldsymbol{\xi})}$ and $L_{\nabla g_j(\boldsymbol{\xi})}$, respectively. Furthermore, the gradients $\nabla f_i(\boldsymbol{x})$ and $\nabla g_j(\boldsymbol{x})$ are Lipschitz continuous with constants $L_{\nabla f_i}$ and $L_{\nabla g_j}$, respectively.*

**Definition 1** (Dominance and Strict Dominance). *Let $\boldsymbol{x}_1, \boldsymbol{x}_2$ be two solutions for Problem (18), $\boldsymbol{x}_1$ is said to dominate $\boldsymbol{x}_2$, denoted as $\boldsymbol{f}(\boldsymbol{x}_1) \prec \boldsymbol{f}(\boldsymbol{x}_2)$, if and only if $f_i(\boldsymbol{x}_1) \leq f_i(\boldsymbol{x}_2), \forall i \in [m]$ and $f_{i'}(\boldsymbol{x}_1) < f_{i'}(\boldsymbol{x}_2), \exists i' \in [m]$. In addition, $\boldsymbol{x}_1$ is said to strictly dominate $\boldsymbol{x}_2$, i.e., $\boldsymbol{f}(\boldsymbol{x}_1) \prec_{\text{strict}} \boldsymbol{f}(\boldsymbol{x}_2)$, if and only if $f_i(\boldsymbol{x}_1) < f_i(\boldsymbol{x}_2), \forall i \in [m]$.*

**Definition 2** (Weakly Pareto Optimality and Pareto Optimality). *A solution $\boldsymbol{x}^{\star}$ is Pareto optimal if there is no $\boldsymbol{x}$ such that $\boldsymbol{f}(\boldsymbol{x}) \prec \boldsymbol{f}(\boldsymbol{x}^{\star})$. A solution $\boldsymbol{x}^{*}$ is weakly Pareto optimal if there is no $\boldsymbol{x}$ such that $\boldsymbol{f}(\boldsymbol{x}) \prec_{\text{strict}} \boldsymbol{f}(\boldsymbol{x}^{\star})$.*

**Definition 3** (The Fritz John Condition (Ehrgott, 2005)). *The Fritz John necessary condition for $\boldsymbol{x}^{*}$ to be weak Pareto optimal is that there exist two nonnegative nonzero vectors $\boldsymbol{\nu}, \boldsymbol{\mu}$ such that*

$$\boldsymbol{0} \in \sum_{i=1}^{m} \nu_i \nabla f_i(\boldsymbol{x}^*) + \sum_{j=1}^{n} \mu_j \nabla g_j(\boldsymbol{x}^*) + N_{\mathcal{X}}(\boldsymbol{x}^*),$$

$$\mu_j g_j(\boldsymbol{x}^*) = 0, \qquad \forall j \in [n], \tag{19}$$

where $N_{\mathcal{X}}(\boldsymbol{x}^*) \triangleq \{\boldsymbol{d} \in \mathbb{R}^d : \boldsymbol{d}^\top(\boldsymbol{y} - \boldsymbol{x}^*) \leq 0, \forall \boldsymbol{y} \in \mathcal{X}\}$ is the normal cone to $\mathcal{X}$ at $\boldsymbol{x}^*$.

### A.2 TCHEBYCHEFF SCALARIZATION

The widely-used Tchebycheff (TCH) scalarization method, supported by rigorous theoretical foundations (Bowman, 1976; Steuer and Eng-Ung, 1983), provides an effective framework for converting multi-objective optimization problems into single-objective ones. The mathematical formulation of this scalarized optimization problem can be expressed as

$$\min_{\boldsymbol{x} \in \mathcal{X}} \max_{1 \leq i \leq m} \quad \{\lambda_i\left(\mathbb{E}_{\boldsymbol{\xi}}(f_i(\boldsymbol{x}, \boldsymbol{\xi})) - z_i\right)\}$$
$$\text{s.t.} \quad \mathbb{E}_{\boldsymbol{\xi}}(g_j(\boldsymbol{x}, \boldsymbol{\xi})) \leq 0, \quad j \in [n], \tag{20}$$

where $\boldsymbol{\lambda} \triangleq (\lambda_1, \cdots, \lambda_m) \in \Delta_m$ is a preference vector over the $m$ objectives. $\boldsymbol{z} \triangleq (z_1, \cdots, z_m) \in \mathbb{R}^m$ is the ideal objective values, where $z_i = \inf_{\boldsymbol{x} \in \mathcal{X}} \mathbb{E}_{\boldsymbol{\xi}}\left(f_i(\boldsymbol{x}, \boldsymbol{\xi})\right)$. The Tchebycheff scalarization guarantees Pareto optimal solutions and facilitates the exploration of both convex and nonconvex regions of the Pareto front (Choo and Atkins, 1983).

## B  IMPORTANT LEMMAS

**Lemma 1.** *(Supermartingale Convergence Theorem, Proposition 4.2 (Bertsekas and Tsitsiklis, 1996)) Let $Y_t$, $X_t$, and $Z_t$, $t = 0, 1, 2, \cdots$, be three sequences of random variables and let $\mathcal{F}_t$, $t = 0, 1, 2, \cdots$, be sets of random variables such that $\mathcal{F}_t \subset \mathcal{F}_{t+1}$ for all $t$. Suppose that:*

    *(a) The random variables $Y_t$, $X_t$, and $Z_t$ are nonnegative, and are functions of the random variables in $\mathcal{F}_t$.*

    *(b) For each $t$, we have $\mathbb{E}[Y_{t+1}|\mathcal{F}_t] \leq Y_t - X_t + Z_t$.*

    *(c) There holds $\sum_{t=0}^{\infty} Z_t < \infty$.*

*Then, we have $\sum_{t=0}^{\infty} X_t < \infty$ and the sequence $Y_t$ converges to a nonnegative random variable $Y$, with probability 1.*

**Lemma 2.** *(Proposition 4.3.1 (Bertsekas, 1999)) Let $\boldsymbol{x}^*$ be a local minimum of the problem*

$$\min_{\boldsymbol{x}} \quad f(\boldsymbol{x})$$
$$\text{subject to} \quad h_i(\boldsymbol{x}) = 0, i = 1, \cdots, m, \tag{21}$$
$$g_j(\boldsymbol{x}) \leq 0, j = 1, \cdots, n,$$

*which is regular and satisfies together with the corresponding Lagrange multiplier vector $\boldsymbol{\lambda}^*$ and $\boldsymbol{\mu}^*$, the second order sufficiency conditions of Lemma 3. Then if*

$$c > \sum_{i=1}^{m} |\lambda_i^*| + \sum_{j=1}^{n} \mu_j^*, \tag{22}$$

*the vector $\boldsymbol{x}^*$ is a strict unconstrained local minimum of $f + cP$, where*

$$P(\boldsymbol{x}) = \max\{0, g_1(\boldsymbol{x}), \cdots, g_n(\boldsymbol{x}), |h_1(\boldsymbol{x})|, \cdots, |h_m(\boldsymbol{x})|\}. \tag{23}$$

**Lemma 3.** *(Second Order Sufficiency Conditions, Proposition 3.3.2 (Bertsekas, 1999)) Let $A(\boldsymbol{x}^*)$ denote the set of active constraints at $\boldsymbol{x}^*$ and $L(\boldsymbol{x}, \boldsymbol{\lambda}, \boldsymbol{\mu})$ is the Lagrangian function. Assume that $f, h,$ and $g$ are twice continuously differentiable, and let $\boldsymbol{x}^*$, $\boldsymbol{\lambda}^*$ and $\boldsymbol{\mu}^*$ satisfy*

$$\nabla_{\boldsymbol{x}} L(\boldsymbol{x}^*, \boldsymbol{\lambda}^*, \boldsymbol{\mu}^*) = 0, \quad , h_i(\boldsymbol{x}^*) = 0, \quad g_j(\boldsymbol{x}^*) \leq 0,$$
$$\mu_j^* \geq 0, \quad j = 1, \cdots, r,$$
$$\mu_j^* = 0, \quad \forall j \notin A(\boldsymbol{x}^*), \tag{24}$$
$$\boldsymbol{y}^T \nabla_{\boldsymbol{xx}}^2 L(\boldsymbol{x}^*, \boldsymbol{\lambda}^*, \boldsymbol{\mu}^*)\boldsymbol{y} > 0,$$

*for all $\boldsymbol{y} \neq \boldsymbol{0}$ such that*

$$\nabla h_i(\boldsymbol{x}^*)^T \boldsymbol{y} = 0, \forall i = 1, \cdots, m, \quad \nabla g_i(\boldsymbol{x}^*)^T \boldsymbol{y} = 0, \forall j \notin A(\boldsymbol{x}^*), \tag{25}$$

**Lemma 4.** *(Lemma 19 (Mokhtari et al., 2020)) Consider the scalars $b \geq 0$ and $c > 1$. Let $\phi_t$ be a sequence of real numbers satisfying*

$$\phi_t \leq \left(1 - \frac{c}{(t+t_0)^\alpha}\right)\phi_{t-1} + \frac{b}{(t+t_0)^{2\alpha}}, \tag{26}$$

*for some $0 \leq \alpha \leq 1$ and $t_0 \geq 0$. Then, the sequence $\phi_t$ converges to zeros at the following rate*

$$\phi_t \leq \frac{Q}{(t+t_0+1)^\alpha}, \tag{27}$$

*where $Q = \max\{\phi_0(t_0+1)^\alpha, b/(c-1)\}$.*

## C   DETAILED PROOFS

### C.1   PROOF OF THEOREM 1

Since the functions $f_i(\boldsymbol{x}_t)$ and $g_j(\boldsymbol{x}_t)$ share the same structure, we present the proof only for $f_i(\boldsymbol{x}_t)$. The proof for $g_j(\boldsymbol{x}_t)$ can be derived similarly.

**1. Asymptotic Consistency**

**1.1. Value Consistency:** The conditional expectation of the difference between the sequences of approximate function value $\hat{f}_i(\boldsymbol{x}_t)$ and true function value $f_i(\boldsymbol{x}_t)$ is given by

$$\mathbb{E}\left[\left|\hat{f}_i(\boldsymbol{x}_t) - f_i(\boldsymbol{x}_t)\right|^2 \bigg| \mathcal{F}_t\right] = \mathbb{E}\left[\left|\rho_t f_i(\boldsymbol{x}_t, \boldsymbol{\xi}_t) + (1-\rho_t)\hat{f}_i(\boldsymbol{x}_{t-1}) - f_i(\boldsymbol{x}_t)\right|^2 \bigg| \mathcal{F}_t\right] \tag{28}$$

The expectation here is only with respect to the choice of random set $\mathcal{F}_t$. By adding and subtracting the gradient of the previous step $f_i(\boldsymbol{x}_{t-1})$ to the expression and expanding the terms, we can obtain

$$\mathbb{E}\left[\left|\rho_t f_i(\boldsymbol{x}_t, \boldsymbol{\xi}_t) + (1-\rho_t)\hat{f}_i(\boldsymbol{x}_{t-1}) - f_i(\boldsymbol{x}_t)\right|^2 \bigg| \mathcal{F}_t\right]$$

$$= \rho_t^2 \mathbb{E}\left[\left|f_i(\boldsymbol{x}_t, \boldsymbol{\xi}_t) - f_i(\boldsymbol{x}_t)\right|^2 \bigg| \mathcal{F}_t\right] + (1-\rho_t)^2 \mathbb{E}\left[\left|\hat{f}_i(\boldsymbol{x}_{t-1}) - f_i(\boldsymbol{x}_{t-1})\right|^2 \bigg| \mathcal{F}_t\right]$$

$$+ (1-\rho_t)^2 \mathbb{E}\left[\left|f_i(\boldsymbol{x}_{t-1}) - f_i(\boldsymbol{x}_t)\right|^2 \bigg| \mathcal{F}_t\right]$$

$$+ 2\rho_t(1-\rho_t)\mathbb{E}\left[(f_i(\boldsymbol{x}_t, \boldsymbol{\xi}_t) - f_i(\boldsymbol{x}_t))(\hat{f}_i(\boldsymbol{x}_{t-1}) - f_i(\boldsymbol{x}_{t-1})) \bigg| \mathcal{F}_t\right]$$

$$+ 2\rho_t(1-\rho_t)\mathbb{E}\left[(f_i(\boldsymbol{x}_t, \boldsymbol{\xi}_t) - f_i(\boldsymbol{x}_t))(f_i(\boldsymbol{x}_{t-1}) - f_i(\boldsymbol{x}_t)) \bigg| \mathcal{F}_t\right] \tag{29}$$

$$+ 2(1-\rho_t)^2 \mathbb{E}\left[(\hat{f}_i(\boldsymbol{x}_{t-1}) - f_i(\boldsymbol{x}_{t-1}))(f_i(\boldsymbol{x}_{t-1}) - f_i(\boldsymbol{x}_t)) \bigg| \mathcal{F}_t\right]$$

$$= \rho_t^2 \mathbb{E}\left[\left|f_i(\boldsymbol{x}_t, \boldsymbol{\xi}_t) - f_i(\boldsymbol{x}_t)\right|^2 \bigg| \mathcal{F}_t\right]$$

$$+ (1-\rho_t)^2 \left|\hat{f}_i(\boldsymbol{x}_{t-1}) - f_i(\boldsymbol{x}_{t-1})\right|^2 + (1-\rho_t)^2 \left|f_i(\boldsymbol{x}_{t-1}) - f_i(\boldsymbol{x}_t)\right|^2$$

$$+ 2(1-\rho_t)^2 (\hat{f}_i(\boldsymbol{x}_{t-1}) - f_i(\boldsymbol{x}_{t-1}))(f_i(\boldsymbol{x}_{t-1}) - f_i(\boldsymbol{x}_t)),$$

where the last equality is due to the definition $f_i(\boldsymbol{x}) = \mathbb{E}_{\boldsymbol{\xi}}(f_i(\boldsymbol{x}, \boldsymbol{\xi}))$. Furthermore, with the inequality $2ab \leq \left(\sqrt{k}a\right)^2 + \left(\frac{b}{\sqrt{k}}\right)^2 = ka^2 + 1/kb^2$ for any $k > 0$, the difference can be upper-bounded by

$$
\mathbb{E}\left[\left|\hat{f}_i(\boldsymbol{x}_t) - f_i(\boldsymbol{x}_t)\right|^2 \Big| \mathcal{F}_t\right]
$$

$$
\leq \rho_t^2 \mathbb{E}\left[|f_i(\boldsymbol{x}_t, \boldsymbol{\xi}_t) - f_i(\boldsymbol{x}_t)|^2 \Big| \mathcal{F}_t\right]
$$

$$
+ (1 + \frac{\rho_t}{2})(1 - \rho_t)^2 \left|\hat{f}_i(\boldsymbol{x}_{t-1}) - f_i(\boldsymbol{x}_{t-1})\right|^2 + (1 + \frac{2}{\rho_t})(1 - \rho_t)^2 |f_i(\boldsymbol{x}_{t-1}) - f_i(\boldsymbol{x}_t)|^2
$$

$$
\leq (1 + \frac{\rho_t}{2})(1 - \rho_t)^2 \left|\hat{f}_i(\boldsymbol{x}_{t-1}) - f_i(\boldsymbol{x}_{t-1})\right|^2 + (1 + \frac{2}{\rho_t})(1 - \rho_t)^2 L_{f_i}^2 \|\boldsymbol{x}_{t-1} - \boldsymbol{x}_t\|^2 + \rho_t^2 \varepsilon^2
$$

$$
\leq (1 - \frac{\rho_t}{2}) \left|\hat{f}_i(\boldsymbol{x}_{t-1}) - f_i(\boldsymbol{x}_{t-1})\right|^2 + (1 + \frac{2}{\rho_t}) L_{f_i}^2 \|\boldsymbol{x}_{t-1} - \boldsymbol{x}_t\|^2 + \rho_t^2 \varepsilon^2,
$$

$$
\tag{30}
$$

where the last inequality is due to $(1 + \frac{\rho_t}{2})(1 - \rho_t)^2 \leq (1 + \frac{\rho_t}{2})(1 - \rho_t) = 1 - \frac{\rho_t}{2} - \frac{\rho_t^2}{2} \leq 1 - \frac{\rho_t}{2}$ and $(1 - \rho_t)^2 \leq 1$.

To construct a supermartingale, define the stochastic process $X_t$, $Y_t$, and $Z_t$ as

$$
X_t \triangleq \frac{\rho_t}{2} \left|\hat{f}_i(\boldsymbol{x}_{t-1}) - f_i(\boldsymbol{x}_{t-1})\right|^2 \geq 0,
$$

$$
Y_t \triangleq \left|\hat{f}_i(\boldsymbol{x}_{t-1}) - f_i(\boldsymbol{x}_{t-1})\right|^2 \geq 0, \tag{31}
$$

$$
Z_t \triangleq (1 + \frac{2}{\rho_t}) L_{f_i}^2 \|\boldsymbol{x}_{t-1} - \boldsymbol{x}_t\|^2 + \rho_t^2 \varepsilon^2 \geq 0.
$$

From Lemma 1, we have $\sum_{t=0}^{\infty} \rho_t \left|\hat{f}_i(\boldsymbol{x}_t) - f_i(\boldsymbol{x}_t)\right|^2 < \infty$ and $\lim_{t\to\infty} \left|\hat{f}_i(\boldsymbol{x}_t) - f_i(\boldsymbol{x}_t)\right|^2$ exists. Hence, it follows from $\sum_{t=0}^{\infty} \rho_t = \infty$ that

$$
\lim_{t\to\infty} \left|\hat{f}_i(\boldsymbol{x}_t) - f_i(\boldsymbol{x}_t)\right| = 0 \tag{32}
$$

**1.2. Gradient Consistency:** The conditional expectation of the difference between the sequences of approximate gradients $\hat{\nabla} f_i(\boldsymbol{x}_t)$ and true gradients $\nabla f_i(\boldsymbol{x}_t)$ is given by

$$
\mathbb{E}\left[\left\|\hat{\nabla} f_i(\boldsymbol{x}_t) - \nabla f_i(\boldsymbol{x}_t)\right\|^2 \Big| \mathcal{F}_t\right] = \mathbb{E}_{\boldsymbol{\xi}_t}\left[\left\|\rho_t \nabla f_i(\boldsymbol{x}_t, \boldsymbol{\xi}_t) + (1 - \rho_t)\hat{\nabla} f_i(\boldsymbol{x}_{t-1}) - \nabla f_i(\boldsymbol{x}_t)\right\|^2 \Big| \mathcal{F}_t\right]
$$

$$
= \rho_t^2 \mathbb{E}\left[\|\nabla f_i(\boldsymbol{x}_t, \boldsymbol{\xi}_t) - \nabla f_i(\boldsymbol{x}_t)\|^2 \Big| \mathcal{F}_t\right] + (1 - \rho_t)^2 \mathbb{E}_{\boldsymbol{\xi}_t}\left[\left\|\hat{\nabla} f_i(\boldsymbol{x}_{t-1}) - \nabla f_i(\boldsymbol{x}_{t-1})\right\|^2 \Big| \mathcal{F}_t\right]
$$

$$
+ (1 - \rho_t)^2 \mathbb{E}\left[\|\nabla f_i(\boldsymbol{x}_{t-1}) - \nabla f_i(\boldsymbol{x}_t)\|^2 \Big| \mathcal{F}_t\right]
$$

$$
+ 2\rho_t(1 - \rho_t)\mathbb{E}\left[(\nabla f_i(\boldsymbol{x}_t, \boldsymbol{\xi}_t) - \nabla f_i(\boldsymbol{x}_t))^\top (\hat{\nabla} f_i(\boldsymbol{x}_{t-1}) - \nabla f_i(\boldsymbol{x}_{t-1})) \Big| \mathcal{F}_t\right]
$$

$$
+ 2\rho_t(1 - \rho_t)\mathbb{E}\left[(\nabla f_i(\boldsymbol{x}_t, \boldsymbol{\xi}_t) - \nabla f_i(\boldsymbol{x}_t))^\top (\nabla f_i(\boldsymbol{x}_{t-1}) - \nabla f_i(\boldsymbol{x}_t)) \Big| \mathcal{F}_t\right]
$$

$$
+ 2(1 - \rho_t)^2 \mathbb{E}\left[(\hat{\nabla} f_i(\boldsymbol{x}_{t-1}) - \nabla f_i(\boldsymbol{x}_{t-1}))^\top (\nabla f_i(\boldsymbol{x}_{t-1}) - \nabla f_i(\boldsymbol{x}_t)) \Big| \mathcal{F}_t\right]
$$

$$
= \rho_t^2 \mathbb{E}\left[\|\nabla f_i(\boldsymbol{x}_t, \boldsymbol{\xi}_t) - \nabla f_i(\boldsymbol{x}_t)\|^2 \Big| \mathcal{F}_t\right]
$$

$$
+ (1 - \rho_t)^2 \left\|\hat{\nabla} f_i(\boldsymbol{x}_{t-1}) - \nabla f_i(\boldsymbol{x}_{t-1})\right\|^2 + (1 - \rho_t)^2 \|\nabla f_i(\boldsymbol{x}_{t-1}) - \nabla f_i(\boldsymbol{x}_t)\|^2
$$

$$
+ 2(1 - \rho_t)^2 (\hat{\nabla} f_i(\boldsymbol{x}_{t-1}) - \nabla f_i(\boldsymbol{x}_{t-1}))^\top (\nabla f_i(\boldsymbol{x}_{t-1}) - \nabla f_i(\boldsymbol{x}_t)),
$$

$$
\tag{33}
$$

where the last equality is due to $\nabla f_i(\boldsymbol{x}) = \nabla \mathbb{E}_{\boldsymbol{\xi}}(f_i(\boldsymbol{x}, \boldsymbol{\xi})) = \mathbb{E}_{\boldsymbol{\xi}}(\nabla f_i(\boldsymbol{x}, \boldsymbol{\xi}))$. Furthermore, the difference can be upper-bounded by

$$
\mathbb{E}\left[\left\|\hat{\nabla} f_i(\boldsymbol{x}_t) - \nabla f_i(\boldsymbol{x}_t)\right\|^2 \bigg| \mathcal{F}_t\right]
$$

$$
\leq \rho_t^2 \mathbb{E}\left[\left\|\nabla f_i(\boldsymbol{x}_t, \boldsymbol{\xi}_t) - \nabla f_i(\boldsymbol{x}_t)\right\|^2 \bigg| \mathcal{F}_t\right] + (1 + \frac{\rho_t}{2})(1 - \rho_t)^2 \left\|\hat{\nabla} f_i(\boldsymbol{x}_{t-1}) - \nabla f_i(\boldsymbol{x}_{t-1})\right\|^2
$$

$$
+ (1 + \frac{2}{\rho_t})(1 - \rho_t)^2 \left\|\nabla f_i(\boldsymbol{x}_{t-1}) - \nabla f_i(\boldsymbol{x}_t)\right\|^2
$$

$$
\leq (1 + \frac{\rho_t}{2})(1 - \rho_t)^2 \left\|\hat{\nabla} f_i(\boldsymbol{x}_{t-1}) - \nabla f_i(\boldsymbol{x}_{t-1})\right\|^2
$$

$$
+ (1 + \frac{2}{\rho_t})(1 - \rho_t)^2 L_{\nabla f_i}^2 \left\|\boldsymbol{x}_{t-1} - \boldsymbol{x}_t\right\|^2 + \rho_t^2 \varepsilon^2
$$

$$
\leq (1 - \frac{\rho_t}{2}) \left\|\hat{\nabla} f_i(\boldsymbol{x}_{t-1}) - \nabla f_i(\boldsymbol{x}_{t-1})\right\|^2 + (1 + \frac{2}{\rho_t}) L_{\nabla f_i}^2 \left\|\boldsymbol{x}_{t-1} - \boldsymbol{x}_t\right\|^2 + \rho_t^2 \varepsilon^2
$$

$$
\tag{34}
$$

To construct a supermartingale, define the stochastic process $X_t$, $Y_t$, and $Z_t$ as

$$
X_t \triangleq \frac{\rho_t}{2} \left\|\hat{\nabla} f_i(\boldsymbol{x}_{t-1}) - \nabla f_i(\boldsymbol{x}_{t-1})\right\|^2 \geq 0,
$$

$$
Y_t \triangleq \left\|\hat{\nabla} f_i(\boldsymbol{x}_{t-1}) - \nabla f_i(\boldsymbol{x}_{t-1})\right\|^2 \geq 0, \tag{35}
$$

$$
Z_t \triangleq (1 + \frac{2}{\rho_t}) L_{\nabla f_i}^2 \left\|\boldsymbol{x}_{t-1} - \boldsymbol{x}_t\right\|^2 + \rho_t^2 \varepsilon^2 \geq 0.
$$

From Lemma 1, we have $\sum_{t=0}^{\infty} \rho_t \|\hat{\nabla} f_i(\boldsymbol{x}_t) - \nabla f_i(\boldsymbol{x}_t)\|^2 < \infty$ and $\lim_{t\to\infty} \|\hat{\nabla} f_i(\boldsymbol{x}_t) - \nabla f_i(\boldsymbol{x}_t)\|^2$ exists. Hence, it follows from $\sum_{t=0}^{\infty} \rho_t = \infty$ that

$$
\lim_{t\to\infty} \left\|\hat{\nabla} f_i(\boldsymbol{x}_t) - \nabla f_i(\boldsymbol{x}_t)\right\| = 0. \tag{36}
$$

**1.3. Gradient Consistency-Cyclic:** We begin by examining the $r$-th block, for which the gradient is updated only at iteration $t = \ell k + r - 1$, with $\ell \in \mathbb{N}$. Note that the $r$-th block and its gradient estimation remain fixed during the preceding $k - 1$ iterations, and thus we have

$$
\boldsymbol{x}_{r,\ell k + r - 2} = \boldsymbol{x}_{r,\ell k + r - 3} = \cdots = \boldsymbol{x}_{r,(\ell-1)k + r - 1},
$$
$$
\hat{\nabla}_r f_i(\boldsymbol{x}_{\ell k + r - 2}) = \hat{\nabla}_r f_i(\boldsymbol{x}_{\ell k + r - 3}) = \cdots = \hat{\nabla}_r f_i(\boldsymbol{x}_{(\ell-1)k + r - 1}). \tag{37}
$$

It follows from the same line of reasoning in the Gradient Consistency (with the following notation mapping: $t \to \ell k + r - 1, t - 1 \to (\ell-1)k + r - 1, \hat{\nabla} f_i(\boldsymbol{x}_t) \to \hat{\nabla}_r f_i(\boldsymbol{x}_{\ell k + r - 1}), \hat{\nabla} f_i(\boldsymbol{x}_{t-1}) \to \hat{\nabla}_r f_i(\boldsymbol{x}_{(\ell-1)k + r - 1}), \nabla f_i(\boldsymbol{x}_t) \to \nabla_r f_i(\boldsymbol{x}_{\ell k + r - 1}), \nabla f_i(\boldsymbol{x}_{t-1}) \to \nabla_r f_i(\boldsymbol{x}_{(\ell-1)k + r - 1}))$ that

$$
\mathbb{E}\left[\left\|\hat{\nabla}_r f_i(\boldsymbol{x}_{\ell k + r - 1}) - \nabla_r f_i(\boldsymbol{x}_{\ell k + r - 1})\right\|^2 \bigg| \mathcal{F}_t\right]
$$

$$
\leq (1 + \frac{\rho_{\ell k + r - 1}}{2})(1 - \rho_{\ell k + r - 1})^2 \left\|\hat{\nabla}_r f_i(\boldsymbol{x}_{(\ell-1)k + r - 1} - \nabla_r f_i(\boldsymbol{x}_{(\ell-1)k + r - 1})\right\|^2
$$

$$
+ (1 + \frac{2}{\rho_{\ell k + r - 1}})(1 - \rho_{\ell k + r - 1})^2 L_{\nabla f_i}^2 \left\|\boldsymbol{x}_{\ell k + r - 1} - \boldsymbol{x}_{(\ell-1)k + r - 1}\right\|^2 + \rho_{\ell k + r - 1}^2 \varepsilon^2
$$

$$
= (1 + \frac{\rho_{\ell k + r - 1}}{2})(1 - \rho_{\ell k + r - 1})^2 \left\|\hat{\nabla}_r f_i(\boldsymbol{x}_{(\ell-1)k + r - 1} - \nabla_r f_i(\boldsymbol{x}_{(\ell-1)k + r - 1})\right\|^2 \tag{38}
$$

$$
+ (1 + \frac{2}{\rho_{\ell k + r - 1}})(1 - \rho_{\ell k + r - 1})^2 L_{\nabla f_i}^2 \left\|\boldsymbol{x}_{\ell k + r - 1} - \boldsymbol{x}_{\ell k + r - 2}\right\|^2 + \rho_{\ell k + r - 1}^2 \varepsilon^2
$$

$$
\leq (1 - \frac{\rho_{\ell k + r - 1}}{2}) \left\|\hat{\nabla}_r f_i(\boldsymbol{x}_{(\ell-1)k + r - 1} - \nabla_r f_i(\boldsymbol{x}_{(\ell-1)k + r - 1})\right\|^2
$$

$$
+ (1 + \frac{2}{\rho_{\ell k + r - 1}}) L_{\nabla f_i}^2 \left\|\boldsymbol{x}_{\ell k + r - 1} - \boldsymbol{x}_{\ell k + r - 2}\right\|^2 + \rho_{\ell k + r - 1}^2 \varepsilon^2
$$

Therefore, the gradient consistency holds at iteration $\ell k + r - 1$:

$$\lim_{\ell \to \infty} \left\| \hat{\nabla}_r f_i(\boldsymbol{x}_{\ell k + r - 1}) - \nabla_r f_i(\boldsymbol{x}_{\ell k + r - 1}) \right\| = 0. \tag{39}$$

Furthermore, the gradient consistency holds over iteration $\ell k + r + j - 1, j = 1, 2, \cdots, K - 1$:

$$\lim_{\ell \to \infty} \left\| \hat{\nabla}_r f_i(\boldsymbol{x}_{\ell k + r + j - 1}) - \nabla_r f_i(\boldsymbol{x}_{\ell k + r + j - 1}) \right\|^2$$

$$= \lim_{\ell \to \infty} \left\| \hat{\nabla}_r f_i(\boldsymbol{x}_{\ell k + r - 1}) - \nabla_r f_i(\boldsymbol{x}_{\ell k + r - 1}) + \nabla_r f_i(\boldsymbol{x}_{\ell k + r - 1}) - \nabla_r f_i(\boldsymbol{x}_{\ell k + r + j - 1}) \right\|^2$$

$$\leq \lim_{\ell \to \infty} 2 \left( \left\| \hat{\nabla}_r f_i(\boldsymbol{x}_{\ell k + r - 1}) - \nabla_r f_i(\boldsymbol{x}_{\ell k + r - 1}) \right\|^2 + \left\| \nabla_r f_i(\boldsymbol{x}_{\ell k + r - 1}) - \nabla_r f_i(\boldsymbol{x}_{\ell k + r + j - 1}) \right\|^2 \right)$$

$$= \lim_{\ell \to \infty} 2 \left\| \nabla_r f_i(\boldsymbol{x}_{\ell k + r - 1}) - \nabla_r f_i(\boldsymbol{x}_{\ell k + r + j - 1}) \right\|^2$$

$$\leq \lim_{\ell \to \infty} 2 L_{\nabla f_i}^2 \sum_{j'=0}^{j-1} \left\| \boldsymbol{x}_{\ell k + r + j' - 1} - \boldsymbol{x}_{\ell k + r + j'} \right\|^2$$

$$\leq \lim_{\ell \to \infty} 2 L_{\nabla f_i}^2 \sum_{j'=0}^{j-1} \frac{\left\| \boldsymbol{x}_{\ell k + r + j' - 1} - \boldsymbol{x}_{\ell k + r + j'} \right\|^2}{\rho_{\ell k + r + j'}} \cdot \rho_{\ell k + r + j'}$$

$$\leq \lim_{\ell \to \infty} 2 L_{\nabla f_i}^2 \max_{j' \in \{0, 1, \cdots j-1\}} (\rho_{\ell k + r + j'}) \sum_{j'=0}^{j-1} \frac{\left\| \boldsymbol{x}_{\ell k + r + j' - 1} - \boldsymbol{x}_{\ell k + r + j'} \right\|^2}{\rho_{\ell k + r + j'}}$$

$$= \lim_{\ell \to \infty} 2 L_{\nabla f_i}^2 \max_{j' \in \{0, 1, \cdots j-1\}} (\rho_{\ell k + r + j'}) \cdot \lim_{\ell \to \infty} \sum_{j'=0}^{j-1} \frac{\left\| \boldsymbol{x}_{\ell k + r + j' - 1} - \boldsymbol{x}_{\ell k + r + j'} \right\|^2}{\rho_{\ell k + r + j'}}$$

$$= 0 \tag{40}$$

Combining the above consistency, we have

$$\lim_{t \to \infty} \left\| \hat{\nabla}_r f_i(\boldsymbol{x}_t) - \nabla_r f_i(\boldsymbol{x}_t) \right\| = 0. \tag{41}$$

Repeating this process for all block variables, we can conclude that

$$\lim_{t \to \infty} \left\| \hat{\nabla} f_i(\boldsymbol{x}_t) - \nabla f_i(\boldsymbol{x}_t) \right\| = 0. \tag{42}$$

**1.4. Gradient Consistency-Randomized:** Introducing a Bernoulli random variable $(R_{r,t})_{r=1}^k$ where $R_{r,t}$ is 1 if the $r$-th block variable is updated or 0 otherwise, we can write the difference between the sequences of approximate gradients $\hat{\nabla} f_i(\boldsymbol{x}_t)$ and true gradients $\nabla f_i(\boldsymbol{x}_t)$ as

$$\left\| \hat{\nabla} f_i(\boldsymbol{x}_t) - \nabla f_i(\boldsymbol{x}_t) \right\|^2$$

$$= \sum_{r=1}^k \left( R_{r,t} \left\| \rho_t \nabla_r f_i(\boldsymbol{x}_t, \boldsymbol{\xi}_t) + (1 - \rho_t) \hat{\nabla}_r f_i(\boldsymbol{x}_{t-1}) - \nabla_r f_i(\boldsymbol{x}_t) \right\|^2 \right. \tag{43}$$

$$\left. + (1 - R_{r,t}) \left\| \hat{\nabla}_r f_i(\boldsymbol{x}_{t-1}) - \nabla_r f_i(\boldsymbol{x}_t) \right\|^2 \right)$$

Since $\mathbb{E}(R_{r,t}|\mathcal{F}_t) = p_{r,t} \geq p_{\min}$, taking the expectation w.r.t. $(R_{r,t})_{r=1}^k$ and $\boldsymbol{\xi}_t$ conditioned on $\mathcal{F}_t$ yields

$$
\begin{aligned}
&\mathbb{E}\left[\left\|\hat{\nabla} f_i(\boldsymbol{x}_t) - \nabla f_i(\boldsymbol{x}_t)\right\|^2 \bigg| \mathcal{F}_t\right] \\
&= \sum_{r=1}^k \left( p_{r,t}\mathbb{E}\left[\left\|\rho_t \nabla_r f_i(\boldsymbol{x}_t, \boldsymbol{\xi}_t) + (1-\rho_t)\hat{\nabla}_r f_i(\boldsymbol{x}_{t-1}) - \nabla_r f_i(\boldsymbol{x}_t)\right\|^2 \bigg| \mathcal{F}_t\right] \right. \\
&\quad \left. + (1-p_{r,t})\mathbb{E}\left[\left\|\hat{\nabla}_r f_i(\boldsymbol{x}_{t-1}) - \nabla_r f_i(\boldsymbol{x}_t)\right\|^2 \bigg| \mathcal{F}_t\right] \right) \\
&= \sum_{r=1}^k \left( p_{r,t}\rho_t^2 \mathbb{E}\left[\left\|\nabla_r f_i(\boldsymbol{x}_t, \boldsymbol{\xi}_t) - \nabla_r f_i(\boldsymbol{x}_t)\right\|^2 \bigg| \mathcal{F}_t\right] \right. \\
&\quad + (1-p_{r,t}+p_{r,t}(1-\rho_t)^2)\left\|\hat{\nabla}_r f_i(\boldsymbol{x}_{t-1}) - \nabla_r f_i(\boldsymbol{x}_{t-1})\right\|^2 \\
&\quad + (1-p_{r,t}+p_{r,t}(1-\rho_t)^2)\left\|\nabla_r f_i(\boldsymbol{x}_{t-1}) - \nabla_r f_i(\boldsymbol{x}_t)\right\|^2 \\
&\quad \left. + 2(1-p_{r,t}+p_{r,t}(1-\rho_t)^2)(\hat{\nabla}_r f_i(\boldsymbol{x}_{t-1}) - \nabla_r f_i(\boldsymbol{x}_{t-1}))^\top(\nabla_r f_i(\boldsymbol{x}_{t-1}) - \nabla_r f_i(\boldsymbol{x}_t)) \right) \\
&\leq \rho_t^2 \mathbb{E}\left[\left\|\nabla f_i(\boldsymbol{x}_t, \boldsymbol{\xi}_t) - \nabla f_i(\boldsymbol{x}_t)\right\|^2 \bigg| \mathcal{F}_t\right] \\
&\quad + (1 + \frac{p_{\min}\rho_t}{2})(1-p_{\min}+p_{\min}(1-\rho_t)^2)\left\|\hat{\nabla} f_i(\boldsymbol{x}_{t-1}) - \nabla f_i(\boldsymbol{x}_{t-1})\right\|^2 \\
&\quad + (1 + \frac{2}{p_{\min}\rho_t})(1-p_{\min}+p_{\min}(1-\rho_t)^2)\left\|\nabla f_i(\boldsymbol{x}_{t-1}) - \nabla f_i(\boldsymbol{x}_t)\right\|^2 \\
&\leq \rho_t^2\varepsilon^2 + (1 + \frac{2}{p_{\min}\rho_t})L_{\nabla f_i}^2\left\|\boldsymbol{x}_{t-1} - \boldsymbol{x}_t\right\|^2 + (1 - \frac{p_{\min}\rho_t}{2})\left\|\hat{\nabla} f_i(\boldsymbol{x}_{t-1}) - \nabla f_i(\boldsymbol{x}_{t-1})\right\|^2
\end{aligned}
\tag{44}
$$

where the last inequality is due to $(1 + \frac{p_{\min}\rho_t}{2})(1-p_{\min}+p_{\min}(1-\rho_t)^2) \leq (1+\frac{p_{\min}\rho_t}{2})(1-p_{\min}+p_{\min}(1-\rho_t)) \leq (1+\frac{p_{\min}\rho_t}{2})(1-p_{\min}\rho_t) \leq 1 - \frac{p_{\min}\rho_t}{2}$ Hence, it follows from the same analysis as the gradient consistency that

$$
\lim_{t\to\infty}\left\|\hat{\nabla} f_i(\boldsymbol{x}_t) - \nabla f_i(\boldsymbol{x}_t)\right\| = 0.
\tag{45}
$$

## 2. Non-asymptotic Analysis

**2.1. Non-asymptotic Value:** Recall the upper bound in Equation (30), we compute the expected value with respect to all random variables starting from the initial step, i.e., $t = 0$.

$$
\begin{aligned}
&\mathbb{E}\left[\left|\hat{f}_i(\boldsymbol{x}_t) - f_i(\boldsymbol{x}_t)\right|^2\right] \\
&\leq \left(1 - \frac{\rho_t}{2}\right)\mathbb{E}\left[\left|\hat{f}_i(\boldsymbol{x}_{t-1}) - f_i(\boldsymbol{x}_{t-1})\right|^2\right] + (1 + \frac{2}{\rho_t})L_{f_i}^2\gamma_t^2 D^2 + \rho_t^2\varepsilon^2,
\end{aligned}
\tag{46}
$$

By setting $\rho_t = \frac{4}{(t+16)^{1/2}}$ and $\gamma_t = \frac{2}{(t+16)^{3/4}}$, we have

$$
\begin{aligned}
&\mathbb{E}\left[\left|\hat{f}_i(\boldsymbol{x}_t) - f_i(\boldsymbol{x}_t)\right|^2\right] \\
&\leq \left(1 - \frac{2}{(t+16)^{1/2}}\right)\mathbb{E}\left[\left|\hat{f}_i(\boldsymbol{x}_{t-1}) - f_i(\boldsymbol{x}_{t-1})\right|^2\right] + \frac{4L_{f_i}^2 D^2}{(t+16)^{3/2}} + \frac{2L_{f_i}^2 D^2 + 16\varepsilon^2}{t+16} \\
&\leq \left(1 - \frac{2}{(t+16)^{1/2}}\right)\mathbb{E}\left[\left|\hat{f}_i(\boldsymbol{x}_{t-1}) - f_i(\boldsymbol{x}_{t-1})\right|^2\right] + \frac{6L_{f_i}^2 D^2 + 16\varepsilon^2}{t+16}
\end{aligned}
\tag{47}
$$

Let $\phi_t = \mathbb{E}\left[\left|\hat{f}_i(\boldsymbol{x}_t) - f_i(\boldsymbol{x}_t)\right|^2\right], t_0 = 16, \alpha = 1/2, c = 2$, and $b = 6L_{f_i}^2 D^2 + 16\varepsilon^2$. From Lemma 4, the sequence $\mathbb{E}\left[\left|\hat{f}_i(\boldsymbol{x}_t) - f_i(\boldsymbol{x}_t)\right|^2\right]$ converges to zeros at the following rate

$$\mathbb{E}\left[\left|\hat{f}_i(\boldsymbol{x}_t) - f_i(\boldsymbol{x}_t)\right|^2\right] \leq \frac{\max\{17^{1/2}(\hat{f}_i(\boldsymbol{x}_0) - f_i(\boldsymbol{x}_0)), 6L_{f_i}^2 D^2 + 16\varepsilon^2\}}{(t+17)^{1/2}}, \tag{48}$$

which implies $\mathbb{E}\left[\left|\hat{f}_i(\boldsymbol{x}_t) - f_i(\boldsymbol{x}_t)\right|^2\right]$ approaches zero at a rate of $\mathcal{O}(t^{-1/2})$.

**2.2. Non-asymptotic Gradient:** Recall the upper bound in Equation (34), we compute the expected value with respect to all random variables starting from the initial step, i.e., $t = 0$.

$$\mathbb{E}\left[\left\|\hat{\nabla}f_i(\boldsymbol{x}_t) - \nabla f_i(\boldsymbol{x}_t)\right\|^2\right] \tag{49}$$
$$\leq \left(1 - \frac{\rho_t}{2}\right)\mathbb{E}\left[\left\|\hat{\nabla}f_i(\boldsymbol{x}_{t-1}) - \nabla f_i(\boldsymbol{x}_{t-1})\right\|^2\right] + (1 + \frac{2}{\rho_t})L_{\nabla f_i}^2 \gamma_t^2 D^2 + \rho_t^2 \varepsilon^2,$$

By setting $\rho_t = \frac{4}{(t+16)^{1/2}}$ and $\gamma_t = \frac{2}{(t+16)^{3/4}}$, we have

$$\mathbb{E}\left[\left\|\hat{\nabla}f_i(\boldsymbol{x}_t) - \nabla f_i(\boldsymbol{x}_t)\right\|^2\right]$$
$$\leq \left(1 - \frac{2}{(t+16)^{1/2}}\right)\mathbb{E}\left[\left\|\hat{\nabla}f_i(\boldsymbol{x}_{t-1}) - \nabla f_i(\boldsymbol{x}_{t-1})\right\|^2\right] + \frac{4L_{\nabla f_i}^2 D^2}{(t+16)^{3/2}} + \frac{2L_{\nabla f_i}^2 D^2 + 16\varepsilon^2}{t+16}$$
$$\leq \left(1 - \frac{2}{(t+16)^{1/2}}\right)\mathbb{E}\left[\left\|\hat{\nabla}f_i(\boldsymbol{x}_{t-1}) - \nabla f_i(\boldsymbol{x}_{t-1})\right\|^2\right] + \frac{6L_{\nabla f_i}^2 D^2 + 16\varepsilon^2}{t+16} \tag{50}$$

Let $\phi_t = \mathbb{E}\left[\left\|\hat{\nabla}f_i(\boldsymbol{x}_t) - \nabla f_i(\boldsymbol{x}_t)\right\|^2\right], t_0 = 16, \alpha = 1/2, c = 2$, and $b = 6L_{\nabla f_i}^2 D^2 + 16\varepsilon^2$. From Lemma 4, the sequence $\mathbb{E}\left[\left\|\hat{\nabla}f_i(\boldsymbol{x}_t) - \nabla f_i(\boldsymbol{x}_t)\right\|^2\right]$ converges to zeros at the following rate

$$\mathbb{E}\left[\left\|\hat{\nabla}f_i(\boldsymbol{x}_t) - \nabla f_i(\boldsymbol{x}_t)\right\|^2\right] \leq \frac{\max\{17^{1/2}\left\|\hat{\nabla}f_i(\boldsymbol{x}_0) - \nabla f_i(\boldsymbol{x}_0)\right\|^2, 6L_{f_i}^2 D^2 + 16\varepsilon^2\}}{(t+17)^{1/2}}, \tag{51}$$

which implies $\mathbb{E}\left[\left\|\hat{\nabla}f_i(\boldsymbol{x}_t) - \nabla f_i(\boldsymbol{x}_t)\right\|^2\right]$ approaches zero at a rate of $\mathcal{O}(t^{-1/2})$.

**2.3. Non-asymptotic Gradient-Cyclic:** Recall the upper bound in Equation (38), we compute the expected value with respect to all random variables starting from the initial step, i.e., $t = 0$.

$$\mathbb{E}\left[\left\|\hat{\nabla}_r f_i(\boldsymbol{x}_{\ell k+r-1}) - \nabla_r f_i(\boldsymbol{x}_{\ell k+r-1})\right\|^2\right]$$
$$\leq (1 - \frac{\rho_{\ell k+r-1}}{2})\mathbb{E}\left[\left\|\hat{\nabla}_r f_i(\boldsymbol{x}_{(\ell-1)k+r-1} - \nabla_r f_i(\boldsymbol{x}_{(\ell-1)k+r-1})\right\|^2\right] \tag{52}$$
$$+ (1 + \frac{2}{\rho_{\ell k+r-1}})L_{\nabla f_i}^2 \gamma_{\ell k+r-1}^2 D^2 + \rho_{\ell k+r-1}^2 \varepsilon^2,$$

By setting $\rho_t = \frac{4}{(t+16)^{1/2}}$ and $\gamma_t = \frac{2}{(t+16)^{3/4}}$, we have

$$\mathbb{E}\left[\left\|\hat{\nabla}_r f_i(\boldsymbol{x}_{\ell k+r-1}) - \nabla_r f_i(\boldsymbol{x}_{\ell k+r-1})\right\|^2\right]$$
$$\leq \left(1 - \frac{2/k^{1/2}}{(\ell + (r+15)/k)^{1/2}}\right)\mathbb{E}\left[\left\|\hat{\nabla}_r f_i(\boldsymbol{x}_{(\ell-1)k+r-1} - \nabla_r f_i(\boldsymbol{x}_{(\ell-1)k+r-1})\right\|^2\right] \tag{53}$$
$$+ \frac{(6L_{\nabla f_i}^2 D^2 + 16\varepsilon^2)/k}{\ell + (r+15)/k}$$

Let $\phi_l = \mathbb{E}\left[\left\|\hat{\nabla}_r f_i(\boldsymbol{x}_{\ell k+r-1}) - \nabla_r f_i(\boldsymbol{x}_{\ell k+r-1})\right\|^2\right]$, $l_0 = (r+15)/k, \alpha = 1/2, c = 2/k^{1/2}$, and $b = (6L_{\nabla f_i}^2 D^2 + 16\varepsilon^2)/k$. Hence, the sequence $\mathbb{E}\left[\left\|\hat{\nabla}_r f_i(\boldsymbol{x}_{\ell k+r-1}) - \nabla_r f_i(\boldsymbol{x}_{\ell k+r-1})\right\|^2\right]$ converges to zeros at the following rate

$$\mathbb{E}\left[\left\|\hat{\nabla}_r f_i(\boldsymbol{x}_{\ell k+r-1}) - \nabla_r f_i(\boldsymbol{x}_{\ell k+r-1})\right\|^2\right]$$

$$\leq \frac{\max\{((r+15)/k+1)^{1/2}\left\|\hat{\nabla}_r f_i(\boldsymbol{x}_{r-1}) - \nabla_r f_i(\boldsymbol{x}_{r-1})\right\|^2, (6L_{\nabla f_i}^2 D^2 + 16\varepsilon^2)/(2k^{1/2}-k)\}}{(l+(r+15)/k+1)^{1/2}}. \tag{54}$$

Moreover, the gradient over iteration $\ell k + r + j - 1, j = 1, 2, \cdots, K - 1$ can be upper-bounded by

$$\left\|\hat{\nabla}_r f_i(\boldsymbol{x}_{\ell k+r+j-1}) - \nabla_r f_i(\boldsymbol{x}_{\ell k+r+j-1})\right\|^2$$

$$= \left\|\hat{\nabla}_r f_i(\boldsymbol{x}_{\ell k+r-1}) - \nabla_r f_i(\boldsymbol{x}_{\ell k+r-1}) + \nabla_r f_i(\boldsymbol{x}_{\ell k+r-1}) - \nabla_r f_i(\boldsymbol{x}_{\ell k+r+j-1})\right\|^2$$

$$\leq 2\left(\left\|\hat{\nabla}_r f_i(\boldsymbol{x}_{\ell k+r-1}) - \nabla_r f_i(\boldsymbol{x}_{\ell k+r-1})\right\|^2 + \|\nabla_r f_i(\boldsymbol{x}_{\ell k+r-1}) - \nabla_r f_i(\boldsymbol{x}_{\ell k+r+j-1})\|^2\right)$$

$$\leq 2L_{\nabla f_i}^2 \sum_{j'=0}^{j-1}\|\boldsymbol{x}_{\ell k+r+j'-1} - \boldsymbol{x}_{\ell k+r+j'}\|^2 + 2\left\|\hat{\nabla}_r f_i(\boldsymbol{x}_{\ell k+r-1}) - \nabla_r f_i(\boldsymbol{x}_{\ell k+r-1})\right\|^2 \tag{55}$$

$$\leq 2L_{\nabla f_i}^2 D^2 \sum_{j'=0}^{j-1}\gamma_{\ell k+r+j'}^2 + 2\left\|\hat{\nabla}_r f_i(\boldsymbol{x}_{\ell k+r-1}) - \nabla_r f_i(\boldsymbol{x}_{\ell k+r-1})\right\|^2$$

$$= 2L_{\nabla f_i}^2 D^2 \sum_{j'=0}^{j-1}\frac{4}{(\ell k + r + j' + 16)^{3/2}} + 2\left\|\hat{\nabla}_r f_i(\boldsymbol{x}_{\ell k+r-1}) - \nabla_r f_i(\boldsymbol{x}_{\ell k+r-1})\right\|^2$$

$$\leq 2L_{\nabla f_i}^2 D^2 \frac{4j}{(\ell k + r + 16)^{3/2}} + 2\left\|\hat{\nabla}_r f_i(\boldsymbol{x}_{\ell k+r-1}) - \nabla_r f_i(\boldsymbol{x}_{\ell k+r-1})\right\|^2$$

Taking expectation with respect to both sides, we have

$$\mathbb{E}\left[\left\|\hat{\nabla}_r f_i(\boldsymbol{x}_{\ell k+r+j-1}) - \nabla_r f_i(\boldsymbol{x}_{\ell k+r+j-1})\right\|^2\right]$$

$$\leq 2L_{\nabla f_i}^2 D^2 \frac{4j}{(\ell k + r + 16)^{3/2}} + 2\mathbb{E}\left[\left\|\hat{\nabla}_r f_i(\boldsymbol{x}_{\ell k+r-1}) - \nabla_r f_i(\boldsymbol{x}_{\ell k+r-1})\right\|^2\right]$$

$$\leq 2L_{\nabla f_i}^2 D^2 \frac{4j}{(\ell k + r + 16)^{3/2}}$$

$$+ \frac{2\max\{((r+15)/k+1)^{1/2}\left\|\hat{\nabla}_r f_i(\boldsymbol{x}_{r-1}) - \nabla_r f_i(\boldsymbol{x}_{r-1})\right\|^2, (6L_{\nabla f_i}^2 D^2 + 16\varepsilon^2)/(2k^{1/2}-k)\}}{(l+(r+15)/k+1)^{1/2}} \tag{56}$$

Combining the above upper bounds, we have $\mathbb{E}\left[\left\|\hat{\nabla}_r f_i(\boldsymbol{x}_t) - \nabla_r f_i(\boldsymbol{x}_t)\right\|^2\right]$ approaches zero at a rate of $\mathcal{O}(t^{-1/2})$. Repeating this process for all block variables, we can conclude that $\mathbb{E}\left[\left\|\hat{\nabla} f_i(\boldsymbol{x}_t) - \nabla f_i(\boldsymbol{x}_t)\right\|^2\right]$ approaches zero at a rate of $\mathcal{O}(t^{-1/2})$.

**2.4. Non-asymptotic Gradient-Randomized:** Recall the upper bound in Equation (44), we compute the expected value with respect to all random variables starting from the initial step, i.e., $t = 0$.

$$\mathbb{E}\left[\left\|\hat{\nabla} f_i(\boldsymbol{x}_t) - \nabla f_i(\boldsymbol{x}_t)\right\|^2\right]$$

$$\leq \left(1 - \frac{p_{\min}\rho_t}{2}\right)\mathbb{E}\left[\left\|\hat{\nabla} f_i(\boldsymbol{x}_{t-1}) - \nabla f_i(\boldsymbol{x}_{t-1})\right\|^2\right] + (1 + \frac{2}{p_{\min}\rho_t})L_{\nabla f_i}^2 \gamma_t^2 D^2 + \rho_t^2 \varepsilon^2, \tag{57}$$

By setting $\rho_t = \frac{4}{(t+16)^{1/2}}$ and $\gamma_t = \frac{2}{(t+16)^{3/4}}$, we have

$$\mathbb{E}\left[\left\|\hat{\nabla}f_i(\boldsymbol{x}_t) - \nabla f_i(\boldsymbol{x}_t)\right\|^2\right]$$

$$\leq \left(1 - \frac{2p_{\min}}{(t+16)^{1/2}}\right)\mathbb{E}\left[\left\|\hat{\nabla}f_i(\boldsymbol{x}_{t-1}) - \nabla f_i(\boldsymbol{x}_{t-1})\right\|^2\right] + \frac{4L_{\nabla f_i}^2 D^2}{(t+16)^{3/2}} + \frac{2L_{\nabla f_i}^2 D^2/p_{\min} + 16\varepsilon^2}{t+16}$$

$$\leq \left(1 - \frac{2p_{\min}}{(t+16)^{1/2}}\right)\mathbb{E}\left[\left\|\hat{\nabla}f_i(\boldsymbol{x}_{t-1}) - \nabla f_i(\boldsymbol{x}_{t-1})\right\|^2\right] + \frac{4L_{\nabla f_i}^2 D^2 + 2L_{\nabla f_i}^2 D^2/p_{\min} + 16\varepsilon^2}{t+16}$$

(58)

Let $\phi_t = \mathbb{E}\left[\left\|\hat{\nabla}f_i(\boldsymbol{x}_t) - \nabla f_i(\boldsymbol{x}_t)\right\|^2\right], t_0 = 16, \alpha = 1/2, c = 2p_{\min}$, and $b = 4L_{\nabla f_i}^2 D^2 + 2L_{\nabla f_i}^2 D^2/p_{\min} + 16\varepsilon^2$. From Lemma 4, the sequence $\mathbb{E}\left[\left\|\hat{\nabla}f_i(\boldsymbol{x}_t) - \nabla f_i(\boldsymbol{x}_t)\right\|^2\right]$ converges to zeros at the following rate

$$\mathbb{E}\left[\left\|\hat{\nabla}f_i(\boldsymbol{x}_t) - \nabla f_i(\boldsymbol{x}_t)\right\|^2\right]$$

$$\leq \frac{\max\{17^{1/2}\left\|\hat{\nabla}f_i(\boldsymbol{x}_0) - \nabla f_i(\boldsymbol{x}_0)\right\|^2, (4L_{\nabla f_i}^2 D^2 + 2L_{\nabla f_i}^2 D^2/p_{\min} + 16\varepsilon^2)/(2p_{\min} - 1)\}}{(t+17)^{1/2}},$$

(59)

which implies $\mathbb{E}\left[\left\|\hat{\nabla}f_i(\boldsymbol{x}_t) - \nabla f_i(\boldsymbol{x}_t)\right\|^2\right]$ approaches zero at a rate of $\mathcal{O}(t^{-1/2})$.

## C.2 PROOF OF THEOREM 2

To begin with, we introduce the auxiliary variable $u_0$, and then the Tchebycheff Problem (2) can be equivalently reformulated as the following constrained optimization problem

$$\begin{aligned} \min_{\boldsymbol{x}\in\mathcal{X}, u_0\in\mathbb{R}} \quad & u_0 \\ \text{s.t.} \quad & \lambda_i\left(f_i(\boldsymbol{x}) - z_i\right) - u_0 \leq 0, i = 1, \cdots, m, \\ & g_j(\boldsymbol{x}) \leq 0, j = 1, \cdots, n. \end{aligned}$$

(60)

**Local optimality:** Suppose $\boldsymbol{x}^*$ is a KKT point of the original Tchebycheff problem (2). Define $u_0^* = \max_{1\leq i\leq m}\{\lambda_i(f_i(\boldsymbol{x}^*) - z_i)\}$. It then follows that $(\boldsymbol{x}^*, u_0^*)$ is a KKT point of Problem (60), which is required to satisfy

$$\boldsymbol{0} \in \eta_0 \begin{bmatrix} \boldsymbol{0} \\ 1 \end{bmatrix} + \sum_{i=1}^m \eta_i \begin{bmatrix} \lambda_i\nabla f_i(\boldsymbol{x}^*) \\ -1 \end{bmatrix} + \sum_{j=1}^n \eta_{m+j} \begin{bmatrix} \nabla g_j(\boldsymbol{x}^*) \\ 0 \end{bmatrix} + \begin{bmatrix} N_{\mathcal{X}}(\boldsymbol{x}^*) \\ 0 \end{bmatrix},$$ (61a)

$$\eta_i\left(\lambda_i\left(f_i(\boldsymbol{x}^*) - z_i\right) - u_0^*\right) = 0,$$ (61b)

$$\eta_{m+j}g_j(\boldsymbol{x}^*) = 0.$$ (61c)

Furthermore, we define

$$u_1^* \triangleq \max\left\{0, \lambda_i\left(f_i(\boldsymbol{x}^*) - z_i\right) - u_0^*, \forall i, g_j(\boldsymbol{x}^*), \forall j\right\}.$$ (62)

According to Lemma 2, there exists a sufficiently large penalty coefficient $c$ such that $(\boldsymbol{x}^*, u_0^*, u_1^*)$ becomes the local minimizer of the following problem

$$\begin{aligned} \min_{\boldsymbol{x}\in\mathcal{X}, u_0\in\mathbb{R}, u_1\geq 0} \quad & u_0 + cu_1 \\ \text{subject to} \quad & \lambda_i\left(f_i(\boldsymbol{x}) - z_i\right) - u_0 \leq u_1, \quad i \in [m], \\ & g_j(\boldsymbol{x}) \leq u_1, \quad j \in [n]. \end{aligned}$$

(63)

To further analyze the problem structure, we introduce the variable substitution $u_2 = u_0 + u_1$, and transform Problem Equation (63) into an equivalent decomposed form

$$
\begin{aligned}
\min_{\boldsymbol{x} \in \mathcal{X}, u_0 \in \mathbb{R}, u_1 \geq 0, u_2 \in \mathbb{R}} \quad & u_2 + (c-1)u_1 \\
\text{subject to} \quad & \lambda_i \left( f_i(\boldsymbol{x}) - z_i \right) - u_2 \leq 0, \quad i \in [m], \\
& g_j(\boldsymbol{x}) \leq u_1, \quad j \in [n], \\
& u_2 = u_0 + u_1.
\end{aligned}
\tag{64}
$$

Since $u_0$ can be explicitly reconstructed through the relation $y_2 = y_0 + y_1$ for any optimized values of $u_2$ and $u_1$, the variable $u_0$ becomes redundant and can be eliminated from the problem formulation without loss of generality. By setting $y_0 = u_2, y_1 = u_1, \beta = c - 1$, we can get that the point $(\boldsymbol{x}^*, u_0^* + u_1^*, u_1^*)$ is the KKT point of the desired Problem (14), i.e.,

$$
\boldsymbol{0} \in \eta_0 \begin{bmatrix} \boldsymbol{0} \\ 1 \\ \beta \end{bmatrix} + \sum_{i=1}^m \eta_i \begin{bmatrix} \lambda_i \nabla f_i(\boldsymbol{x}^*) \\ -1 \\ 0 \end{bmatrix} + \sum_{j=1}^n \eta_{m+j} \begin{bmatrix} \nabla g_j(\boldsymbol{x}^*) \\ 0 \\ -1 \end{bmatrix} + \begin{bmatrix} N_{\mathcal{X}}(\boldsymbol{x}^*) \\ 0 \\ N_{\mathbb{R}_+}(u_1^*) \end{bmatrix}, \tag{65a}
$$

$$
\eta_i \left( \lambda_i \left( f_i(\boldsymbol{x}^*) - z_i \right) - u_0^* - u_1^* \right) = 0, \tag{65b}
$$

$$
\eta_{m+j}(g_j(\boldsymbol{x}^*) - u_1^*) = 0. \tag{65c}
$$

A comparison between Equations (61) and (65) demonstrates that when $u_1^* = 0$, the point $(\boldsymbol{x}^*, u_0^*, 0)$, equivalent to $(\boldsymbol{x}^*, y_0^*, 0)$, is a KKT point of the Problem (14).

**Global optimality:** The feasible region of points $(\boldsymbol{x}, y_0, y_1)$ in Problem (14) can be partitioned into two distinct cases based on whether the parameter $y_1$ satisfies the threshold condition $y_1 < \epsilon$ or $y_1 \geq \epsilon$ for some small $\epsilon$.

*Case A $y_1 < \epsilon$:* Based on the local optimality conditions at each solution point $(\boldsymbol{x}^*, y_0^*, y_1^*)$, we derive the fundamental inequality $y_0 + \beta y_1 \geq y_0^*$. Furthermore, since $\boldsymbol{x}^\star$ constitutes the global optimum for Problem (2), the inherent problem structure enforces the dominance relation $y_0^* \geq y_0^\star$ between these solutions. Combining these results through transitive inequality propagation, we ultimately establish the key relationship: $y_0 + \beta y_1 \geq y_0^\star$.

*Case B $y_1 \geq \epsilon$:* Let $D'$ denote the gap between the upper and lower bounds of $y_0$. For any regularization parameter $\beta$ satisfying $\beta \geq D'/\epsilon$, the inequality $y_0 + \beta y_1 \geq y_0 + D' > y_0^*$ holds.

Hence $(\boldsymbol{x}^\star, y_0^\star, 0)$ is a globally optimal solution of the desired Problem (14).

### C.3 PROOF OF THEOREM 3

If $(\boldsymbol{x}^*, y_0^*, 0)$ satisfies KKT condition of Problem (14), we have

$$
\boldsymbol{0} \in \eta_0 \begin{bmatrix} \boldsymbol{0} \\ 1 \\ \beta \end{bmatrix} + \sum_{i=1}^m \eta_i \begin{bmatrix} \lambda_i \nabla f_i(\boldsymbol{x}^*) \\ -1 \\ 0 \end{bmatrix} + \sum_{j=1}^n \eta_{m+j} \begin{bmatrix} \nabla g_j(\boldsymbol{x}^*) \\ 0 \\ -1 \end{bmatrix} + \begin{bmatrix} N_{\mathcal{X}}(\boldsymbol{x}^*) \\ 0 \\ N_{\mathbb{R}_+}(0) \end{bmatrix}, \tag{66a}
$$

$$
\eta_i \left( \lambda_i \left( f_i(\boldsymbol{x}^*) - z_i \right) - y_0^* \right) = 0, \tag{66b}
$$

$$
\eta_{m+j} g_j(\boldsymbol{x}^*) = 0. \tag{66c}
$$

By setting $\rho_i = \eta_i \lambda_i, i = 1, \cdots, m$ and $\mu_j = \eta_{m+j}$, we have the $\boldsymbol{x}^*$ satisfies

$$
\begin{aligned}
\boldsymbol{0} &\in \sum_{i=1}^m \rho_i \nabla f_i(\boldsymbol{x}^*) + \sum_{j=1}^n \mu_j \nabla g_j(\boldsymbol{x}^*) + N_{\mathcal{X}}(\boldsymbol{x}^*), \\
0 &= \mu_{m+j} g_j(\boldsymbol{x}^*),
\end{aligned}
\tag{67}
$$

which implies $\boldsymbol{x}^*$ is the weak Pareto solution of the original problem.

## D EXTENSION

### D.1 PARAMETER REGULARIZATION PROBLEM

Regularization techniques play a crucial role in mitigating statistical overfitting in a predictive model. This section demonstrates the capability of our proposed CSMOO framework in handling

composite optimization problems characterized by the simultaneous presence of multiple objectives, stochastic uncertainty, and structural regularization terms. The objective function and constraints are, respectively, given by

$$
\begin{aligned}
f_i(\boldsymbol{x}) &\triangleq \mathbb{E}_{\boldsymbol{\xi}}(f_i(\boldsymbol{x}, \boldsymbol{\xi})) + r_{f_i}(\boldsymbol{x}) \\
g_j(\boldsymbol{x}) &\triangleq \mathbb{E}_{\boldsymbol{\xi}}(g_j(\boldsymbol{x}, \boldsymbol{\xi})) + r_{g_j}(\boldsymbol{x})
\end{aligned}
\tag{68}
$$

where the nonsmooth convex terms $r_{f_i}(\boldsymbol{x})$ and $r_{g_j}(\boldsymbol{x})$ serve as regularization components to ensure stability and promote sparsity.

To address the nonsmooth nature of the original functions, an auxiliary vector $\boldsymbol{s} \triangleq (s_{f_1}, \cdots, s_{f_m}, s_{g_1}, \cdots, s_{g_n})$ is introduced, which facilitates the transformation of the composite functions and the associated decision space into an equivalent smooth reformulation that is more amenable to optimization. The objective functions, the constraints, and the decision space are, respectively, transformed into

$$
\begin{aligned}
\bar{f}_i(\boldsymbol{x}, \boldsymbol{s}) &\triangleq \mathbb{E}_{\boldsymbol{\xi}}(f_i(\boldsymbol{x}, \boldsymbol{\xi})) + s_{f_i} \\
\bar{g}_j(\boldsymbol{x}, \boldsymbol{s}) &\triangleq \mathbb{E}_{\boldsymbol{\xi}}(g_j(\boldsymbol{x}, \boldsymbol{\xi})) + s_{g_j} \\
\bar{\mathcal{X}} &\triangleq \{(\boldsymbol{x}, \boldsymbol{s}) : \boldsymbol{x} \in \mathcal{X}, r_{f_i}(\boldsymbol{x}) \le s_{f_i}, \forall i, r_{g_j}(\boldsymbol{x}) \le s_{g_j}, \forall j\}.
\end{aligned}
\tag{69}
$$

Through this transformation, the nonsmooth and nonconvex multi-objective optimization problem is equivalently reformulated into the standard CSMOO form in Equation (18). The reformulation preserves both the feasibility structure and the optimality properties of the original problem, thereby ensuring theoretical equivalence while enabling the direct application of our proposed framework.

### D.2 Constrained Multi-Objective Federated Learning (CMOFL) Task

This section aims to adapt our proposed CSMOO framework to a CMOFL task, where a central server coordinates with $p$ distributed clients to maximize the performance across $m$ tasks while respecting $n$ constraints. Assume the $k$-th client maintains $l_k$ local samples and define the $l$-th local sample at the $k$-th client as $\boldsymbol{\xi}_{k,l}$. Then the total number of samples is $L = \sum_k l_k$. At the $k$-th client, the local loss function of the $i$-th task and the $j$-th local constraint are, respectively, defined as

$$
\begin{aligned}
f_{i,k}(\boldsymbol{x}) &\triangleq \frac{1}{l_k} \sum_{l=1}^{l_k} f_{i,k}(\boldsymbol{x}, \boldsymbol{\xi}_{k,l}), \\
g_{j,k}(\boldsymbol{x}) &\triangleq \frac{1}{l_k} \sum_{l=1}^{l_k} g_{j,k}(\boldsymbol{x}, \boldsymbol{\xi}_{k,l}).
\end{aligned}
\tag{70}
$$

Furthermore, the global loss function of the $i$-th task and the $j$-th global constraint are

$$
f_i(\boldsymbol{x}) = \sum_{k=1}^{p} \frac{l_k}{L} f_{i,k}(\boldsymbol{x}), \quad g_j(\boldsymbol{x}) = \sum_{k=1}^{p} \frac{l_k}{L} g_{j,k}(\boldsymbol{x}).
\tag{71}
$$

The central server performs three critical stages to train the global model:

1. Collecting local loss function estimates from all clients, including the value estimation

$$
\begin{aligned}
\hat{f}_{i,k}(\boldsymbol{x}_t) &= \rho_t f_{i,k}(\boldsymbol{x}_t, \boldsymbol{\xi}_{k,l,t}) + (1 - \rho_t)\hat{f}_{i,k}(\boldsymbol{x}_{t-1}), \\
\hat{g}_{j,k}(\boldsymbol{x}_t) &= \rho_t g_{j,k}(\boldsymbol{x}_t, \boldsymbol{\xi}_{k,l,t}) + (1 - \rho_t)\hat{g}_{j,k}(\boldsymbol{x}_{t-1}),
\end{aligned}
\tag{72}
$$

and gradient estimation

$$
\begin{aligned}
\hat{\nabla} f_{i,k}(\boldsymbol{x}_t) &= \rho_t \nabla f_{i,k}(\boldsymbol{x}_t, \boldsymbol{\xi}_{k,l,t}) + (1 - \rho_t)\hat{\nabla} f_{i,k}(\boldsymbol{x}_{t-1}), \\
\hat{\nabla} g_{j,k}(\boldsymbol{x}_t) &= \rho_t \nabla g_{j,k}(\boldsymbol{x}_t, \boldsymbol{\xi}_{k,l,t}) + (1 - \rho_t)\hat{\nabla} g_{j,k}(\boldsymbol{x}_{t-1}).
\end{aligned}
\tag{73}
$$

2. Aggregating the function values and gradient estimates of all local loss functions

$$
\begin{aligned}
\hat{f}_i(\boldsymbol{x}_t) &= \sum_{k=1}^{p} \frac{l_k}{L} \hat{f}_{i,k}(\boldsymbol{x}_t), \qquad \hat{g}_j(\boldsymbol{x}_t) = \sum_{k=1}^{p} \frac{l_k}{L} \hat{g}_{j,k}(\boldsymbol{x}_t) \\
\hat{\nabla} f_i(\boldsymbol{x}_t) &= \sum_{k=1}^{p} \frac{l_k}{L} \hat{\nabla} f_{i,k}(\boldsymbol{x}_t), \quad \hat{\nabla} g_j(\boldsymbol{x}_t) = \sum_{k=1}^{p} \frac{l_k}{L} \hat{\nabla} g_{j,k}(\boldsymbol{x}_t)
\end{aligned}
\tag{74}
$$

3. Distributing the updated variable $\boldsymbol{x}_{t+1}$, which is computed by solving the surrogate quadratically constrained quadratic program.

Building upon the analytical framework established in Theorem 1, we conclude that both value and gradient consistency properties hold for the local functions maintained at individual client nodes. By rigorous mathematical derivation, these guarantees are shown to extend to the global function aggregated at the central server. More formally, it follows from

$$\lim_{t\to\infty} \left| \hat{f}_i(\boldsymbol{x}_t) - f_i(\boldsymbol{x}_t) \right| = \lim_{t\to\infty} \left| \sum_{k=1}^{p} \frac{l_k}{L} \left( \hat{f}_{i,k}(\boldsymbol{x}_t) - f_{i,k}(\boldsymbol{x}_t) \right) \right| \tag{75}$$

$$\leq \lim_{t\to\infty} \sum_{k=1}^{p} \frac{l_k}{L} \left| \hat{f}_{i,k}(\boldsymbol{x}_t) - f_{i,k}(\boldsymbol{x}_t) \right| \tag{76}$$

$$= \sum_{k=1}^{p} \frac{l_k}{L} \lim_{t\to\infty} \left| \hat{f}_{i,k}(\boldsymbol{x}_t) - f_{i,k}(\boldsymbol{x}_t) \right| \tag{77}$$

$$= 0 \tag{78}$$

that the value consistency $\lim_{t\to\infty} \left| \hat{f}_i(\boldsymbol{x}_t) - f_i(\boldsymbol{x}_t) \right| = 0$ holds at the central server. Similarly, from

$$\lim_{t\to\infty} \left\| \hat{\nabla} f_i(\boldsymbol{x}_t) - \nabla f_i(\boldsymbol{x}_t) \right\| = \lim_{t\to\infty} \left\| \sum_{k=1}^{p} \frac{l_k}{L} \left( \hat{\nabla} f_{i,k}(\boldsymbol{x}_t) - \nabla f_{i,k}(\boldsymbol{x}_t) \right) \right\|$$

$$\leq \lim_{t\to\infty} \sum_{k=1}^{p} \frac{l_k}{L} \left\| \hat{\nabla} f_{i,k}(\boldsymbol{x}_t) - \nabla f_{i,k}(\boldsymbol{x}_t) \right\| \tag{79}$$

$$= \sum_{k=1}^{p} \frac{l_k}{L} \lim_{t\to\infty} \left\| \hat{\nabla} f_{i,k}(\boldsymbol{x}_t) - \nabla f_{i,k}(\boldsymbol{x}_t) \right\|$$

$$= 0,$$

we have the gradient consistency $\lim_{t\to\infty} \left\| \hat{\nabla} f_i(\boldsymbol{x}_t) - \nabla f_i(\boldsymbol{x}_t) \right\| = 0$ holds at the central server.

# E ADDITIONAL EXPERIMENTS, DETAILS AND RESULTS

## E.1 ADDITIONAL EXPERIMENT DETAILS AND RESULTS OF THE SYNTHETIC PROBLEM

**Experimental setup.** To estimate expectations, we use a single-sample exponential moving average (EMA) for both values and gradients: $\rho_t = (t+1)^{-a}$ and $\gamma_t = \gamma_0 (t+1)^{-b}$ for the outer update, with $0.5 < a < 1$, $0.5 < b < 1$, and $2b - a > 1$ to ensure $\sum_t \gamma_t^2 / \rho_t < \infty$. In all runs we choose $a = 0.8$, $b \in [0.9, 0.95]$, $\gamma_0 \in [0.2, 0.8]$. At each iteration, we linearize the (scalarized) objective and constraints around the current iterate and add a quadratic proximal term with coefficient $\tau \in [0.05, 0.1]$ to obtain a strongly convex quadratic (conic) subproblem. The outer iteration budget is $T = 3\,000$. After training, we report $\mathbb{E}[f_i]$ and $\mathbb{E}[g_j]$ using a large Monte Carlo held-out evaluation (typically $2 \times 10^5$ samples).

**Ideal point in Tchebycheff scalarization.** We consider standard Tchebycheff scalarization $\min_x \max_i \{\lambda_i (\mathbb{E} f_i(x) - z_i)\}$ with $\boldsymbol{\lambda} \in \Delta_2$. Different choices of the ideal point $\boldsymbol{z}$ lead to different scalarized problems. We support: (i) *analytic* $z_i = \sigma^2$ (unconstrained ideal point); (ii) *feasible-grid*, which first approximates the feasible set via a fine grid on $\mathcal{X}$ and then sets $z_i = \min_{x \in \mathcal{F}} f_i(x)$; and (iii) *fixed*, a user-specified $\boldsymbol{z}$. When comparing against closed-form baselines, we use feasible-grid so that the scalarization matches the analytic definition.

**Results and Analysis.** Across all tested preferences $\boldsymbol{\lambda} = (\lambda_1, 1 - \lambda_1)$, the three variants (**basic**, **CSMOO-1**, **CSMOO-2**) exhibit fast and stable convergence: the smoothed Tchebycheff objective decreases monotonically and plateaus, while the maximal constraint violation is driven to (nearly) zero and remains there throughout training. When the per-iteration quadratic surrogate is feasible from

early iterations, **basic** and **CSMOO-1** produce essentially identical trajectories and final solutions; the relaxation in CSMOO-1 is mainly beneficial when transient infeasibility arises, after which it rejoins the same solution path. **CSMOO-2** yields the smoothest histories, as its always-feasible reformulation (with $(y_0, y_1)$) eliminates stalls due to infeasible subproblems and reduces sensitivity to early estimation noise.

The choice of the ideal point $\mathbf{z}$ behaves as expected. With the *feasible-grid* protocol, where $z_i = \min_{x \in \mathcal{F}} f_i(x)$ estimated on a dense feasible grid, the scalarized problems align with the constrained landscape: endpoint Tchebycheff values concentrate near zero and the final points match the grid-based reference to within small Monte Carlo discrepancies. With an *analytic* (unconstrained) ideal point, the scalarization becomes intentionally optimistic; final Tchebycheff values can turn negative even at high-quality solutions. This does not indicate inconsistency—it simply reflects that $\mathbf{z}$ was defined without feasibility in mind, and the relative ordering of methods is unaffected.

The estimated Pareto front (PF), obtained by sweeping $\lambda_1$ on a uniform grid and evaluating $(\mathbb{E}[f_1(x)], \mathbb{E}[f_2(x)])$ at convergence, is continuous and well behaved when a *single* problem instance (seed) and a *fixed* ideal point $\mathbf{z}$ are used across the sweep, and points are connected in the *order of* $\lambda_1$ (without resorting by any objective). Residual jitter vanishes with larger held-out evaluation budgets; in our setting, increasing the evaluation sample size suffices to render the PF visually smooth and reproducible.

Method-wise, all three approaches converge to nearly identical final points whenever the surrogate remains feasible; nonetheless, **CSMOO-2** is preferable in practice because it (i) avoids feasibility checks altogether, (ii) shows the most regular (nearly monotone) decrease of the Tchebycheff objective and the fastest reduction of the violation, and (iii) is less sensitive to early-step stochasticity. Ablations on the EMA/outer-step schedules confirm the theoretical guideline $2b - a > 1$: overly aggressive $\gamma_t$ (large $\gamma_0$ or small $b$) induces oscillations and delays feasibility, whereas too conservative schedules slow progress. The proximal strength $\tau$ is critical for the curvature of the surrogate: moderate values ensure strong convexity and stable descent; values that are too small weaken curvature and can lead to erratic steps. Finally, we find that using small mini-batches to form value/gradient estimates and a large, held-out evaluation set reduces variance in both the convergence histories and the final reported metrics, without altering the qualitative conclusion that CSMOO-2 is the most robust and reliable of the three variants.

The additional experiment results of the synthetic problem are Figure 6 to Figure 19.

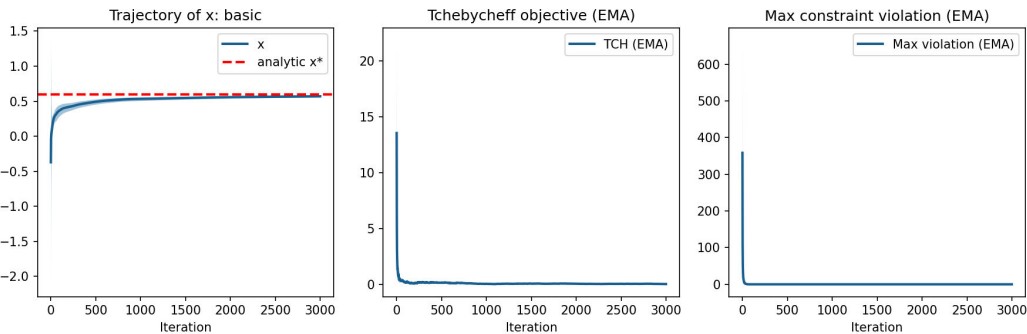

Figure 6: The convergence curve of $x$, Tchebycheff objective and Max constraint violation when $\lambda = [0.1, 0.9]$ using basic method.

### E.2 ADDITIONAL EXPERIMENT DETAILS AND RESULTS OF THE PHYSICAL LAYER SECURITY COMMUNICATION PROBLEM

**Problem Definition.** Consider a downlink secure communication system that comprises one base station (BS) equipped with $n_t$ antennas, $m$ single-antenna legitimate users, and one single-antenna eavesdropper. The eavesdropper may attempt to deliberately eavesdrop on the signals transmitted to the legitimate users. Let $\mathbf{h}_i \in \mathbb{C}^{n_t \times 1}$ be the channel state information (CSI) between the BS and user $i$ while $\mathbf{h}_e \in \mathbb{C}^{n_t \times 1}$ be the CSI between the BS and the eavesdropper. However, the BS only communicates with legitimate users and reduces the signal leakage based on the estimated CSI $\hat{\mathbf{h}}_i$

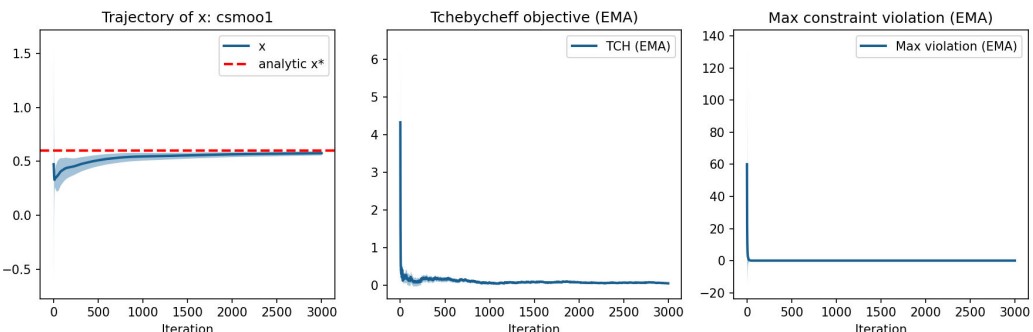

Figure 7: The convergence curve of $x$, Tchebycheff objective and Max constraint violation when $\lambda = [0.1, 0.9]$ using CSMOO-1 method.

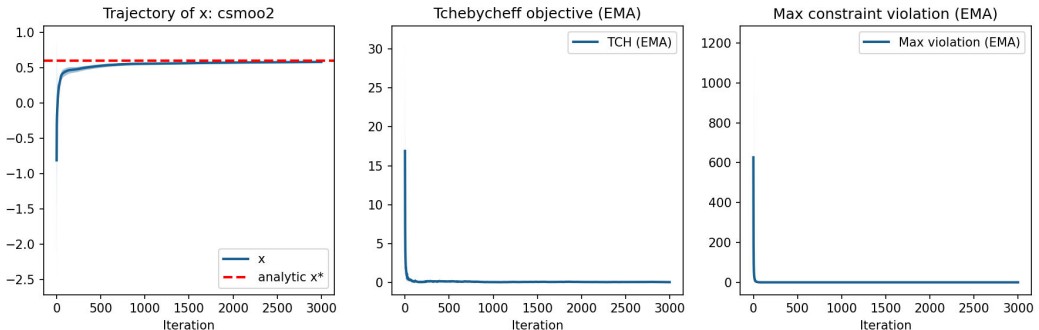

Figure 8: The convergence curve of $x$, Tchebycheff objective and Max constraint violation when $\lambda = [0.1, 0.9]$ using CSMOO-2 method.

and $\hat{\mathbf{h}}_e$. The true CSI can be modeled as $\mathbf{h}_i = \hat{\mathbf{h}}_i + \mathbf{e}_i$ and $\mathbf{h}_e = \hat{\mathbf{h}}_e + \mathbf{e}_e$, where $\mathbf{e}_i$ and $\mathbf{e}_e$ represents the channel estimation error. Thus, the expected rate at user $i$ is given by

$$\bar{R}_i(\mathbf{W}) = \mathbb{E}_{\mathbf{e}_i}\left[\log_2\left(1 + \frac{|\mathbf{h}_i^H \mathbf{w}_i|^2}{\sigma_i^2 + \sum_{l \neq i}|\mathbf{h}_i^H \mathbf{w}_l|^2}\right)\right], \quad i = 1, 2, \cdots, m, \tag{80}$$

where $\sigma_i^2$ is the additive white Gaussian noise (AWGN) at legitimate user $i$; $\mathbf{w}_i$ denotes the beamforming vector intended for user $i$. By stacking all beamforming vectors together, we can define $\mathbf{W} = [\mathbf{w}_1, \cdots, \mathbf{w}_m]$.

Then, for security provisioning, we make a worst-case assumption regarding the capabilities of the potential eavesdropper. In particular, we assume that the potential eavesdroppers have unlimited computational resources and therefore are able to cancel all multiuser interference before decoding the information transmitted to a given legitimate user. Therefore, the expected rate between the BS and potential eavesdropper for decoding the signal of legitimate user $i$ is given by

$$\bar{C}_i(\mathbf{W}) = \mathbb{E}_{\mathbf{h}_e}\left[\log_2\left(1 + \frac{1}{\sigma_e^2}|\mathbf{h}_e^H \mathbf{w}_i|^2\right)\right], \quad i = 1, 2, \cdots, m, \tag{81}$$

where $\sigma_i^2$ is the additive white Gaussian noise (AWGN) at legitimate user $i$.

Our goal is to maximize the system rate while keeping the maximum information leakage to the eavesdroppers below a desired level. For any given error, the optimization problem is formulated as:

$$\begin{aligned} \underset{\mathbf{W} \in \{\mathbf{W}: \|\mathbf{W}\|^2 \leq P_{\max}\}}{\text{maximize}} \quad & \bar{\mathbf{R}}(\mathbf{W}) = (\bar{R}_1(\mathbf{W}), \cdots, \bar{R}_m(\mathbf{W})) \\ \text{subject to} \quad & \bar{C}_i(\mathbf{W}) = \mathbb{E}_{\mathbf{h}_e}\left[\log_2\left(1 + \frac{1}{\sigma_e^2}|\mathbf{h}_e^H \mathbf{w}_i|^2\right)\right] \leq C_{\text{th}}, \quad i = 1, \cdots, m, \end{aligned} \tag{82}$$

where $C_{\text{th}}$ is a predefined parameter which limits the maximum tolerable information leakage to the potential eavesdropper for wiretapping the signal transmitted to each legitimate user.

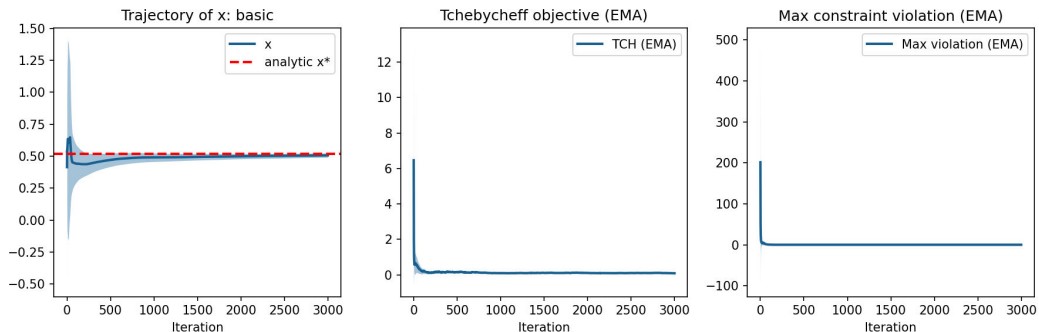

Figure 9: The convergence curve of $x$, Tchebycheff objective and Max constraint violation when $\lambda = [0.3, 0.7]$ using basic method.

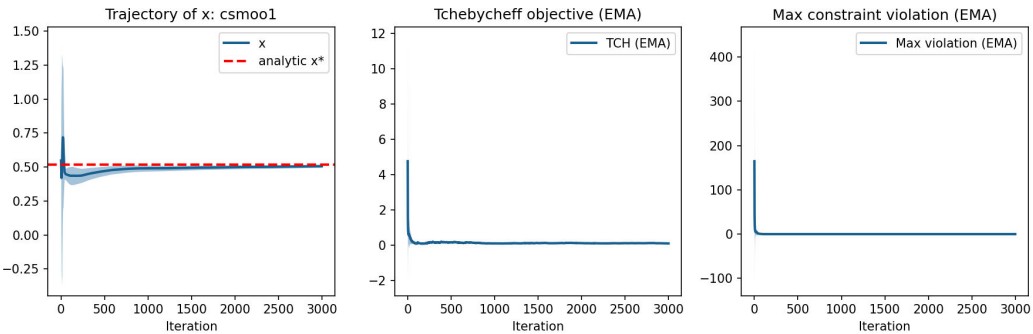

Figure 10: The convergence curve of $x$, Tchebycheff objective and Max constraint violation when $\lambda = [0.3, 0.7]$ using CSMOO-1 method.

**Experimental setup.** We instantiate a standard downlink physical-layer security (PLS) testbed with $n_t = 2$ transmit antennas and $m = 2$ single-antenna users. The additive noise powers are set to $\sigma_i^2 = \sigma_e^2 = 1$, and the total power budget is constrained by $\|\mathbf{W}\|_F^2 \leq P_{\max}$ with $P_{\max} = 5$. Imperfect channel state information is modeled by proper complex Gaussian estimation errors $\mathbf{e}_i \sim \mathcal{CN}(\mathbf{0}, \delta_u^2\mathbf{I})$, $\mathbf{e}_e \sim \mathcal{CN}(\mathbf{0}, \delta_e^2\mathbf{I})$ with $\delta_u = \delta_e = 0.05$; the worst-case eavesdropper is assumed to be able to cancel multiuser interference. The information leakage constraints are enforced via $\bar{C}_i(\mathbf{W}) \leq C_{\text{th}}$ with $C_{\text{th}} = 0.4$. For preference-based multi-objective optimization, we use Tchebycheff scalarization with a preference vector $\boldsymbol{\lambda} = (\lambda_1, 1 - \lambda_1)$, and we sweep $\lambda_1$ over a grid in $[0.1, 0.9]$ to complete the experiment.

To stabilize the stochastic optimization, we adopt the single-sample exponential moving average (EMA) estimators in Section 4: the momentum sequence is $\rho_t = (t+1)^{-a}$ with $a = 0.8$, and the outer step sizes are $\gamma_t = \gamma_0(t+1)^{-b}$ with $\gamma_0 \in [0.3, 0.5]$ and $b = 0.95$, which satisfy the summability conditions in Assumptions 2 and 3. Each iteration uses a small mini-batch (e.g., 2-4 samples) to reduce gradient variance. In the per-iteration surrogate, we use a quadratic proximal regularizer with coefficient $\tau \in [0.05, 0.06]$ to ensure strong convexity and numerical stability, and a penalty weight $\beta$ (Section 4) to control feasibility; we set $\beta$ large enough so that the maximal violation is driven to (nearly) zero throughout training. For evaluation, we report $\mathbb{E}[\bar{R}_i]$ and $\mathbb{E}[\bar{C}_i]$ via a large Monte Carlo held-out set (typically $1.5 \times 10^5$ samples).

**Ideal-point selection.** To make Tchebycheff values comparable across preferences, we fix the ideal point $\mathbf{z}$ over the entire Pareto sweep. Concretely, we estimate $\mathbf{z}$ once on the same problem instance using two extreme preferences $\boldsymbol{\lambda} \in \{(0.99, 0.01), (0.01, 0.99)\}$ and set $z_1$ (resp. $z_2$) to the best observed value of $f_1$ (resp. $f_2$). This "fixed-$\mathbf{z}$" protocol is standard and prevents preference-dependent drift in the scalarized objective.

**Results and Analysis.** We evaluate the proposed CSMOO-2 against the first-order baseline described in the main text, using the same preference sweep, EMA schedules, proximal regularization,

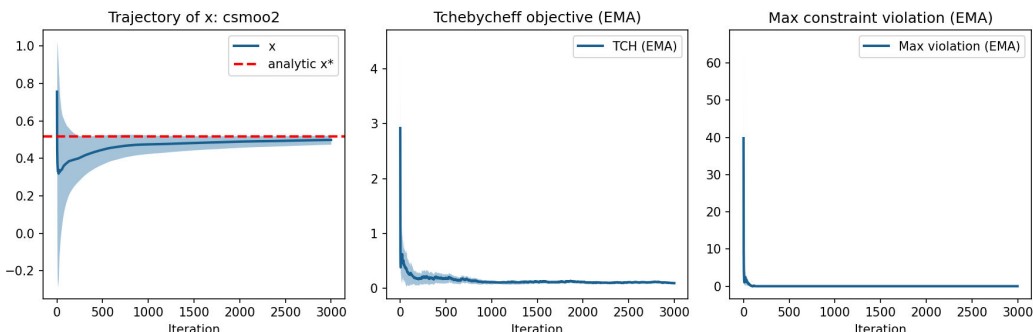

Figure 11: The convergence curve of $x$, Tchebycheff objective and Max constraint violation when $\lambda = [0.3, 0.7]$ using CSMOO-2 method.

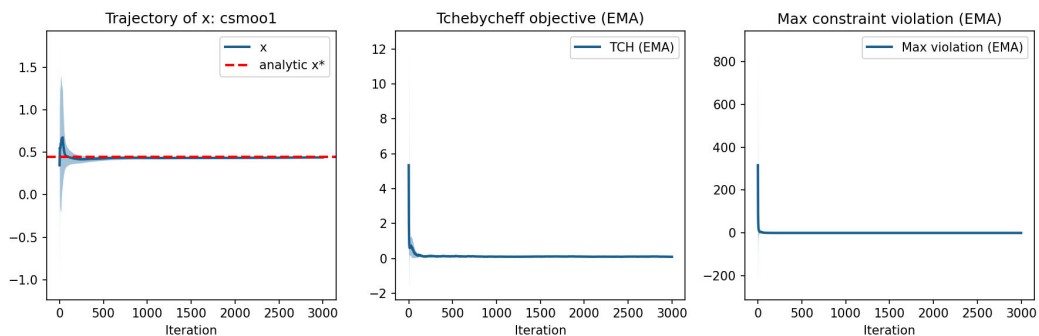

Figure 12: The convergence curve of $x$, Tchebycheff objective and Max constraint violation when $\lambda = [0.5, 0.5]$ using CSMOO-1 method.

and held-out evaluation protocol. Unless otherwise specified, all results are reported as the mean and standard deviation across independent runs with fixed power-budget projection and a common ideal point $\mathbf{z}$ shared over preferences.

For all tested preferences $\boldsymbol{\lambda} = (\lambda_1, 1 - \lambda_1)$, both methods rapidly drive the maximal constraint violation to (nearly) zero and maintain feasibility thereafter. The Tchebycheff objective (with fixed $\mathbf{z}$ and log-sum-exp smoothing) decreases monotonically and stabilizes. Negative final Tchebycheff values occasionally occur because the fixed ideal point, estimated at extreme preferences, can be slightly optimistic; this does not impact Pareto optimality or the comparison between methods.

When the preference strongly emphasizes one user, two effects become more pronounced. First, the expected rate of the de-emphasized user exhibits visibly larger across-run variability. This is inherent: (i) that objective receives a much smaller weight in the scalarization, leading to weaker effective gradients; and (ii) random channel realizations induce heterogeneous headroom on the de-emphasized link once feasibility is enforced. Second, transient "kink-like" patterns may appear in convergence histories when the iterate is projected to the power budget early and then proceeds with small corrective steps under a strong penalty/regularization; this is a benign artifact of the projection-plus-proximal dynamics rather than a sign of instability. Importantly, even in these extreme settings, CSMOO-2 achieves lower (better) Tchebycheff values and higher expected rates for the emphasized user than the baseline, while keeping the maximal leakage violation near zero.

Larger error bars at $\lambda_1 \in \{0.1, 0.9\}$ primarily reflect the de-emphasized user's sensitivity to stochastic channels and to the evaluation protocol (e.g., whether runs are aligned by the first time the violation falls below a threshold). Using a common problem instance and evaluation RNG across preferences, increasing the mini-batch size during training, or disabling alignment in plotting all reduce this effect. These adjustments do not change the qualitative ordering: CSMOO-2 remains consistently better than the baseline in the final metrics under the same feasibility target.

The additional experiment results of the communication problem are Figure 20 to Figure 27.

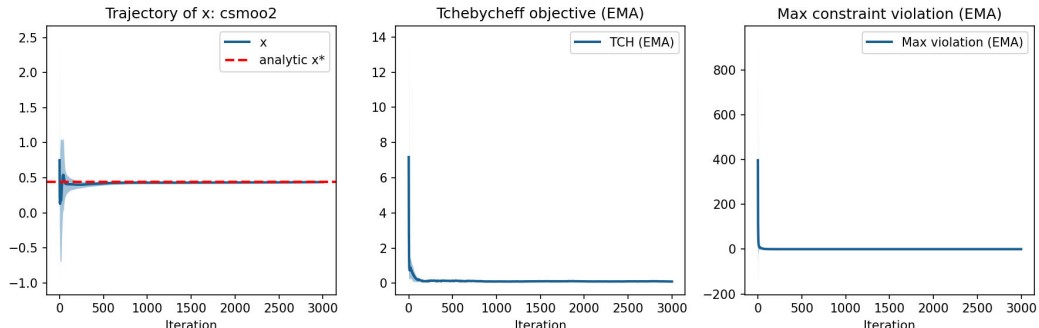

Figure 13: The convergence curve of $x$, Tchebycheff objective and Max constraint violation when $\lambda = [0.5, 0.5]$ using CSMOO-2 method.

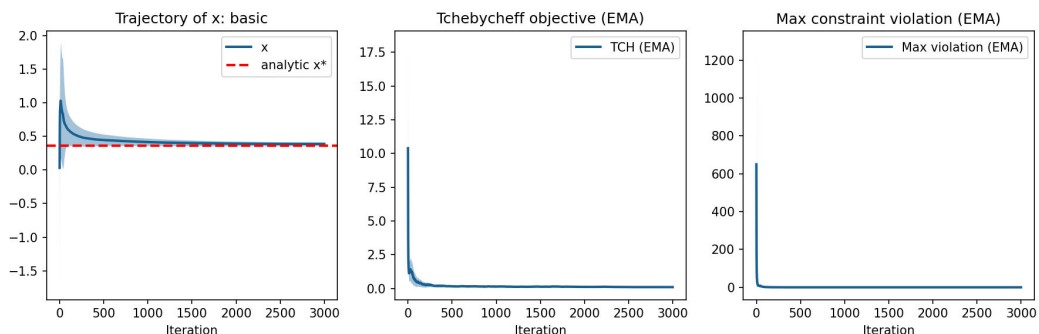

Figure 14: The convergence curve of $x$, Tchebycheff objective and Max constraint violation when $\lambda = [0.7, 0.3]$ using basic method.

### E.3 COMPARATIVE EXPERIMENTS WITH DIFFERENT $\beta$

To empirically substantiate the role of the penalty parameter $\beta$ in CSMOO-2, we conduct an ablation study on the synthetic problem from Section 5.1. Recall that, under the equivalent penalty reformulation, $\beta$ balances progress on the scalarized objective against the auxiliary slack $y_1$ that absorbs constraint violations. Although the theory guarantees existence of a finite $\beta^\star$ beyond which the reformulation matches the original Tchebycheff problem, practitioners often ask (i) how sensitive convergence is to $\beta$, and (ii) whether a nonzero penalty noticeably accelerates feasibility restoration compared with the degenerate case $\beta = 0$.

**Experiment Details.** We sweep $\beta \in \{0, 0.1, 0.5, 2, 5, 10\}$ and run CSMOO-2 for 3,000 iterations per value, keeping all other hyperparameters identical to the main setup (single-sample EMA with $a = 0.8$, outer step sizes with $b = 0.95$ and $\gamma_0 = 0.4$, proximal strength $\tau = 0.08$, and preference vector $\lambda = [0.5, 0.5]$). Each run starts from the same decision point and stochastic seed to isolate the impact of $\beta$. We record the final Tchebycheff value obtained after the outer iteration budget with feasible ratio. These statistics are then averaged over five repeated runs to smooth out sampling noise.

**Results.** Figure 33 reports the aggregate metrics. The left panel shows that setting $\beta = 0$ yields the highest final Tchebycheff value (worst performance), whereas $\beta \geq 0.5$ quickly plateaus near the best achievable value ($\approx 9.45 \times 10^{-2}$). The feasible ratio—the proportion of runs whose last iterate satisfies $\max_j g_j(x) \leq 10^{-4}$—is unity across all $\beta$ values in this simple problem, highlighting that even a weak penalty suffices to restore feasibility thanks to the benign geometry of the quartic constraints. Nevertheless, figure 33) reveals an important nuance: when $\beta = 0$, the constraint violation decays more slowly during the first 50–80 iterations, whereas $\beta \geq 0.5$ drives the violation to numerical zero almost immediately. This behavior corroborates the theoretical guidance in Section 3: a positive penalty reinforces the descent direction on the slack variable $y_1$, preventing the algorithm from lingering near infeasible surrogate solutions. In contrast, $\beta = 0$ effectively removes that

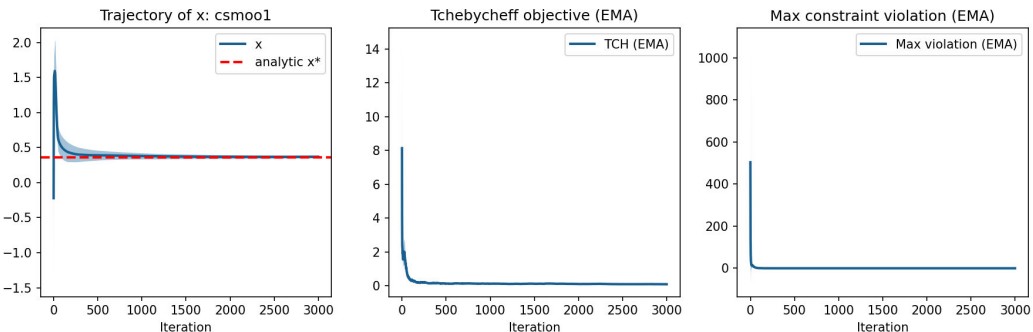

Figure 15: The convergence curve of $x$, Tchebycheff objective and Max constraint violation when $\lambda = [0.7, 0.3]$ using CSMOO-1 method.

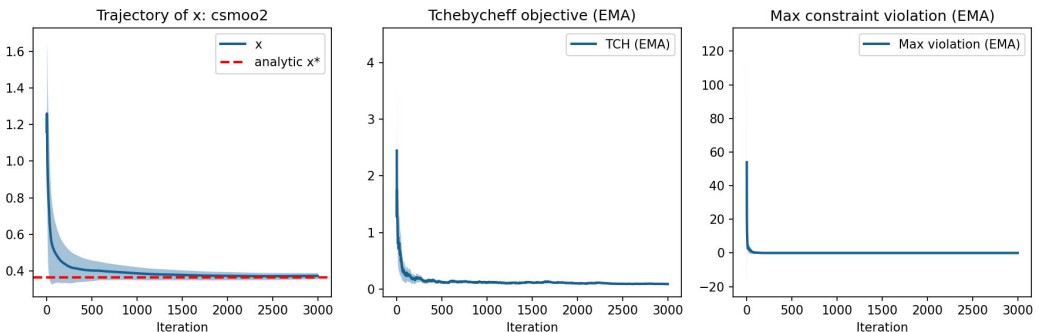

Figure 16: The convergence curve of $x$, Tchebycheff objective and Max constraint violation when $\lambda = [0.7, 0.3]$ using CSMOO-2 method.

corrective force, so the algorithm relies solely on the proximal regularization to satisfy the constraints, resulting in a longer transient.

Taken together, the ablation suggests that $\beta$ acts primarily as an accelerator for feasibility restoration rather than a determinant of the ultimate solution quality in this example. Moderate choices ($\beta \in [2, 10]$) strike a pragmatic balance: they maintain the best Tchebycheff value, keep the violation curve sharply decreasing, and incur no discernible numerical instability. This observation matches the design intuition of CSMOO-2, namely that the penalty term should be large enough to discourage persistent constraint violations but not so large as to overshadow progress on the scalarized objective.

### E.4 EXPERIMENT DETAILS AND RESULTS ON THREE-OBJECTIVE SYNTHETIC PROBLEM

**Problem Setup and Experiment Details.** To further stress-test the proposed framework on a higher dimensional objective space, we conduct the experiment based on the synthetic problem from Section 5.1 with an added third quadratic objective

$$f_3(x) = \mathbb{E}_{\epsilon \sim \mathcal{N}(0, 0.25)}\big((x + \epsilon + 0.5)^2\big), \tag{83}$$

while keeping the quartic constraint family $(g_1, g_2)$ unchanged. The decision space therefore remains one-dimensional ($x \in [-5, 5]$), but the Pareto set now lies in $\mathbb{R}^3$. We sweep preference vectors over the probability simplex by sampling $\boldsymbol{\lambda} \in \Delta_3$ on a uniform grid (`simplex_steps`=20 in the released code). For each preference, we run **basic**, **CSMOO-1**, and **CSMOO-2** for 3,000 iterations with the same EMA schedule ($a = 0.8, b = 0.95, \gamma_0 = 0.4$) and proximal coefficient $\tau = 0.08$ as in the two-objective experiment. After convergence, we evaluate $\big(\mathbb{E}[f_1], \mathbb{E}[f_2], \mathbb{E}[f_3]\big)$ using $10\,000$ Monte Carlo samples per preference.

**Results.** Despite the additional objective, the one-dimensional nature of the feasible set implies that the Pareto frontier is still a smooth curve embedded in $\mathbb{R}^3$. Figure 38 visualizes the estimated frontier obtained by connecting the evaluated points according to the sweep order of $\boldsymbol{\lambda}$. All three methods

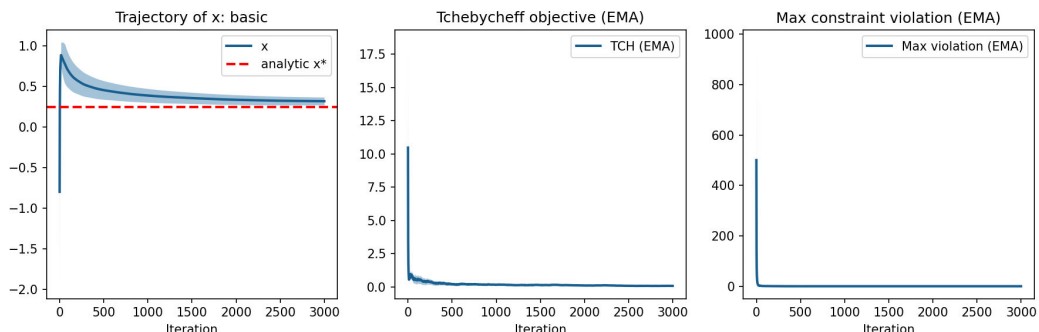

Figure 17: The convergence curve of $x$, Tchebycheff objective and Max constraint violation when $\lambda = [0.9, 0.1]$ using basic method.

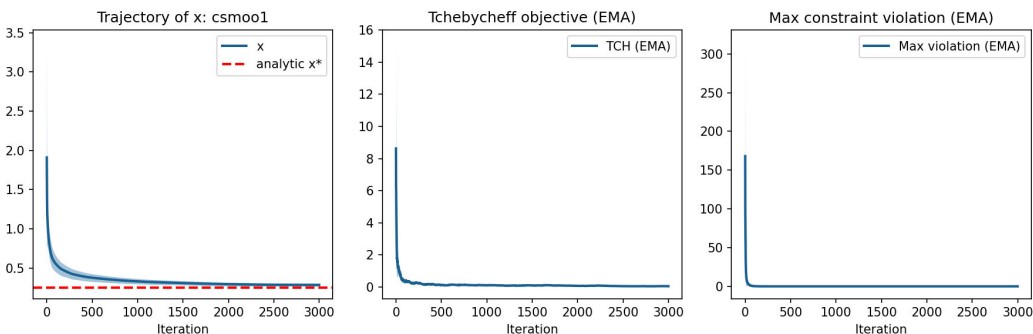

Figure 18: The convergence curve of $x$, Tchebycheff objective and Max constraint violation when $\lambda = [0.9, 0.1]$ using CSMOO-1 method.

overlap almost perfectly, confirming that the per-iteration quadratic surrogates stay feasible even in the higher dimensional objective space. The curve bends toward higher $E[f_3]$ whenever the third preference weight dominates, while sacrificing $E[f_1]$ and $E[f_2]$ in a predictable manner. Importantly, the frontier contains no gaps or self-intersections, which verifies that the preference sweep with a fixed stochastic instance and ideal point delivers a coherent three-objective trade-off. This stresses that the proposed algorithms can seamlessly scale to additional objectives without any algorithmic modification: the EMA estimators remain asymptotically consistent, the quadratic surrogate stays strongly convex, and CSMOO-2 continues to enforce feasibility through its penalty-reformulation.

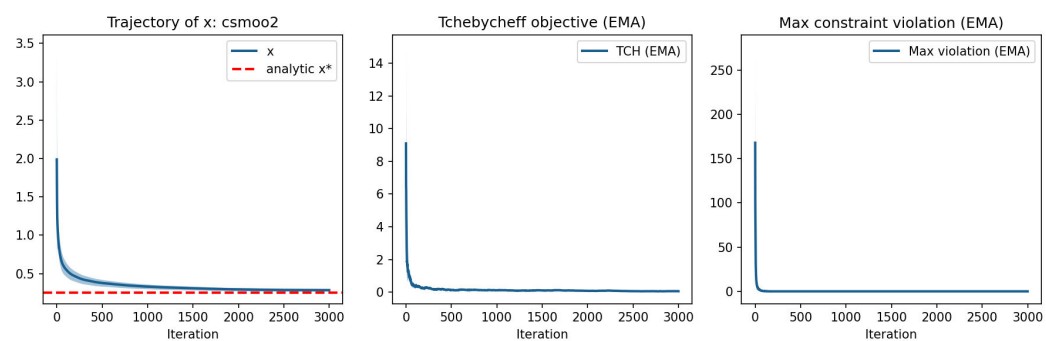

Figure 19: The convergence curve of $x$, Tchebycheff objective and Max constraint violation when $\lambda = [0.9, 0.1]$ using CSMOO-2 method.

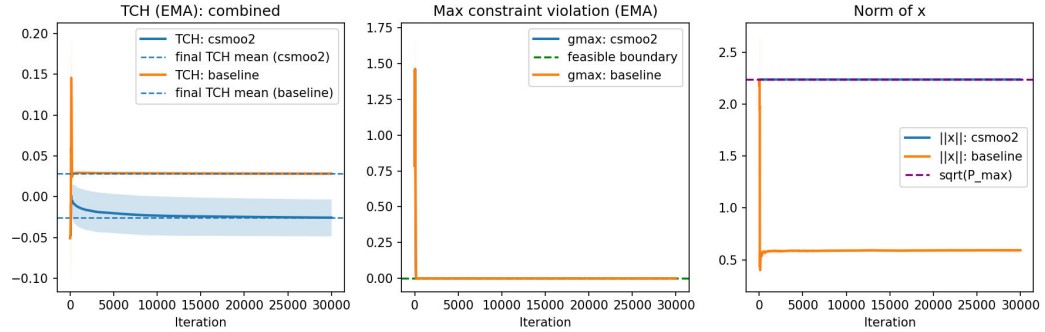

Figure 20: The convergence of Tchebycheff Objective, Max Constraint Violation, and the Transmit Power(norm of $\mathbf{W}$) of PLS beamforming experiment: CSMOO-2 against baseline when $\lambda = [0.1, 0.9]$.

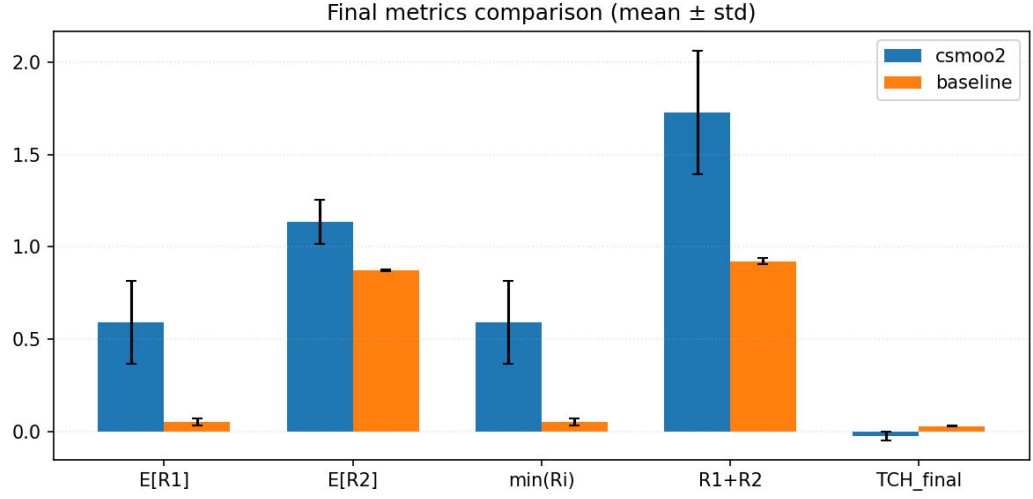

Figure 21: The expected rate metrics comparison of PLS beamforming experiment: CSMOO-2 against baseline when $\lambda = [0.1, 0.1]$.

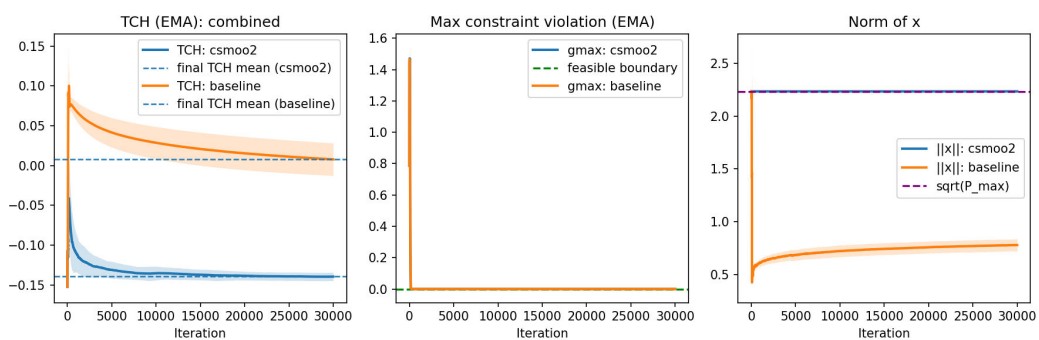

Figure 22: The convergence of Tchebycheff Objective, Max Constraint Violation, and the Transmit Power(norm of $\mathbf{W}$) of PLS beamforming experiment: CSMOO-2 against baseline when $\lambda = [0.3, 0.7]$.

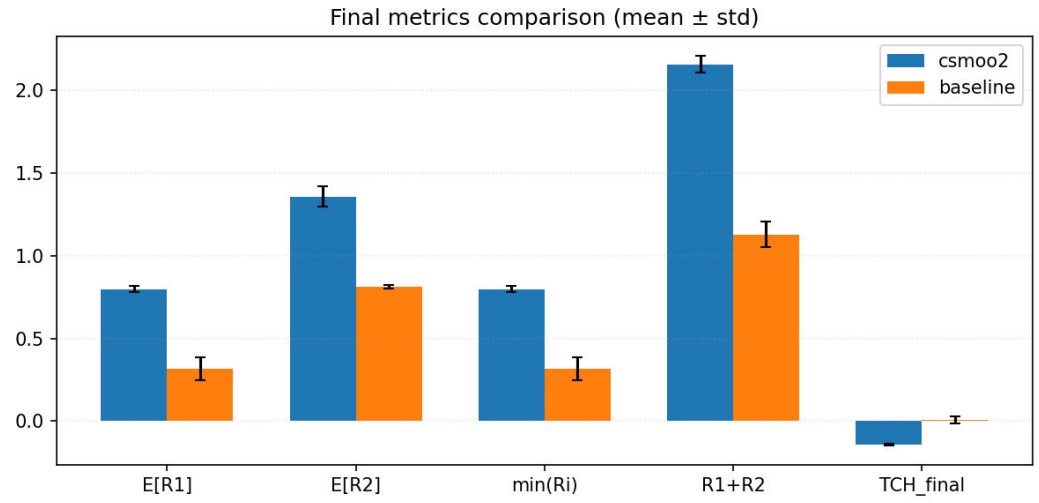

Figure 23: The expected rate metrics comparison of PLS beamforming experiment: CSMOO-2 against baseline when $\lambda = [0.3, 0.7]$.

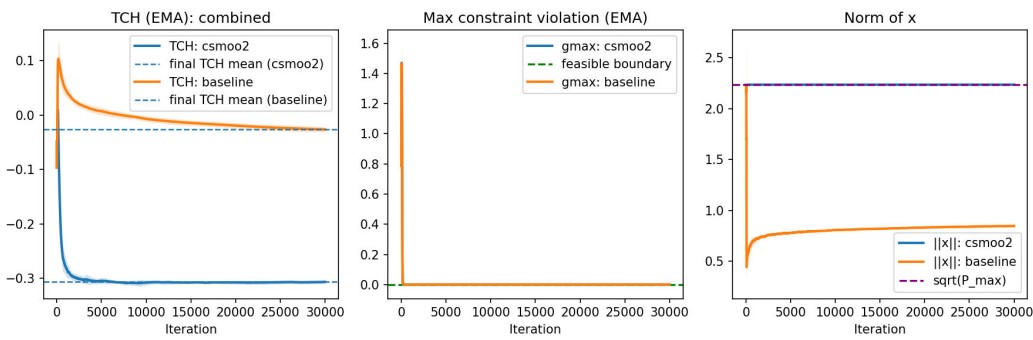

Figure 24: The convergence of Tchebycheff Objective, Max Constraint Violation, and the Transmit Power(norm of $\mathbf{W}$) of PLS beamforming experiment: CSMOO-2 against baseline when $\lambda = [0.7, 0.3]$.

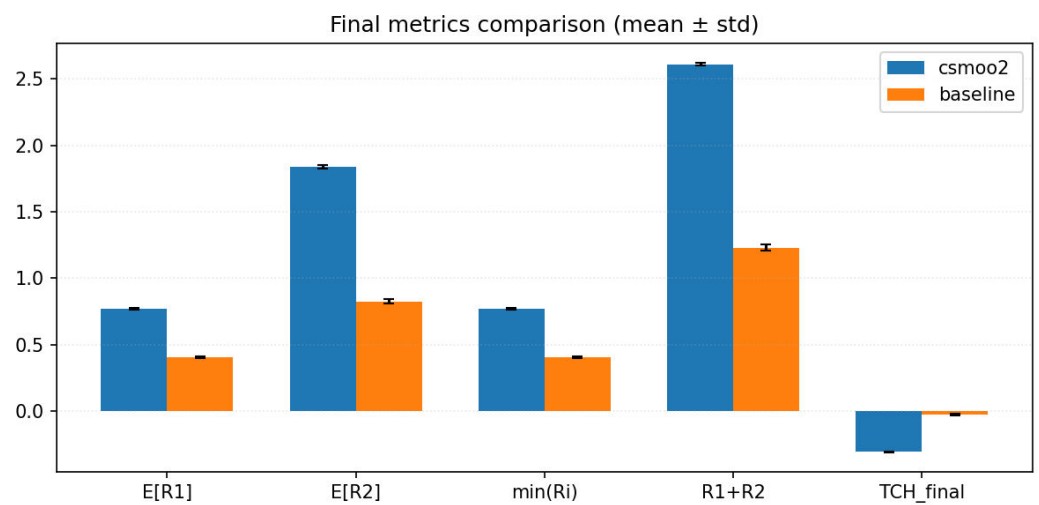

Figure 25: The expected rate metrics comparison of PLS beamforming experiment: CSMOO-2 against baseline when $\lambda = [0.7, 0.3]$.

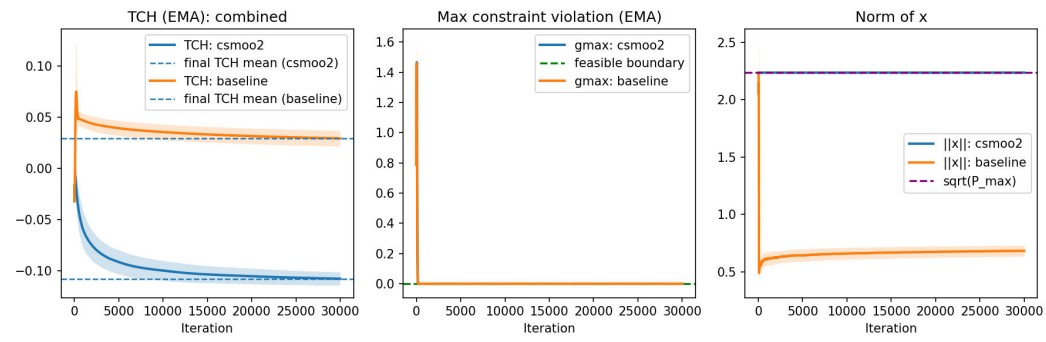

Figure 26: The convergence of Tchebycheff Objective, Max Constraint Violation, and the Transmit Power(norm of $\mathbf{W}$) of PLS beamforming experiment: CSMOO-2 against baseline when $\lambda = [0.9, 0.1]$.

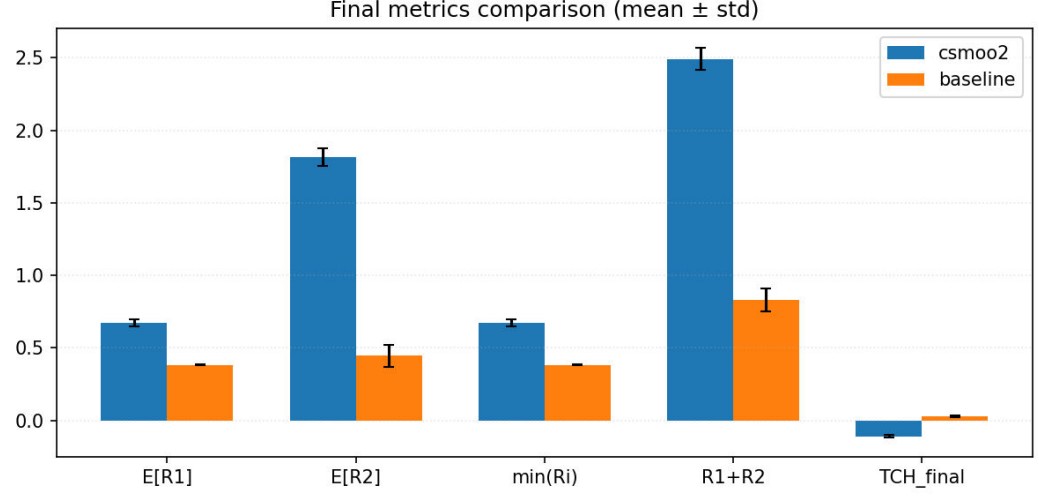

Figure 27: The expected rate metrics comparison of PLS beamforming experiment: CSMOO-2 against baseline when $\lambda = [0.9, 0.1]$.

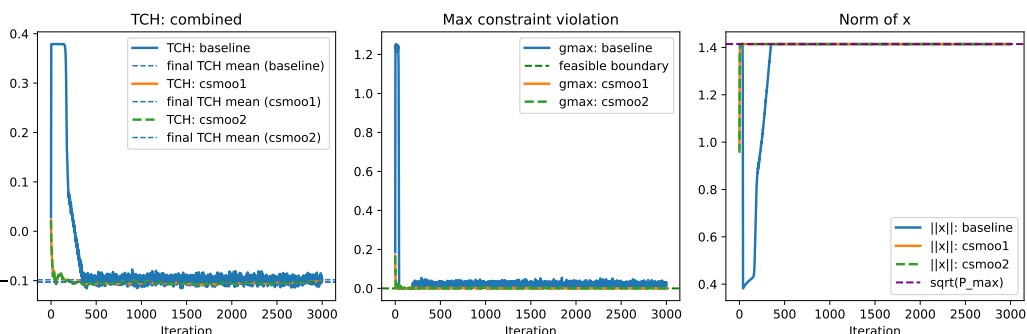

Figure 28: The convergence of Tchebycheff Objective, Max Constraint Violation, and the Transmit Power(norm of $\mathbf{W}$) of PLS beamforming experiment: CSMOO-1/2 against baseline when $\lambda = [0.5, 0.5]$.

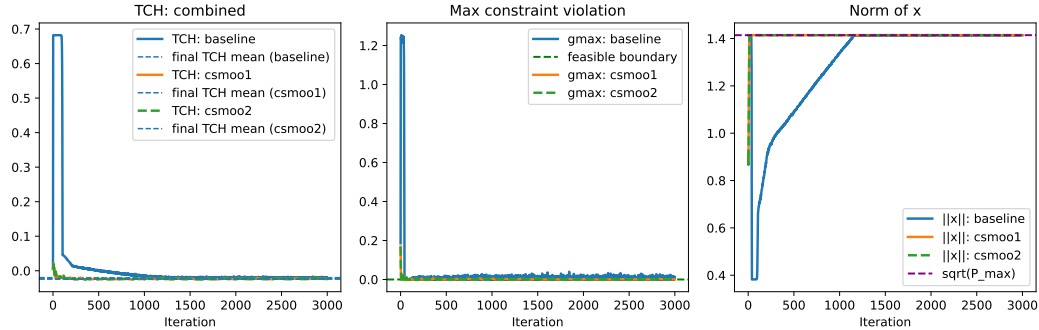

Figure 29: The convergence of Tchebycheff Objective, Max Constraint Violation, and the Transmit Power(norm of $\mathbf{W}$) of PLS beamforming experiment: CSMOO-1/2 against baseline when $\lambda = [0.1, 0.9]$.

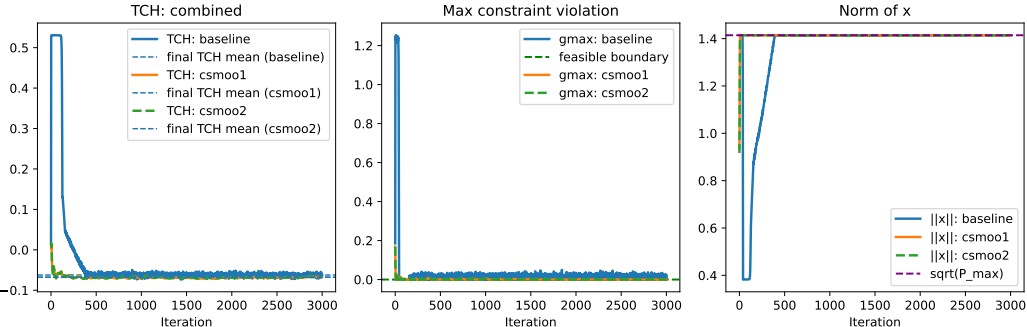

Figure 30: The convergence of Tchebycheff Objective, Max Constraint Violation, and the Transmit Power(norm of $\mathbf{W}$) of PLS beamforming experiment: CSMOO-1/2 against baseline when $\lambda = [0.3, 0.7]$.

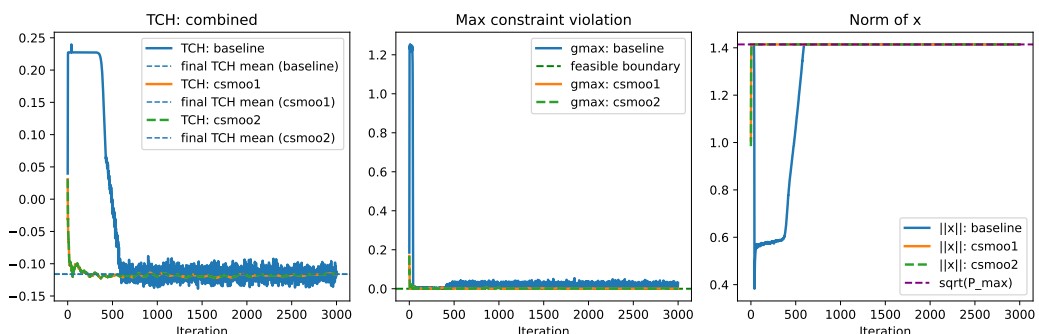

Figure 31: The convergence of Tchebycheff Objective, Max Constraint Violation, and the Transmit Power(norm of $\mathbf{W}$) of PLS beamforming experiment: CSMOO-1/2 against baseline when $\lambda = [0.7, 0.3]$.

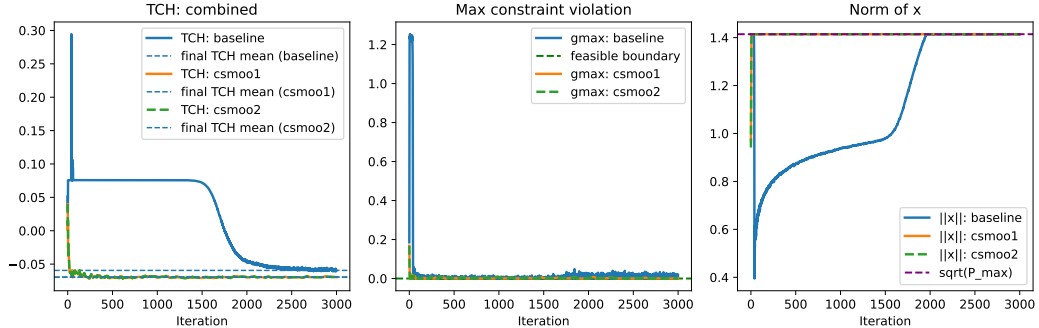

Figure 32: The convergence of Tchebycheff Objective, Max Constraint Violation, and the Transmit Power(norm of $\mathbf{W}$) of PLS beamforming experiment: CSMOO-1/2 against baseline when $\lambda = [0.9, 0.1]$.

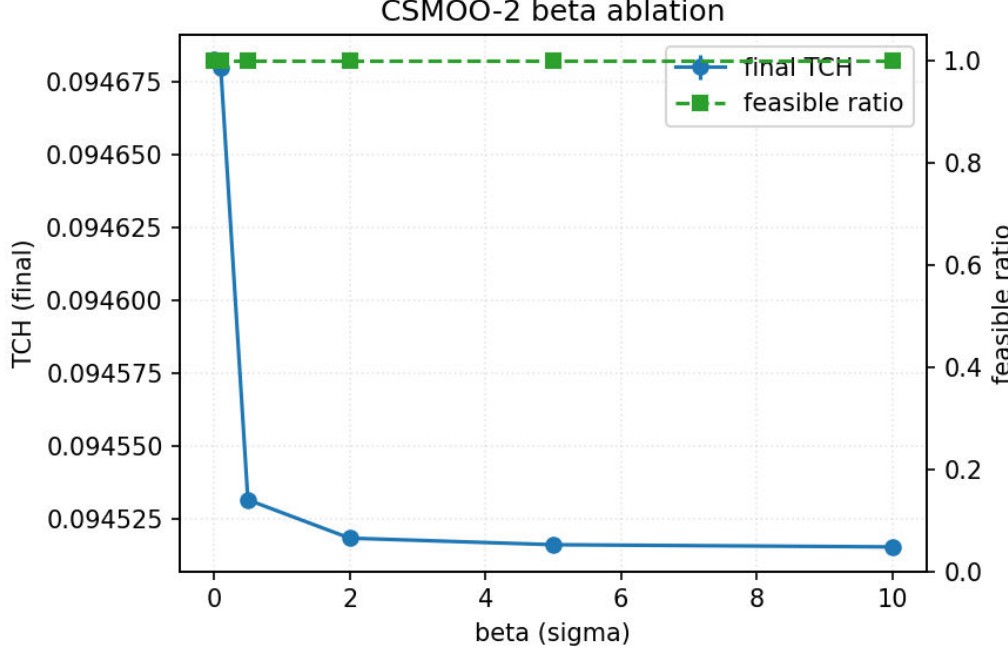

Figure 33: Final Tchebycheff value and feasible ratio of CSMOO-2 under different $\beta$. Larger penalties reduce the stationary objective value and keep feasibility intact.

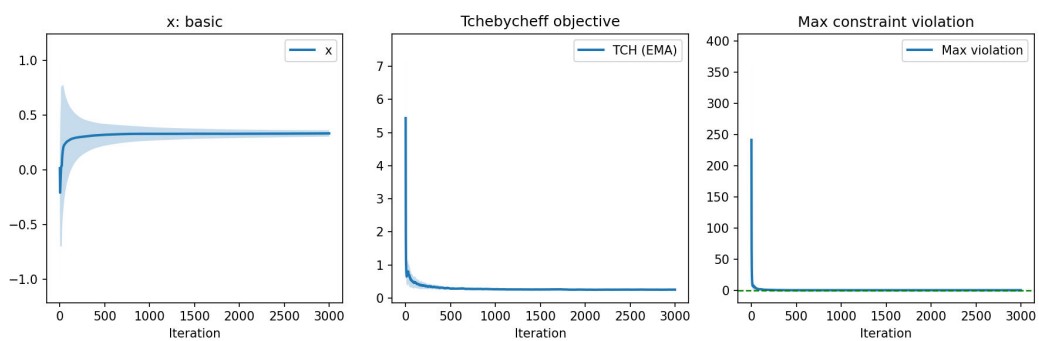

Figure 34: The convergence curve of $x$, Tchebycheff objective and Max constraint violation when $\lambda = [0.4, 0.35, 0.25]$ using basic method in three-objective synthetic problems.

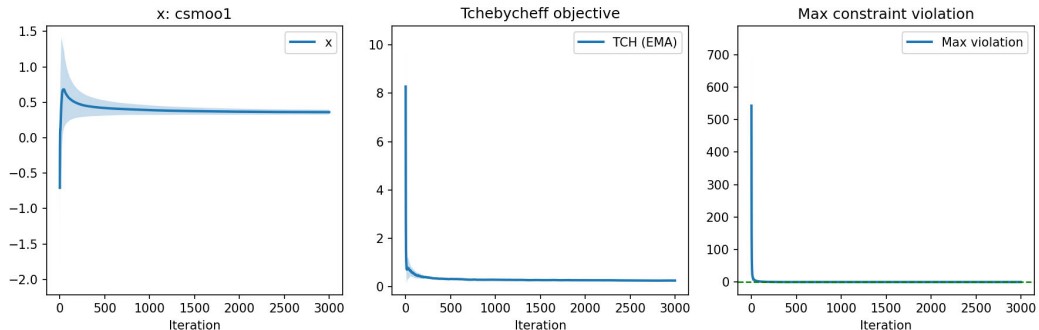

Figure 35: The convergence curve of $x$, Tchebycheff objective and Max constraint violation when $\lambda = [0.4, 0.35, 0.25]$ using CSMOO-1 method in three-objective synthetic problems.

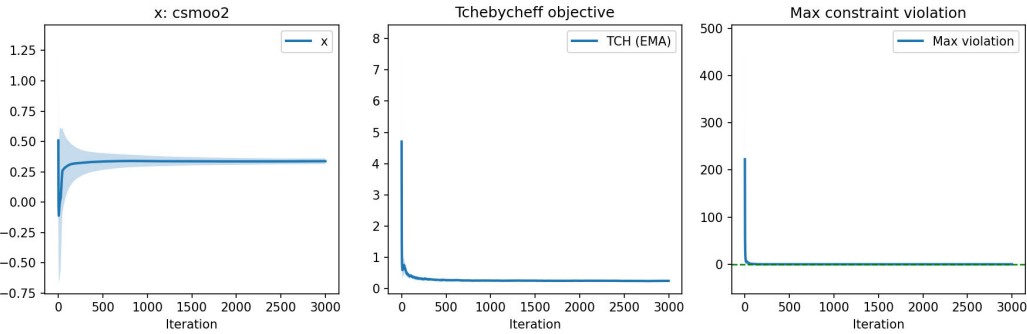

Figure 36: The convergence curve of $x$, Tchebycheff objective and Max constraint violation when $\lambda = [0.4, 0.35, 0.25]$ using CSMOO-2 method in three-objective synthetic problems.

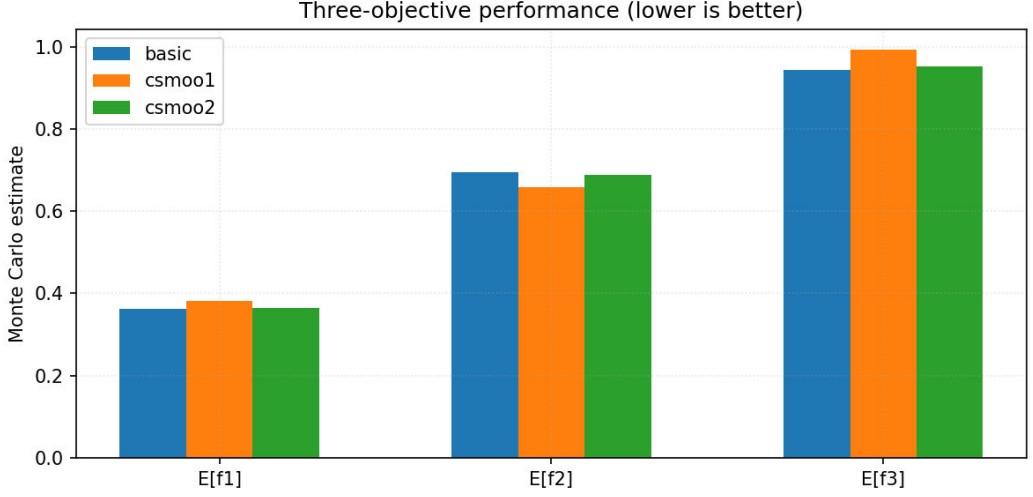

Figure 37: The performance when $\lambda = [0.4, 0.35, 0.25]$ using three methods in three-objective synthetic problems.

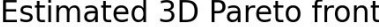
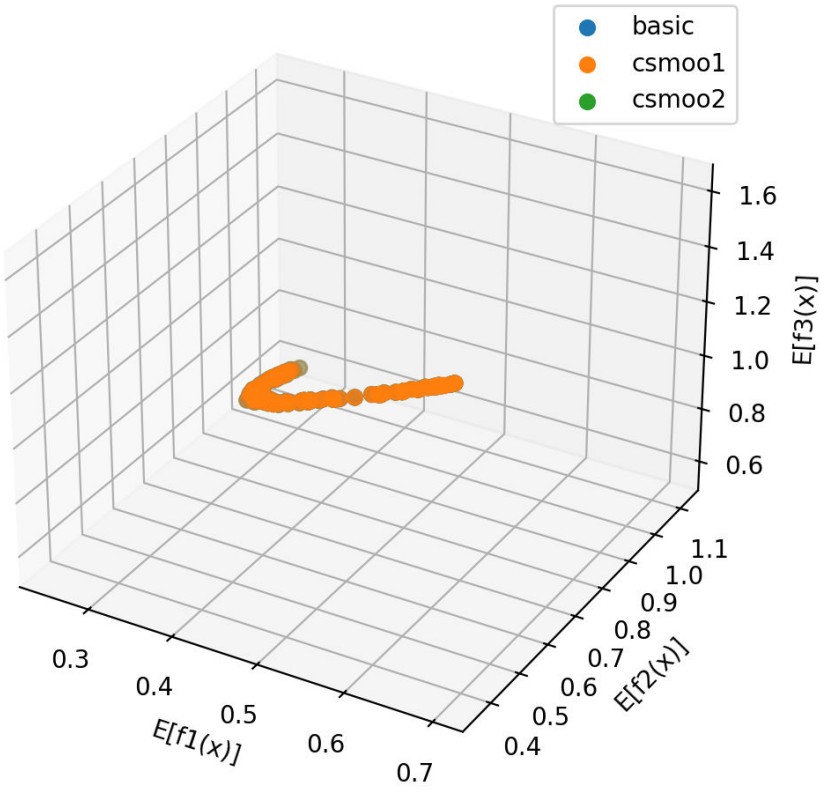

Figure 38: Estimated three-objective Pareto front on the augmented synthetic problem. Each marker corresponds to a distinct preference vector on a simplex grid; the curves from **basic**, **CSMOO-1**, and **CSMOO-2** are visually indistinguishable, indicating consistent convergence across algorithms.

