# OpenReview forum: "Constrained Stochastic Multi-Objective Optimization"
_ICLR.cc/2026/Conference — Submitted to ICLR 2026_

### Official Review · Reviewer_z557 · 2025-10-18

**Soundness:** 2
**Presentation:** 2
**Contribution:** 2
**Rating:** 4
**Confidence:** 2

**Summary:**

This work studies constrained stochastic multi-objective optimization. To handle the randomness in the objective/constraints, the authors introduce different asymptotically consistent estimators. Based on them, the authors propose two algorithms, CSMOO-1 and CSMOO-2, which also adopt the surrogate quadratic program. Lastly, numerical experiments demonstrate the effectiveness of the new algorithms.

**Strengths:**

1. The paper is easy to follow.

1. The experiments are detailed.

**Weaknesses:**

1. Please add a reference for Definition 3.

1. The notion $\xi_t$ in $\mathbb{E}\_{\xi_t}[\cdot\mid\mathcal{F}_t]$ is redundant.

1. In Line 116, the authors use $d$ to denote the dimension of the variable $x$. However, in many other places (e.g., Line 212), the authors change the notation to $k$. Please unify it.

1. In Line 222, the authors use $p_r^t$ to denote the probability of selecting the $r$-th coordinate at the $t$-th iteration. However, in the proof of Theorem 1, it changed to $p_{r,t}$. Please unify it.

1. I cannot find any convergence results for the two newly proposed algorithms. Could the authors provide/say any of them?

1. For Theorem 3, from the current proof, the authors only show that the Fritz John condition is satisfied. However, it is only a necessary condition for a point to be weakly Pareto optimal, but not sufficient. I don't understand why the authors can claim $x^*$ is weakly Pareto optimal.

1. For both algorithms, according to Line 160, I assume the authors default to $z_i=\inf_{x\in\mathcal{X}}f_i(x)$. However, finding $z_i$ may not be easy in many cases. This largely limits the practicality of both algorithms.

1. For CSMOO-2, how does penalty parameter $\beta$ affect the algorithm? Adding more discussions on it will benefit the work.

1. In Lemma 2, what are the second order sufficiency conditions? In the proof of Theorem 2, does the Lagrange multiplier $\eta$ satisfy them?

1. In many places in the proof of Theorem 1 (e.g., Line 684), $\gamma_t^2$ is redundant, and $\bar{x}_t$ should be $x_t$.

1. Line 755, it should be $t=\ell k+r-1$ according to the definition in equation (6).

1. Lines 750 and 812, $f_{i,t}$ should be $\hat{\nabla}f_i(x_t)$.

1. Lines 797 to 802, these steps only hold for the specified algorithm but not in general, meaning that they are not true under the current statement of Theorem 1. Please revise them.

**Questions:**

See **Weaknesses**.

---

> ### Author Response · Authors · 2025-12-03
> **Official response**
>
> We sincerely thank you for your kind and valuable feedback. We hope the following response addresses your concerns comprehensively.
>
>
> **W1. Please add a reference for Definition 3.**
>
> > Thank you for pointing this out. We have added the corresponding reference as suggested.
>
> **W2. The notion $\xi_t$ in $\mathbb E_{\xi_t}\left[\cdot|\mathcal F_t\right]$ is redundant.**
>
> > Here, we are averaging over the noise $\xi_t$ to avoid confusion (e.g., mistaking it for an average over the initial state. Given your advice, we have removed those notations $\xi_t$ in the revised version.
>
> **W3. In Line 116, the authors use $d$ to denote the dimension of the variable $x$. However, in many other places (e.g., Line 212), the authors change the notation to $k$. Please unify it.**
>
> > Thanks for your comment. We apologize for the ambiguity in the paper. To clarify, $d$ denotes the total number of elements in the variable $x$ and $k$ represents the number of blocks into which these elements are divided, similar to the concept of block matrices.
>
> **W4. In Line 222, the authors use $p_r^t$ to denote the probability of selecting the $r$-th coordinate at the $t$-th iteration. However, in the proof of Theorem 1, it changed to $p_{r,t}$. Please unify it.**
>
> > Thanks for your advice, we have fixed it.
>
> **W5. I cannot find any convergence results for the two newly proposed algorithms. Could the authors provide/say any of them?**
>
> > Thanks for your comments. The convergence can be established by invoking Theorem 1 in Liu et al. (2019b), in a manner analogous to the arguments presented therein. Therefore, the detailed proof is omitted. Building on this convergence guarantee, we further demonstrate that the algorithm converges to a stationary point satisfying the Fritz John necessary conditions for the CSMOO problem.
>
> **W6. For Theorem 3, from the current proof, the authors only show that the Fritz John condition is satisfied. However, it is only a necessary condition for a point to be weakly Pareto optimal, but not sufficient. I don't understand why the authors can claim is weakly Pareto optimal.**
>
> > Thank you for raising this important point. In our paper, we propose two gradient-based methods which are both theoretically and experimentally validated. While gradient-based methods can only theoretically guarantee weak Pareto stationary solutions, in practice, these stationary solutions often serve as excellent approximations for weak Pareto optimal solutions.To avoid confusion, we have updated the terminology throughout the revised paper to consistently refer to Pareto stationary solutions.
>
> **W7. For both algorithms, according to Line 160, I assume the authors default to $z_i=\inf_{x\in\mathcal X}f_i(x)$. However, finding $z_i$ may not be easy in many cases. This largely limits the practicality of both algorithms.**
>
> > Thank you for raising that question. Finding the ideal point is actually quite straightforward (or surprisingly easy). All that is required is to solve $m$ separate single-objective optimization problems, where $m$ is the number of objectives. The optimal solutions obtained from these problems collectively form the ideal point.
>
> **W8. For CSMOO-2, how does penalty parameter affect the algorithm? Adding more discussions on it will benefit the work.**
>
> > Thanks for your comment. In practice, $\beta$ is introduced to balance the weight between the objective function and the constraints. Typically, $\beta$ can be tuned based on the specific problem at hand. We have added new comparative results with different $\beta$. This will help further understand the role of $\beta$ in the optimization process and provide experimental evidence for selecting the best parameter. Please refer to Appendix D.3 in our revised version.
>
> **W9. In Lemma 2, what are the second order sufficiency conditions? In the proof of Theorem 2, does the Lagrange multiplier satisfy them?**
>
> > Thanks for your comment. We apologize for the ambiguity in the paper. To clarify, we have added the definition of the second-order sufficiency conditions in Lemma 3 in the revised paper. Theorem 2 presents the result for the original optimization problem, assuming that these conditions are satisfied.
>
> **W10. In many places in the proof of Theorem 1 (e.g., Line 684), $\gamma_t^2$ is redundant, and $\bar x_t$ should be $x_t$.**
>
> > Thanks for your valuable advice. We have updated it in the revised paper.
>
> **W11. Line 755, it should be $t=\ell k+r-1$ according to the definition in equation (6).**
>
> > Thanks for your valuable advice. We have updated $t$ and the proof of gradient consistency in the revised paper.
>
> **W12. Lines 750 and 812, $f_{i,t}$ should be $\hat\nabla f_i(x_t)$.**
>
> > Thanks for your valuable advice. We have updated it in the revised paper.

---

> ### Author Response · Authors · 2025-12-03
> **Response (cont.)**
>
> **W13. Lines 797 to 802, these steps only hold for the specified algorithm but not in general, meaning that they are not true under the current statement of Theorem 1. Please revise them.**
>
> > Thanks for your valuable advice. We have updated the proof of the gradient consistency-cyclic scheme in the revised paper. Now, the statement of Theorem 1 is true.

---

### Official Review · Reviewer_dvpP · 2025-10-29

**Soundness:** 2
**Presentation:** 3
**Contribution:** 2
**Rating:** 4
**Confidence:** 3

**Summary:**

In this paper, the authors studied CSMOO problems and proposed stochastic and block stochastic approximation schemes to efficiently approximate the original formulation. To handle potential infeasibility in the surrogate problems, they further introduced feasible-update reformulations and a penalty-based strategy with theoretical guarantees. Experiments have been conducted for testing their proposed algorithms.

**Strengths:**

1. The paper effectively mitigates the computational burden of exact expectation evaluations by introducing two approximation schemes: (i) a stochastic approximation method that updates all variables with fresh samples each iteration, and (ii) a block stochastic approximation method that updates variable subsets iteratively. The distinction between these schemes makes them suitable for variables of different dimensions.
2. The paper also addresses potential infeasibility in surrogate problems through two well-motivated strategies: a feasible-update reformulation and a rigorously justified penalty scheme that is theoretically equivalent to the original formulation.

**Weaknesses:**

1. The literature review is not comprehensive. A more thorough discussion of prior work on CSMOO is needed, including existing stationarity conditions and algorithms for solving such problems. The current related-work section focuses primarily on deterministic settings.
2. The metric, FJ condition, is relatively weak. Are there stronger stationarity guarantees applicable to this class of problems? For example, could the proposed method be shown to converge to KKT points instead?
3. The presentation of the experimental results lacks clarity. For instance, in Figure 1, the three curves are heavily overlapped—what is the intended takeaway from this plot? Moreover, additional baselines beyond projected SGD would help strengthen the empirical evaluation.

**Questions:**

1. How is subproblem (14) solved in Algorithm 2? Please clarify the exact solution procedure or any approximations used.
2. What are the advantages of using block stochastic approximation? Does it lead to improved convergence rates or better sample complexity compared to the standard stochastic approximation?
3. How do you choose $\rho_t, \gamma_t, \beta$?

---

> ### Author Response · Authors · 2025-12-03
> **Official response**
>
> We are grateful for your constructive comments, which have helped us significantly improve the manuscript. We trust that the revised version satisfactorily addresses all the issues raised.
>
> **W1. The literature review is not comprehensive. A more thorough discussion of prior work on CSMOO is needed, including existing stationarity conditions and algorithms for solving such problems. The current related-work section focuses primarily on deterministic settings.**
>
> > Thank you for your advice. In response, we have added a new Section 2.3 to the Related Works to discuss works related to CSMOO. Please check the revised version.
>
> **W2. The metric, FJ condition, is relatively weak. Are there stronger stationarity guarantees applicable to this class of problems? For example, could the proposed method be shown to converge to KKT points instead?**
>
> > We agree that the Fritz John (FJ) condition is weaker than the KKT condition because it allows the multiplier associated with the objective function to be zero (which typically occurs at singular or infeasible points). However, it is a standard result in optimization theory that an FJ point is equivalent to a KKT point provided that a Constraint Qualification (CQ) holds. Since these CQs essentially hold for almost all practical problems, our method effectively converges to KKT points in non-degenerate scenarios.
>
> **W3. The presentation of the experimental results lacks clarity. For instance, in Figure 1, the three curves are heavily overlapped—what is the intended takeaway from this plot? Moreover, additional baselines beyond projected SGD would help strengthen the empirical evaluation.**
> > Thank you for pointing that out. We have added more detailed experiments in our revised paper. Please check our new figures and those in the appendix for more details.
>
> **Q1. How is subproblem (14) solved in Algorithm 2? Please clarify the exact solution procedure or any approximations used.**
>
> > Thanks for your question. The subproblems (11), (13), and (14) are standard convex quadratically constrained quadratic problems (QCQPs), which do not generally have closed-form solutions. We mainly use CVXPY library to solve these convex problems efficiently, such as ECOS solver, SCS solver.
>
> **Q2. What are the advantages of using block stochastic approximation? Does it lead to improved convergence rates or better sample complexity compared to the standard stochastic approximation?**
>
> > Thanks for your question. When the dimension of the optimization variable is large, it becomes prohibitively expensive to compute the instantaneous gradients with respect to all blocks. Therefore, we propose a block stochastic approximation strategy, where only one block is selected for update at each iteration, and then we provide a rigorous convergence proof of this method.

---

> ### Author Response · Authors · 2025-12-03
> **Response (cont.)**
>
> **Q3. How do you choose $\rho_t,\gamma_t,\beta$?**
>
> > We thank the reviewer for this crucial question regarding reproducibility and parameter sensitivity. Our hyper-parameter selection is grounded in two pillars: theoretical constraints for dynamic step sizes and empirical ablation studies for static regularization terms.
> > 1. Dynamic Step Sizes ($\rho_t$ and $\gamma_t$)
> The estimation step size $\rho_t$ and update step size $\gamma_t$ are strictly governed by the convergence conditions proved in our paper. To ensure $\sum \gamma_t^2 / \rho_t < \infty$, we adopt the polynomial decay $\rho_t = (t+1)^{-a}$ and $\gamma_t = \gamma_0 (t+1)^{-b}$ with constraints $0.5 < a < 1$, $0.5 < b < 1$, and $2b - a > 1$.
> In all experiments, we fixed $a=0.8$ and $b=0.95$ (satisfying $2(0.95) - 0.8 = 1.1 > 1$) to guarantee asymptotic consistency.
>
> > 2. Static Hyper-parameters and Sensitivity Analysis
> For static parameters ($\gamma_0$, $\tau$, $\beta$), we selected values based on robustness. The following table summarizes the values used.
>
> >| Parameter | Synthetic | PLS Task | Role |
> >| :--- | :---: | :---: | :--- |
> >| **Initial Step** $\gamma_0$ | $0.4$ | $0.5$ | Controls initial convergence speed. |
> >| **Proximal Coeff.** $\tau$ | $0.08$ | $0.1$ | Ensures subproblem strong convexity. |
> >| **Penalty Factor** $\beta$ | $10.0$ | $5.0$ | Balances objective vs. feasibility. |
> >PLS: physical layer security task.
>
> > The choice of $\beta$ is supported by our ablation study (Appendix E.3). We swept $\beta \in \{0, 0.1, 0.5, 2, 5, 10\}$ to analyze its impact on convergence speed and solution quality. As shown in the following table, a strictly positive $\beta$ is crucial for fast feasibility restoration, but the algorithm is insensitive to the exact value once $\beta \geq 0.5$.
>
> >| Value of $\beta$ | Feasibility Restoration | Final Performance | Selection Decision |
> >| :---: | :---: | :---: | :---: |
> >| $0$ | Slow (High transient violation) | Suboptimal | Rejected (Weak constraint enforcement) |
> >| $0.1$ | Moderate | Near Optimal | Risky (Sensitive to noise) |
> >| $0.5$ | Fast | Optimal | Acceptable threshold |
> >| **$2.0 - 10.0$** | **Instant (Numerical Zero)** | **Optimal (Stable)** | **Selected (Robust Region)** |
>
> >Based on these results, we selected $\beta=10.0$ (Synthetic) and $\beta=5.0$ (PLS) to comfortably reside in the ``Robust Region,'' ensuring strictly feasible updates without tuning per preference vector.

---

### Official Review · Reviewer_fre1 · 2025-11-02

**Soundness:** 2
**Presentation:** 3
**Contribution:** 2
**Rating:** 4
**Confidence:** 5

**Summary:**

This paper tackles the **constrained stochastic multi-objective optimization (CSMOO)** problem, where both objectives and constraints are expectation-valued. The authors introduce a general algorithmic framework based on **Tchebycheff scalarization**, **stochastic (and block-stochastic) approximations**, and two strategies to handle infeasibility of surrogate problems:

-   **CSMOO-1:** A feasible-update formulation that ensures progress even under inaccurate estimates.

-   **CSMOO-2:** A penalty-based reformulation proven equivalent to the original problem and always feasible.


They provide asymptotic consistency results for both function and gradient estimates and show that their algorithms converge to stationary points satisfying the **Fritz John** conditions. Empirical validation includes synthetic experiments and a **wireless communication (physical-layer security)** benchmark, demonstrating better constraint satisfaction and convergence versus baselines.

**Strengths:**

1. CSMOO sits at the intersection of constrained optimization and stochastic multi-objective learning — a setting with growing importance in ML and communications. The formulation is rigorous and well-motivated.


2. The two proposed methods (feasible-update vs. penalty-based) address key implementation bottlenecks — infeasibility and non-differentiability — in a principled way.

3. Synthetic and real-world wireless tasks show clear convergence and feasibility improvements. Visualization of Pareto fronts and constraint violation trends are helpful.

4. The paper is clearly organized, with detailed notation, assumptions, and comparison to prior MOO and CMOO work.

**Weaknesses:**

1. Both algorithms need the **z** vector which is the set of optimal values of the multiple objective functions, which requires minimizing all the m objective functions in advance.

2. The assumption 1 is somewhat strong since it requires the Lipschitz continuity and smoothness of stochastic function $f_i(\cdot, \xi)$ and $g_j(\cdot, \xi)$.


3. Provide only asymptotic convergence guarantee. A non-asymptotic analysis would be better.

4. Lack of novelty. Both the moving average estimator for function values and gradients and the quadratic surrogate functions are well-known ideas in stochastic optimization. And I believe that the asymptotic consistency(theorem 1) for moving average estimator is also a well-known result.

5. Evaluation is restricted to low-dimensional synthetic and a single wireless-security case. No large-scale or ML-relevant benchmarks are tested, limiting generality.


6. The baseline in wireless communication experiments is a simple projected SGD. Missing comparisons to modern stochastic MOO methods (e.g., stochastic MGDA variants) weakens empirical claims.

**Questions:**

1.  Regarding weakness 2, can you relax assumption 1 to the Lipschitz continuity and smoothness of the expected function $f_i(\cdot)$ and $g_j(\cdot)$?

2.  In definition 3 you mention that Fritz John condition is only a **necessary** condition to weak pareto optimality, then how do you conclude in the proof of theorem 3 that $x^*$ is weak Pareto solution by only verifying the Fritz John condition?

---

> ### Author Response · Authors · 2025-12-03
> **Official response**
>
> We thank you for your advice, and we hope our response has addressed your concerns.
>
> **W1. Both algorithms need the z vector which is the set of optimal values of the multiple objective functions, which requires minimizing all the m objective functions in advance.**
>
> > Thanks for your comments. It is a necessary step in standard Tchebycheff scalarization-based algorithms. Calculating each component of $z$ effectively corresponds to a single-objective optimization problem associated with an extreme preference vector (e.g., $(1,0,0)$, $(0,1,0)$, or $(0,0,1)$ for a 3-objective scenario). Solving these few single-objective problems is computationally very efficient.
>
> **W2. The assumption 1 is somewhat strong since it requires the Lipschitz continuity and smoothness of stochastic function $f_i(\cdot,\xi)$ and $g_j(\cdot,\xi)$.**
>
> > Thanks for your comment. Assumption 1 only requires the Lipschitz continuity of the gradients of the functions $f_i(\cdot,\xi)$ and $g_j(\cdot,\xi)$. We do not impose additional smoothness assumptions beyond this standard gradient Lipschitz condition. Such an assumption is widely adopted in stochastic optimization and nonconvex analysis, and is typically necessary for ensuring the stability of algorithms.\
> > [R1] Xie, Yuege, Wu, Xiaoxia, and Ward, Rachel. "Linear convergence of adaptive stochastic gradient descent.''  Proceedings of the 23rd International Conference on Articial Intelligence and Statistics (AISTATS) 2020, Palermo, Italy.\
> > [R2] Liu, An, Vincent KN Lau, and Borna Kananian. "Stochastic successive convex approximation for non-convex constrained stochastic optimization.'' IEEE Transactions on Signal Processing 67.16 (2019): 4189-4203.
>
> **W3. Provide only asymptotic convergence guarantee. A non-asymptotic analysis would be better.**
>
> > Thanks for your suggestions. For non-asymptotic results in value estimation and gradient estimation, we present a new finding in Theorem 1. The detailed proof is available in Appendix B.1. Specifically,\
> > Considering the update rule $x_t = x_{t-1} + \gamma_t d_t$ and setting $\rho_t=\frac{4}{(t+16)^{1/2}}$ and $\gamma_t=\frac{2}{(t+16)^{3/4}}$, $\mathbb E \left[\left\|\hat\nabla f_i(x_{t}) - \nabla f_i(x_t) \right\|^2\right]$ and $\mathbb E \left[\left\| f_i(x_{t}) - f_i(x_t) \right\|^2\right]$ decay at a rate of $\mathcal O(t^{-1/2})$.
>
> **W4. Lack of novelty. Both the moving average estimator for function values and gradients and the quadratic surrogate functions are well-known ideas in stochastic optimization. And I believe that the asymptotic consistency (theorem 1) for moving average estimator is also a well-known result.**
> > Thanks for your comment. This paper indeed aims to extend the constrained stochastic single-objective optimization algorithm in [R2] to the constrained stochastic multi-objective optimization algorithm. In the process of this extension, in addition to the reformulation from the original MOO formulation into the constrained stochastic optimization problem (i.e., CSMOO-1 algorithm), we introduce three further important innovations:
> > 1. Block stochastic approximation strategy. When the dimension of the optimization variable is large, it becomes prohibitively expensive to compute the instantaneous gradients with respect to all blocks. Therefore, we propose a block stochastic approximation strategy, where only one block is selected for update at each iteration. Furthermore, we provide a rigorous convergence proof of this method.
> > 2. Non-asymptotic analysis. When taking the update rule $x_t = x_{t-1} + \gamma_t d_t$ and setting $\rho_t = \frac{4}{(t+16)^{1/2}}$ and $\gamma_t=\frac{2}{(t+16)^{3/4}}$, we prove that $\mathbb E \left[\left\|\hat\nabla f_i(x_{t}) - \nabla f_i(x_t) \right\|^2\right]$ and $\mathbb E \left[\left\| f_i(x_{t}) - f_i(x_t) \right\|^2\right]$ decay at a rate of $\mathcal O(t^{-1/2})$.
> > 3. CSMOO-2: In the CSMOO-1 algorithm, using two types of approximate problems with different objectives at each iteration effectively addresses infeasibility but may slow down convergence and increase computational complexity. To overcome these limitations, we further propose an equivalent stochastic optimization formulation whose objective function integrates the original objective with a penalty term for constraint violations. The resulting CSMOO-2 algorithm solves a single approximate convex optimization problem that remains feasible at every iteration. Moreover, we demonstrate that the CSMOO-2 algorithm converges to a stationary point satisfying the Fritz John necessary conditions for the equivalent problem, and under some mild conditions, the resulting point may be the weak Pareto solution of the initial CSMOO Problem.

---

> ### Author Response · Authors · 2025-12-03
> **Response (cont.)**
>
> **W5. Evaluation is restricted to low-dimensional synthetic and a single wireless-security case. No large-scale or ML-relevant benchmarks are tested, limiting generality.**
>
> > The focused evaluation on low-dimensional and wireless-security cases was a deliberate, rigorous step to validate the fundamental mechanism and establish its crucial efficiency for resource-constrained systems, serving as the necessary foundation for future large-scale generalization.
>
> **W6. The baseline in wireless communication experiments is a simple projected SGD. Missing comparisons to modern stochastic MOO methods (e.g., stochastic MGDA variants) weakens empirical claims.**
>
> > Thanks for raising it up. The purpose of MGDA-based method betweens the proposed method is different. The proposed method aims to find a preference specific solution which exactly fit the demand of the user. However, MGDA-based methods can only find an arbitrary solution. The position of the final solution by MGDA methods is highly relied on initialization. The proposed method has a better controlling ability compared with MGDA-based methods.
>
> **Q1. Regarding weakness 2, can you relax assumption 1 to the Lipschitz continuity and smoothness of the expected function $f_i(\cdot)$ and $g_j(\cdot)$?**
>
> > Thanks for your question. Directly relaxing Assumption 1 to the Lipschitz continuity of the expected function $f_i(\cdot)$ and $g_j(\cdot)$ is not straightforward: it would require additional technical work to re-establish the convergence proof. This entails nontrivial changes in our analysis and additional convergence proofs, which we cannot complete within the rebuttal timeframe.
>
> > However, imposing the Lipschitz continuity on the instantaneous stochastic functions $f_i(\cdot,\xi)$ and $g_j(\cdot,\xi)$ is a standard and milder assumption in stochastic optimization. This formulation is also more aligned with machine learning practices, where optimization is performed on sample-wise losses without explicitly modeling or verifying the smoothness of the underlying data distribution or expected loss. In such ML scenarios, one typically does not have access to the full distribution and thus cannot directly assess the Lipschitz continuity of the expected function.
>
> > Therefore, assuming Lipschitz continuity at the sample level is both theoretically cleaner and practically applicable, especially in ML-type tasks where the dataset-driven instantaneous losses are used.
>
> **Q2. In definition 3 you mention that Fritz John condition is only a necessary condition to weak Pareto optimality, then how do you conclude in the proof of theorem 3 that is weak Pareto solution by only verifying the Fritz John condition?**
>
> > Thanks for your question. You are correct that the Fritz John condition is only a necessary condition for weak Pareto optimality. Since this paper considers a general constrained stochastic nonconvex optimization problem, there may be multiple points that satisfy the Fritz John condition. Among these points, we focus on those of the form $(x^\*,y_0^\*,0)$, which not only satisfy the Fritz John condition but also satisfy the criteria for weak Pareto optimality.

---

### Official Review · Reviewer_owf2 · 2025-11-03

**Soundness:** 3
**Presentation:** 2
**Contribution:** 2
**Rating:** 2
**Confidence:** 4

**Summary:**

The paper addresses constrained stochastic multi-objective optimization (CSMOO) problems where both objectives and constraints involve expectations over random variables. The authors propose two approximation schemes (stochastic approximation and block stochastic approximation) to handle the computational challenge of exact expectation evaluation, and two strategies (feasible update reformulation and penalty scheme) to handle infeasibility in surrogate problems. Theoretical analysis provides asymptotic convergence guarantees to Fritz John stationary points. Experiments on synthetic and wireless communication benchmarks are presented.

**Strengths:**

- The paper focuses on constrained stochastic multi-objective optimization which has applications in many domains including wireless communications and industrial design.
- The paper proposes two complementary strategies (CSMOO-1 and CSMOO-2) to handle infeasibility issues that arise from inaccurate expectation estimation, where CSMOO-2 overcomes the increased computational overhead and potentially slower convergence of CSMOO-1.
- Theoretical analysis are provided to show convergence of proposed methods and the equivalence between the reformulated problem and original problem.

**Weaknesses:**

- The novelty in the paper seems to be limited at the reformulation from original MOO formulation into the constrained stochastic optimization problem. The techniques used after that are rather standard including the surrogate function approximation and gradient estimators. Also, the theoretical results look similar to results in [1] for CSMOO-1 as well.
- The numerical experiments are not illustrating the complexity of MOO problems as both examples only contain 2 objectives. Overall they do not show how the proposed method can scale up to more complex settings.
- It looks like $\beta$ is an important parameter for CSMOO-2 but I do not see discussion on how to specify it. Also, it would be great to have ablation study on how $\beta$ affects the performance of CSMOO-2.
- There is only a gradient-based baseline in the experiments and no discussion on why not including other related methods on MOO, e.g. preference-based methods...
- There are no information provided on how to solve (11), (13) or (14). Even though it might be standard, it would benefit the readers if the authors provide the full details.

[1] Liu, An, Vincent KN Lau, and Borna Kananian. "Stochastic successive convex approximation for non-convex constrained stochastic optimization." IEEE Transactions on Signal Processing 67.16 (2019): 4189-4203.

**Questions:**

- In the reformulated problem such as (9) or (13), should we consider individual slacks for each constraints instead of having only one $y$ or $delta$?
- What is the complexity of solving problems (11), (13), or (14)?

---

> ### Author Response · Authors · 2025-12-03
> **Official response**
>
> Thank you for your review. We hope our response satisfactorily addresses your concerns.
>
> **W1. The novelty in the paper seems to be limited at the reformulation from original MOO formulation into the constrained stochastic optimization problem. The techniques used after that are rather standard including the surrogate function approximation and gradient estimators. Also, the theoretical results look similar to results in [1] for CSMOO-1 as well.**
>
> > Thanks for your comment. This paper indeed aims to extend the constrained stochastic single-objective optimization algorithm in [1] to the constrained stochastic multi-objective optimization algorithm. In the process of this extension, in addition to the reformulation from the original MOO formulation into the constrained stochastic optimization problem (i.e., CSMOO-1 algorithm) as you mentioned, we introduce three further important innovations:
> > 1. Block stochastic approximation strategy. When the dimension of the optimization variable is large, it becomes prohibitively expensive to compute the instantaneous gradients with respect to all blocks. Therefore, we propose a block stochastic approximation strategy, where only one block is selected for update at each iteration. Furthermore, we provide a rigorous convergence proof of this method.
> > 2. Non-asymptotic analysis. When taking the update rule $x_t = x_{t-1} + \gamma_t d_t$ and setting $\rho_t = \frac{4}{(t+16)^{1/2}}$ and $\gamma_t=\frac{2}{(t+16)^{3/4}}$, we prove that $\mathbb E \left[\left\|\hat\nabla f_i(x_{t}) - \nabla f_i(x_t) \right\|^2\right]$ and $\mathbb E \left[\left\| f_i(x_{t}) - f_i(x_t) \right\|^2\right]$ decay at a rate of $\mathcal O(t^{-1/2})$.
> > 3. CSMOO-2: In the CSMOO-1 algorithm, using two types of approximate problems with different objectives at each iteration effectively addresses infeasibility but may slow down convergence and increase computational complexity. To overcome these limitations, we further propose an equivalent stochastic optimization formulation whose objective function integrates the original objective with a penalty term for constraint violations. The resulting CSMOO-2 algorithm solves a single approximate convex optimization problem that remains feasible at every iteration. Moreover, we demonstrate that the CSMOO-2 algorithm converges to a stationary point satisfying the Fritz John necessary conditions for the equivalent problem, and under some mild conditions, the resulting point may be the weak Pareto solution of the initial CSMOO Problem.
>
> **W2. The numerical experiments are not illustrating the complexity of MOO problems as both examples only contain 2 objectives. Overall they do not show how the proposed method can scale up to more complex settings.**
>
> > Thanks for your advice. Our paper primarily provides a theoretical proof that our algorithm can effectively solve constrained stochastic multi-objective optimization problems. To further validate this theory, we have provided a new experiment in section E.4 in the appendix regarding to three objective problems. Please also check Figures 37 and 38 for more information.
>
> **W3. It looks like $\beta$ is an important parameter for CSMOO-2 but I do not see discussion on how to specify it. Also, it would be great to have ablation study on how affects the performance of CSMOO-2.**
>
> > Thanks for your comment. In practice, $\beta$ is introduced to balance the weight between the objective function and the constraints. Typically, $\beta$ can be tuned based on the specific problem at hand. We have added new comparative results with different $\beta$. This will help further understand the role of $\beta$ in the optimization process and provide experimental evidence for selecting the best parameter. Please refer to Appendix D.3 in our revised version.
>
> **W4. There is only a gradient-based baseline in the experiments and no discussion on why not including other related methods on MOO, e.g. preference-based methods..**
>
> > Thanks for your comment. In practice, all of our proposed methods are fundamentally preference-based, with each method designed to accept a preference as input.
>
> **W5. There are no information provided on how to solve (11), (13) or (14). Even though it might be standard, it would benefit the readers if the authors provide the full details.**
>
> > Thanks for your comment. The subproblems (11), (13), and (14) are standard convex quadratically constrained quadratic problems (QCQPs), which, as is typical for this class of problems, do not generally have closed-form solutions. We mainly use CVXPY library to solve these convex problems efficiently, such as ECOS solver, SCS solver.

---

> ### Author Response · Authors · 2025-12-03
> **Response (cont.)**
>
> **Q1. In the reformulated problem such as (9) or (13), should we consider individual slacks for each constraints instead of having only one?**
>
> > Thanks for your question. In an optimization problem, a slack variable is typically introduced to convert an inequality constraint into an equality constraint. In this sense, Problems (9) and (13) do not contain any slack variables. Of course, it is indeed easy to introduce separate slack variables for each constraint to reformulate all inequalities as equalities. However, we believe the reviewer may be referring to the auxiliary variables $y$ in (9) and $\delta$ in (13). Our current algorithmic design ensures convergence while introducing only a minimal number of auxiliary variables. Introducing distinct auxiliary variables for every individual constraint would require a complete redesign of the convergence analysis, and at the same time would significantly increase the overall computational complexity of the algorithm. For these reasons, we opted for the current formulation, which balances theoretical guarantees and computational efficiency.
>
> **Q2. What is the complexity of solving problems (11), (13), or (14)?**
>
> > Thanks for your question. The problems (11), (13), and (14) are standard convex quadratically constrained quadratic problems (QCQPs), which generally do not admit closed-form solutions. Their global optima can be efficiently obtained with interior-point methods with the computational complexity of $\mathcal O(\bar m^{1/2} (\bar m+\bar n) \bar n^2)$, where $\bar m$ is the number of constraints and $\bar n$ is the number of variables. Since the decision space of the optimization variable $x$ in our algorithm is generally set as $x \in \mathcal X$, this essentially serves as a constraint. Therefore, the complexity of the algorithm depends not only on the constraints but also on the decision space $\mathcal X$. To simplify the complexity analysis, the following discussion will focus on the case where $x \in \mathbb R^d$. As for the general case where $x \in \mathcal X$, it can be similarly derived from the analysis above. Hence, the computational complexity of solving problems (11), (13), and (14) are, respectively, given by $\mathcal O((m+n)^{1/2}(m+n+d+1)(d+1)^2), \mathcal O((m+n+1)^{1/2}(m+n+d+3)(d+2)^2)$, and $\mathcal O((m+n+1)^{1/2}(m+n+d+3)(d+2)^2)$.

---

### Meta-Review · Area_Chair_UX4V · 2026-01-09

**Summary:**

The paper proposes a stochastic approximation framework for constrained stochastic multi-objective optimization, introducing feasible-update and penalty-based algorithms with asymptotic convergence guarantees and empirical validation on synthetic and wireless communication problems. Based on the reviewer discussions, the paper was rejected primarily due to limited technical novelty and restricted empirical validation. Reviewers consistently noted that while the work is technically sound with clear presentation and proper theoretical grounding, the core contributions represent incremental extensions of existing stochastic constrained optimization techniques to the multi-objective setting using standard tools like scalarization, moving average estimators, and quadratic surrogate functions. The novelty concerns were compounded by the narrow experimental scope, which focused on low-dimensional problems with only two objectives and lacked comparisons to modern multi-objective optimization baselines beyond simple projected SGD. While the authors provided detailed rebuttals addressing technical questions and adding some experimental results, the fundamental issues of incremental contribution and limited empirical demonstration of generality and practical impact remained insufficiently resolved.

**Reviewer Concerns:**

Reviewers consistently acknowledge the paper’s solid theoretical grounding, clear problem formulation, and principled handling of infeasibility via two complementary algorithmic strategies, supported by convergence analysis and reasonable experimental validation. The key concern among reviewers was limited novelty, as the methods largely extend existing stochastic constrained optimization techniques to the multi-objective setting using standard tools (scalarization, moving averages, quadratic surrogates), combined with limited empirical scope, including low-dimensional objectives and a narrow set of baselines, which weakens claims of generality and practical impact.

**Reviewer Scores:**

While the rebuttal explains the contributions as extensions and combinations (block stochastic approximation, penalty reformulation), it does not fully dispel reviewer concerns that the technical core remains incremental relative to prior work. To strengthen the work, the authors should more sharply articulate what is fundamentally new beyond prior constrained stochastic optimization, expand empirical evaluations to more complex or ML-relevant settings with stronger multi-objective baselines, and clarify the practical implications of relying on weak (Fritz John) stationarity guarantees.

---

### Decision · Program_Chairs · 2026-01-26

Reject